# Partons as unique ground states of quantum Hall parent Hamiltonians: The case of Fibonacci anyons

**Mostafa Tanhayi Ahari[1,2], Sumanta Bandyopadhyay[3,4], Zohar Nussinov[3], Alexander Seidel,[3] and Gerardo Ortiz[1]\***

**1** Department of Physics, Indiana University, Bloomington, IN 47405, USA
**2** Department of Physics and Astronomy, University of California, Los Angeles, CA 90095, USA
**3** Department of Physics, Washington University in St. Louis, USA
**4** Nordita, KTH Royal Institute of Technology and Stockholm University, SE-106 91 Stockholm, Sweden

\* ortizg@iu.edu

## Abstract

We present microscopic, multiple Landau level, (frustration-free and positive semi-definite) parent Hamiltonians whose ground states, realizing different quantum Hall fluids, are parton-like and whose excitations display either Abelian or non-Abelian braiding statistics. We prove ground state energy monotonicity theorems for systems with different particle numbers in multiple Landau levels, demonstrate S-duality in the case of toroidal geometry, and establish complete sets of zero modes of special Hamiltonians stabilizing parton-like states, specifically at filling factor $\nu = 2/3$. The emergent Entangled Pauli Principle (EPP), introduced in Phys. Rev. B 98, 161118(R) (2018) and which defines the "DNA" of the quantum Hall fluid, is behind the exact determination of the topological characteristics of the fluid, including charge and braiding statistics of excitations, and effective edge theory descriptions. When the closed-shell condition is satisfied, the densest (i.e., the highest density and lowest total angular momentum) zero-energy mode is a unique parton state. We conjecture that parton-like states generally span the subspace of many-body wave functions with the two-body $M$-clustering property within any given number of Landau levels, that is, wave functions with $M$th-order coincidence plane zeroes and both holomorphic and anti-holomorphic dependence on variables. General arguments are supplemented by rigorous considerations for the $M = 3$ case of fermions in four Landau levels. For this case, we establish that the zero mode counting can be done by enumerating certain patterns consistent with an underlying EPP. We apply the coherent state approach of Phys. Rev. X 1, 021015 (2011) to show that the elementary (localized) bulk excitations are Fibonacci anyons. This demonstrates that the DNA associated with fractional quantum Hall states encodes all universal properties. Specifically, for parton-like states, we establish a link with tensor network structures of *finite* bond dimension that emerge via root level entanglement.

# 1   Introduction

Realistic many-body problems, in which interactions play an important role can rarely be exactly solved. Over the decades, a rather fruitful *modus operandi* for analyzing certain many-body systems has been to construct physically motivated variational wave functions. This particular approach has been extremely insightful and witnessed monumental successes in several arenas including the BCS theory of superconductivity [1] and Laughlin's description of the simplest odd-denominator Fractional Quantum Hall (FQH) states [2]. The investigation of numerous variational wave functions and associated "parent Hamiltonians" (i.e., Hamiltonians whose ground states are the posited variational wave functions) has attracted renewed attention. This has, perhaps, been most acute for the rich plethora of FQH states. Certain FQH states have, for some time by now, been suspected of featuring non-Abelian exchange statistics [3, 4]. Complementing variational techniques, many other celebrated theoretical frameworks have been advanced to investigate these systems. These notably include effective field theories [5,6], Jain's composite fermion picture [7], general parton constructions [8–11], and the study of spectral properties of pseudopotentials [12–15] that allows for a systematic expansion of general rotationally symmetric interactions. Pseudopotentials and parton states and, in particular, their connection are a central focus of our study.

In the current work, we will demonstrate that an extensive set of systems with only two-body interactions have ground states that represent arbitrary quantum Hall (QH) fluids. The kinetic energy will be quenched in low-lying Landau level (LL) states. The resulting associated Hamiltonians will be positive semi-definite operators whose densest (i.e, minimum total angular momentum consistent with the largest filling fraction) zero-energy modes realize particular Abelian or non-Abelian QH vacua. We will investigate the universal short-range components of these two-body interacting Hamiltonians in the presence of low-lying LLs mixing. By fixing the subspace determined by a chosen number of LLs, we will outline a general scheme to obtain such positive semi-definite, and frustration-free, parent Hamiltonians and investigate their many-body (zero-energy) ground states. By altering the number of LLs and pseudopotentials, we will determine FQH states at various filling fractions as ground states of those parent Hamiltonians.

The recent renewed interest in parton-like FQH states [8–11, 16] is, in part, driven by the advent of new platforms for the physics of the QH effect, specifically graphene and related structures. Notably, in multi-layer graphene a degeneracy or near degeneracy of multiple LLs [10, 17–19] invites a study through guiding principles based on mixed-LL wave functions. On the other hand, contrary to the multiple-LL arena, powerful tools to identify the universality class of (especially non-Abelian) FQH trial wave functions have traditionally favored holomorphic, lowest Landau level (LLL), guiding principles. The seminal insights by Moore and Read [3] on conformal block-type holomorphic wave functions and their direct association to an edge effective theory allow for an unambiguous transition between microscopic wave functions and universal physics. It is, a priori, not clear how to achieve such a conversion between microscopic and universal properties, in similarly general terms, for non-holomorphic, multiple-LL wave functions. Our recent work [14, 15], however, suggests that such a tool is now emerging, specifically for a large class of states falling into a paradigm which we called

the "Entangled Pauli Principle" (EPP). In this work, we will elaborate why this, in particular, includes all parton-like states.

Our approach rests on three pillars. First, we establish one-dimensional reductions for the states in question as well as their quasihole/edge excitations. This relies on the generalization of concepts involving "dominance" or "root patterns", first discussed for holomorphic LLL wave functions [20–25], to the non-holomorphic case. The crucial enrichment resulting from this generalization is that root states also become locally entangled, as opposed to their holomorphic counterparts. These root states can be understood as the "DNA" of the underlying QH states [14]. This understanding arguably becomes complete only if one allows for the possibility of entanglement, as some of us recently demonstrated for (Abelian) Jain composite fermion states [15].

The second pillar involves the machinery used to derive the aforementioned EPPs not as properties of trial wave functions, but as necessary criteria satisfied by "root states" of zero-modes of an associated parent Hamiltonian. This step depends crucially on the correct generalization of the concept of "dominance" from the holomorphic wave function context to that of mixed-LL wave functions. It is central to establishing the full zero-mode space of the given Hamiltonian, thus replacing the formalism based on symmetric polynomials characteristic of the LLL context. This formalism is generally not applicable to non-holomorphic wave functions. Through matching of mode counting with an appropriate conformal field theory (CFT), the correct edge theory can, in principle, be identified beyond doubt, within the setting of microscopic wave functions and their parent Hamiltonians. We have demonstrated this procedure for a variety of pseudopotential and other frustration-free Hamiltonians in Refs [14], and [15], leading to a variety of non-holomorphic wave functions of interest. As we argued, the identification of universal physics rests on as solid grounds as it does for any holomorphic, LLL, wave function. The detailed structure of the EPP, however, depends on the parent Hamiltonians themselves. These details of the EPP are necessary to establish the connection between the microscopic ground state and the corresponding edge excitations. To streamline the flow of the logic, we have concentrated on a particular Hamiltonian (Trugman-Kivelson Hamiltonian [13], projected onto four LLs) to further establish the broad applicability of these techniques. This Hamiltonian is a particular type of positive semi-definite projected density-density interaction, which enforces a certain analytic clustering condition in its zero modes. This is analogous to similar interactions for simpler parton states, [10, 14] the $\nu = 2/5$ Jain state [26–28], and indeed the Laughlin state itself. [12, 13] The formalism is, however, not limited to density-density interactions. Indeed, as the example of general members of the Jain sequence shows, [15] more intricate action on Landau level indices is both needed and tractable within our formalism in order to stabilize states characterized by different types of non-holomorphic clustering conditions.

The third pillar concerns the bulk properties of the system more directly. It consists of a method to work out the statistics of the quasiparticles directly from the DNA as defined by the EPP. While the EPP efficiently encodes field theoretic concepts such as fusion rules [29, 30], our method is different in that it is not built on the assumptions of an effective theory that adheres to the axioms of local quantum field theory [31, 32]. In particular, no explicit contact with modular tensor categories is made. Instead, the formalism proceeds based on the knowledge that a complete set of quasihole excitations is encoded in patterns satisfying the EPP, and on an Ansatz of how localized quasihole excitations can be expressed through coherent states formed from a basis that is in one-to-one correspondence with these patterns. Consequences of locality and S-duality on the torus are naturally enforced within this Ansatz, without reliance on suppositions regarding underlying field-theoretic frameworks. This formalism, too, has been first worked out in the context of holomorphic LLL wave functions [21, 33–36]. As we will see, through the notion of an EPP, the formalism generalizes to the context of mixed-LL wave

functions, where one has to consider the entire root state with its entanglement as opposed to simple root patterns previously used in the LLL case [33]. It is here where the approach unfolds its full utility, as alternative methods to ascertain the statistics and underlying topological quantum field theory are far less abundant and general. The present formalism offers a general, consistent and highly constraining approach to determine field theoretic makeup from microscopic principles.

Interestingly, our approach provides a microscopic many-body account for long-sought excitations exhibiting non-trivial anyonic exchange statistics. Non-Abelian anyons are essential for viable topological quantum computing platforms [37]. Ising anyons have been earlier identified as excitations of the Moore-Read [3] (MR) Pfaffian and Jain-221 vacua [14]. However, Ising anyons cannot realize universal topological gates. By contrast, the non-Abelian Fibonacci anyons obey integer $SU(2)_3$ (or, equivalently, $SO(3)_3$) fusion algebra [38] allowing for universal quantum computation [39,40]. In this paper, we will pay particular attention to the subspace of four LLs. We will compute the Berry (more precisely, the Wilczek-Zee [41,42]) phase and braiding matrix associated with the braiding of zero-mode excitations [33,34], and show that the four LLs ground state precisely features Fibonacci anyons. Prior to our work, it was known that excitations of FQH Hamiltonians with k-body (k > 2) interactions exhibit Fibonacci anyons. This is the case of the k = 4 Read-Rezayi (RR) state [43,44] which can be obtained from correlation functions of certain CFTs. Important differences exist between our results and the prominent candidate RR state. Our Hamiltonian only contains (k = 2) *two-body* interactions projected onto four LLs as opposed to a (k = 4) four-body interacting Hamiltonian with an RR ground state in the LLL. Related to this, our ground state has order $M > 1$ zeros on a two-body (as opposed to a k-body) coincidence plane. Finally, our state appears at a filling fraction of $\nu = 2/3$, whereas the RR state corresponds to $\nu = 3/5$. Several earlier investigations depicted putative $\nu = 2/3$ Abelian and non-Abelian phases in terms of a bilayer FQH system featuring a 1/3 Laughlin state in each layer [45–49]. In these works, different phases were found when varying interlayer and intralayer interactions of the Hamiltonian. In particular, in Refs. [45] and [46] a stable phase with Fibonacci anyon quasiparticles has been obtained in the thin torus limit. Contrary to these previous studies, our Hamiltonian has no free parameters. Moreover, our exact calculations are not, in any way, restricted to the thin torus limit.

In addition, we establish a profound connection between the theory of (anti-)symmetric multivariate polynomials in holomorphic and anti-holomorphic variables, displaying special clustering properties, and the zero-modes of certain QH Hamitonians. In first quantization, a state that is a product of $M$ Slater determinants, formed out of single-particle orbitals, is a parton-like state. Correspondingly, a closed-shell parton state is a parton-like state with Slater determinants that have the lowest possible total angular momentum (in the case of Landau orbitals), rendering them unique. A closed-shell constraint provides the necessary and sufficient condition for the existence of unique densest parton-like states, which can be classified according to the order of their zeros in the vicinity of coincidence planes. The algebraic order of these zeros relates to the two-body $M$-clustering exponents for arbitrary particle pairs in the wave function. As will be discussed and proved for some cases, parton-like states span the subspace of many-particle wave functions with the two-body $M$-clustering property. Furthermore, we will demonstrate that both the closed-shell condition and the fixed two-body clustering exponent, lead to a unique expression for the densest ground state of the corresponding frustration-free (two-body) QH parent Hamiltonian.

The remainder of this Introduction highlights the organization and original contributions of the current paper. In Section 2, we will sketch the formalism that we employ to obtain the frustration-free QH two-body parent Hamiltonian in the subspace of $N_L$ LLs. In Section 3, we discuss the determination of its ground states and, in particular, the densest ground state.

Here, the concept of EPP [14] will be made vivid for the case of four LLs. For the general class of k-body, positive semi-definite, parent Hamiltonians with multiple-LLs (and arbitrary internal degrees of freedom) we show that the ground state energy increases monotonically with the number of particles. Moreover, we introduce a pseudospin algebra, in terms of pseudofermion operators, that will turn out to be decisive to establish the EPPs. In Section 4, we prove an S-duality for our class of multiple-LL Hamiltonians in toroidal geometry, and show how this duality together with the EPP imply braiding statistics without leaving the microscopic setting. In general, for multiple-LL systems one requires a non-trivial generalization of the framework of Ref. [33] that utilizes the entanglement of root states, i.e., the EPP, since knowledge of the root pattern alone cannot establish the braiding statistics. Interestingly, for the case of four LLs, we will show that the excitations posses Fibonacci anyon statistics. In Section 5, we discuss more general propositions on parton states and relate the two-body $M$-clustering exponent to necessary and sufficient conditions for parton states to be the unique ground states of projected frustration-free QH Hamiltonians, providing general considerations and a simple application of our conjecture. Finally, we close the paper with Section 6 paying special attention to the case $M = 3$ in four LLs. We provide a simple algebraic recipe to determine the root pattern and state of an arbitrary parton-like state. Root states, or DNAs, are obtained as the solutions to entanglement rules, the EPPs, and encode universal features of the QH fluid. We will show that the underlying entanglement has a simple tensor network structure rendering the root states (fermionic) matrix-product states. The inverse problem, that is, given a root pattern, establishing the parton-like states compatible with such a pattern, is also addressed algorithmically. This step is crucial to argue for the (over)completeness of parton-like states in spanning the zero-mode subspace. We conclude by rigorously showing completeness in the case $M = 3$ and $N_L = 4$.

## 2   Frustration-free QH Hamiltonians

In this section, we present a general formalism for establishing the second-quantized frustration-free Hamiltonians of interacting electrons confined to two spatial dimensions in the presence of an applied (perpendicular to the plane) magnetic field. As long known [50], under the influence of such a magnetic field, electrons occupy LL orbitals. Strong interactions among electrons may, however, effectively lead to the occupation of multiple LLs that Jain denominated as Λ-levels [51]. We focus on two-body interactions with rotational and translational symmetry although the general formalism extends to k-body interactions with k > 2. It is therefore convenient to employ the relative angular momentum eigenstates in order to construct a basis.

### 2.1   Building a two-fermion basis

Consider electrons of mass $m_e$ and charge $e < 0$ moving on the infinite $xy$-plane in the presence of an external perpendicular magnetic field $\mathbf{B} = \nabla \times \mathbf{A} = -B\hat{z}$, $(B > 0)$. Let us start with a brief review of LL physics and establish the notation used in this paper. Denoting the ith particle's location in the plane by the complex number(s) $z_i = x_i + iy_i$ ($\bar{z}_i = x_i - iy_i$), the kinetic energy of $N$ electrons is given by,

$$H_K = \sum_{i=1}^{N} \frac{\pi_i^2}{2m_e} = \sum_{i=1}^{N} \left( \hat{n}_i + \frac{1}{2} \right) \hbar \omega_c \,, \tag{1}$$

where $\pi_i = -i\hbar\nabla_i - \frac{e}{c}\mathbf{A}(x_i, y_i)$ is the kinematic momentum, $\hbar$ the reduced Planck's constant, and $c$ the speed of light. Ladder operators $a_i$ and $a_i^\dagger$ given by,

$$a_i = \frac{i\ell}{\hbar\sqrt{2}}(\pi_{x_i} + i\pi_{y_i}), \qquad a_i^\dagger = \frac{-i\ell}{\hbar\sqrt{2}}(\pi_{x_i} - i\pi_{y_i}), \tag{2}$$

where $\ell = \sqrt{\frac{\hbar c}{|e|B}}$ is the magnetic length, define the (LL index) number operator $\hat{n}_i = a_i^\dagger a_i$, and $\omega_c = \frac{|e|B}{m_e c}$ the cyclotron frequency. One can also define a new set of dynamical variables,

$$b_i = \frac{1}{\ell\sqrt{2}}\bar{z}_i - a_i^\dagger, \qquad b_i^\dagger = \frac{1}{\ell\sqrt{2}}z_i - a_i, \tag{3}$$

which are known as the cyclotron-orbit-center or guiding center operators. The ladder operators $(a_i, b_i)$ with the algebra

$$[b_i, b_j^\dagger] = \delta_{ij} = [a_i, a_j^\dagger], \qquad [a_i, b_j] = [a_i, b_j^\dagger] = 0, \tag{4}$$

provide a complete description of LL physics, where single particle basis states are given by

$$|n_i, s_i\rangle = \frac{1}{\sqrt{n_i! s_i!}} a_i^{\dagger n_i} b_i^{\dagger s_i} |0, 0\rangle, \tag{5}$$

and the integers $n_i$ and $s_i$ are the eigenvalues of the number operators $\hat{n}_i$ and $\hat{n}_i^b = b_i^\dagger b_i$, respectively. The vacuum state $|0, 0\rangle$ is obtained by solving $a_i|0, 0\rangle = 0 = b_i|0, 0\rangle$, with $n_i = 0$ corresponding to the LLL. With the aid of the above operators, the total angular momentum operator of $N$ particles can be written as

$$\hat{J} = \sum_{i=1}^{N} \hat{J}_i, \quad \text{with} \quad \hat{J}_i = \hbar(\hat{n}_i^b - \hat{n}_i), \tag{6}$$

and, therefore, the single particle basis states satisfy

$$\hat{J}_i |n_i, s_i\rangle = \hbar(s_i - n_i)|n_i, s_i\rangle = \hbar j_i |n_i, s_i\rangle = J_i |n_i, s_i\rangle. \tag{7}$$

For two particles, raising and lowering operators in the center of mass coordinate frame are given by [52]

$$\begin{aligned} a_c &= \frac{1}{\sqrt{2}}(a_1 + a_2), & a_r &= \frac{1}{\sqrt{2}}(a_1 - a_2), \\ b_c &= \frac{1}{\sqrt{2}}(b_1 + b_2), & b_r &= \frac{1}{\sqrt{2}}(b_1 - b_2), \end{aligned} \tag{8}$$

where subindex $c$ stands for center of mass and $r$ for relative. Here, $\hat{n}_c + \hat{n}_r = \hat{n}_1 + \hat{n}_2$, with $\hat{n}_{c,r} = a_{c,r}^\dagger a_{c,r}$ (whose eigenvalues are $n_{c,r}$), and $\hat{n}_{c,r}^b = b_{c,r}^\dagger b_{c,r}$ (whose eigenvalues are $2j - m$ and $m$, respectively). Note that $j$ can be an integer or half-integer as $2j = n_c + n_r$ is always an integer. The relative and total angular momentum operators in the two-particle system are, respectively, given by

$$\hat{L}_r = \hbar(\hat{n}_r^b - \hat{n}_r), \qquad \hat{J} = \hbar(\hat{J}_1 + \hat{J}_2). \tag{9}$$

These center of mass frame operators enable the construction of a two-fermion basis. A normalized fermionic two-particle state of a definite relative angular momentum $L_r = \hbar(m - n_r)$ and total angular momentum $J = \hbar(2j - n_c - n_r)$ can be written as

$$|n_c, n_r, 2j - m, m\rangle = \frac{a_c^{\dagger n_c} a_r^{\dagger n_r} b_c^{\dagger 2j - m} b_r^{\dagger m}}{\sqrt{n_c! n_r! (2j - m)! m!}} |0, 0\rangle. \tag{10}$$

While the basis states in Eq. (10) are suitable to describe a system with rotational symmetry, the LL indices of the individual particles, $n_i$, are not fixed. Our aim, however, is to define a two-fermion basis confined in the subspace of $N_L$ lowest lying LLs, i.e., $0 \leq n_i \leq N_L - 1$, for i = 1, 2. To systematically generate the fermionic basis with a well-defined LL index for individual particles, we introduce the following fermionic basis states labeled by $\{n_1, n_2, j, m\}$

$$|I\rangle_F = G_\pm^{n_1, n_2} |0, 0, 2j-m, m\rangle = \sum_{n_c, n_r} C_{n_c n_r} |n_c, n_r, 2j-m, m\rangle \,, \tag{11}$$

where

$$G_\pm^{n_1, n_2} = \frac{1}{\sqrt{n_1! n_2! \, 2(1 + \delta_{n_1, n_2})}} (a_1^{\dagger n_1} a_2^{\dagger n_2} \pm a_1^{\dagger n_2} a_2^{\dagger n_1}),$$

$$C_{n_c n_r} = \langle n_c, n_r, 2j-m, m | I \rangle_F$$

$$= \sum_{s=0}^{n_r} \sum_{l=0}^{n_c} \frac{\sqrt{n_c! n_r!}}{\sqrt{2^{n_1 + n_2 + 1}(1 + \delta_{n_1, n_2})}}$$

$$\times \frac{(-1)^{n_r - s} \sqrt{(l+s)! (n_1 + n_2 - l - s)!}}{(n_c - l)! \, l! \, (n_r - s)! \, s!}$$

$$\times \left[ \delta_{l+s, n_1} \delta_{n_c + n_r - l - s, n_2} \pm \delta_{l+s, n_2} \delta_{n_c + n_r - l - s, n_1} \right].$$

The $+(-)$ sign is used whenever $m \in$ odd(even).

In a disk geometry, within the symmetric gauge $\mathbf{A}(x_i, y_i) = \frac{B}{2}(y_i \hat{x} - x_i \hat{y})$, as further elaborated on in Appendix A, we obtain

$$|I\rangle_F = \frac{1}{2\sqrt{1 + \delta_{n_1, n_2}}} \sum_{k=-j}^{j} \eta_k(j, m) |\alpha_1, \alpha_2\rangle \,. \tag{12}$$

Here, we employed the following $2 \times 2$ determinant $D_{\alpha_1 \alpha_2}(1, 2) = \langle z_1, \bar{z}_1; z_2, \bar{z}_2 | \alpha_1, \alpha_2 \rangle \equiv \langle 1, 2 | \alpha_1, \alpha_2 \rangle$, and $\alpha_i = (n_i, s_i)$,

$$D_{\alpha_1 \alpha_2}(1, 2) = \begin{vmatrix} \phi_{n_1, j-k}(1) & \phi_{n_1, j-k}(2) \\ \phi_{n_2, j+k}(1) & \phi_{n_2, j+k}(2) \end{vmatrix}, \tag{13}$$

where $\phi_{n_i, s_i}(i) = \langle z_i, \bar{z}_i | n_i, s_i \rangle \equiv \phi_{\alpha_i}$, for i = 1, 2, with $s_1 = j - k$ and $s_2 = j + k$. In coordinate representation

$$\phi_{n,s}(z, \bar{z}) = \frac{(-1)^n \sqrt{n!} \, e^{-\frac{z\bar{z}}{4\ell^2}}}{\sqrt{2\pi\ell^2} \sqrt{2^{s-n}s!}} \left(\frac{z}{\ell}\right)^{s-n} L_n^{s-n}\left(\frac{z\bar{z}}{2\ell^2}\right), \tag{14}$$

where $L_n^{s-n}(x)$ is the associated Laguerre polynomial.

The functional form of the coefficients $\eta_k(j, m)$ contains information about the geometry of the system [24]. For the disk geometry, it is given by

$$\eta_k(j, m) = (-1)^{m+j-k} \sqrt{\frac{(j-k)! (j+k)!}{2^{2j-1}(2j-m)! m!}} \sum_{q=0}^{j-k} (-1)^q \binom{2j-m}{q} \binom{m}{j-k-q}. \tag{15}$$

For a given $j$, each state $|I\rangle_F$ will be specified by the reduced set $\{n_1, n_2, m\}$. By imposing $0 \leq n_i \leq N_L - 1$, the basis states of Eq. (12) span a two-fermion basis projected onto the subspace of the lowest $N_L$ LLs.

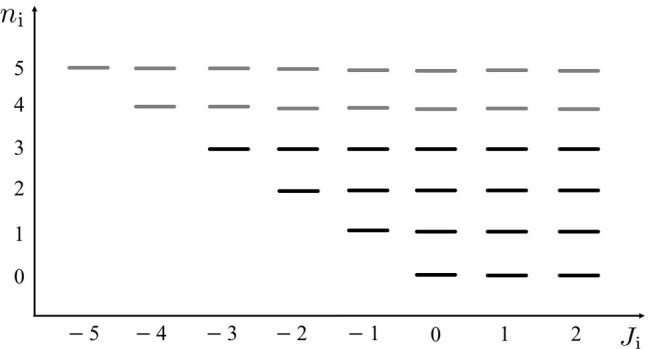

Figure 1: Projection onto the lowest $N_L = 4$ LLs (black solid bars) with $L = 6$. Each solid bar represents a LL orbital $\phi_{n_i,s_i}$ with horizontal and vertical axis representing angular momentum $J_i$ (in units of $\hbar$) and LL index $n_i$, respectively.

We can express the two-fermion basis in a second quantization representation. This is especially advantageous when discussing the QH parent Hamiltonian projected onto the subspace of $N_L$ LLs, and its ground states. Equation (13) suggests a natural map

$$\frac{1}{\sqrt{2}} D_{\alpha_1 \alpha_2} \rightarrow c^\dagger_{n_1,j-n_1-k} c^\dagger_{n_2,j-n_2+k} |0\rangle . \tag{16}$$

Here, $|0\rangle$ is the Fock space vacuum and $c^\dagger_{n,l}$ ($c_{n,l}$) are fermionic creation (annihilation) operators, creating (annihilating) an electron with LL index $n$ and angular momentum $\hbar l$. Thus, one may transition from the fermionic states of Eq. (12) to a second quantized representation by a replacement of the type [24]

$$|I\rangle_F \rightarrow T^{n_1,n_2 +}_{j;m} |0\rangle , \tag{17}$$

where

$$T^{n_1,n_2 +}_{j;m} = \frac{1}{\sqrt{2(1+\delta_{n_1,n_2})}} \sum_{k=-j-n_2}^{j+n_1} \eta_{k+\frac{n_2-n_1}{2}} \left( j + \frac{n_2+n_1}{2}, m \right) c^\dagger_{n_1,j-k} c^\dagger_{n_2,j+k} . \tag{18}$$

It can be checked that the two-fermion operators $T^{n_1,n_2 +}_{j;m}$ satisfy

$$\langle 0| T^{n_1,n_2 -}_{j;m} T^{n_1,n_2 +}_{j;m} |0\rangle = 1 , \tag{19}$$

where $T^{n_1,n_2 -}_{j;m} = (T^{n_1,n_2 +}_{j;m})^\dagger$.

So far, we examined a system on a plane of an unbounded spatial extent. For finite size systems, the number of angular momentum orbitals in each LL is restricted by the number $L$ of available distinct single particle angular momentum modes. As an example, in Fig. 1 we depict the LL orbitals (solid bars) and project only up to four LLs (black solid bars). The horizontal axis represents the angular momentum of the LL orbitals and the vertical axis provides the LL index. Notice that the highest LL will always have $L$ orbitals.

For $N_L$ LLs, each single particle angular momentum mode may, at most, correspond to $N_L$ orthogonal orbitals. Consequently, in Eq. (18), $j$ must be restricted to the interval $[-N_L + 1, L - N_L]$. Assuming integer orbital numbers $j \pm k$, it can be checked that $j$ may assume the $2L-1$ consecutive values [24]

$$j = -N_L + 1, -N_L + 3/2, -N_L + 2, \cdots, L - N_L . \tag{20}$$

Here, $-\min(\tilde{j}, L-1-\tilde{j}) - \frac{n_1-n_2}{2} \leq k \leq \min(\tilde{j}, L-1-\tilde{j}) + \frac{n_1-n_2}{2}$, where $\tilde{j} = j + \frac{n_1+n_2}{2}$. (A word of caution: Whenever $j$ refers to angular momentum it must be an integer).

## 2.2 Projected two-body Hamiltonians

We next outline a simple general recipe for writing down QH parent Hamiltonians in terms of fermionic operators. The positive semi-definite property of these Hamiltonians will, importantly, give rise to a systematic way of generating ground states (zero-energy modes) for $N_L$ LLs. To this end, we utilize the two-fermion basis derived above and project a two-body QH Hamiltonian onto $N_L$ LLs. By expressing the projected Hamiltonian in a second quantized form, we show that the projected Hamiltonian is a "frustration-free Hamiltonian".

Consider a (repulsive) short range interaction potential

$$H_{\text{int}} = \sum_{i<j} V(\mathbf{r}_i - \mathbf{r}_j),\tag{21}$$

that enjoys rotational and translational symmetry. The pair interaction $V(\mathbf{r}_i - \mathbf{r}_j) = V(\mathbf{r}_{ij})$ can, generally, be represented as an infinite sum [52]

$$V(\mathbf{r}_i - \mathbf{r}_j) = \sum_{\alpha=0}^{\infty} V_\alpha L_\alpha(-\ell^2\nabla_{ij}^2)\,\delta^2(\mathbf{r}_i - \mathbf{r}_j),\tag{22}$$

where $L_\alpha(x)$ is the $\alpha^{\text{th}}$ Laguerre polynomial [53]. The expansion coefficients $V_\alpha$ can be determined from the specific form of the interaction, viz.,

$$V_\alpha = 4\pi\ell^2 \int \frac{d^2\mathbf{k}}{(2\pi)^2} \tilde{V}(\mathbf{k})\, L_\alpha(\ell^2 k^2)\, e^{-\ell^2 k^2},\tag{23}$$

where $\tilde{V}(\mathbf{k})$ is the Fourier transform of the potential (see Appendix B for a derivation). For the LLL, $\alpha$ can be identified with the relative angular momentum of the pair, and as such, $V_\alpha$ would represent the energy penalty for having a pair in such a state. This approach, known as the pseudopotential expansion, was first pioneered in the context of FQH physics and LLL by Haldane [12]. Generically, [13] Eq. (22) may be considered as an expansion of the interaction potential in powers of its range (magnetic length) $\ell$. This can be seen by noting that, for a ground state of $H_{\text{int}}$ with filling fraction $\nu$, a relevant correlation length is [50] $\sim \ell/\sqrt{\nu}$, proportional to the Wigner-Seitz radius. Thus, for a short range two-body interaction, it is typically sufficient to keep the first few pseudopotentials.

As shown below, the interaction potential $H_{\text{int}}$, when projected onto $N_L$ LLs, is a positive semi-definite and frustration-free operator. These universal properties may be made explicit by keeping $\alpha = 0, 1$,

$$V(\mathbf{r}_i - \mathbf{r}_j) = (V_0 + V_1 + V_1\ell^2\,\nabla_{ij}^2)\,\delta^2(\mathbf{r}_i - \mathbf{r}_j).\tag{24}$$

Due to the antisymmetry of the fermionic wave function, the first two terms on the righthand side of Eq. (24) have vanishing expectation values. Therefore, we analyze only $V(\mathbf{r}_i - \mathbf{r}_j) \equiv V_1\ell^2\nabla_{ij}^2\delta^2(\mathbf{r}_i - \mathbf{r}_j)$, as our interaction potential. We will refer to this potential as the Trugman-Kivelson (TK) [13] Hamiltonian

$$H_{\text{int}} = V_1\ell^2 \sum_{i<j} \nabla_{ij}^2\delta^2(\mathbf{r}_i - \mathbf{r}_j),\tag{25}$$

whose ground states satisfy the $M = 3$-clustering property in the coordinate representation. For ground states satisfying the $M > 3$-clustering property we should either consider higher-order terms in the pseudopotential expansion (see Appendix B), assuring its positive semi-definite character, or engineer special positive semi-definite Hamiltonians with gradient-density expansions [15].

The spectral decomposition of the Hamiltonian in the projected two-fermion basis reads

$$\hat{H}_{\text{int}} \equiv P_{N_L} H_{\text{int}} P_{N_L} = \sum_j \sum_\xi E_\xi |\xi\rangle\langle\xi|. \tag{26}$$

Here, $P_{N_L}$ represents the projection operator onto the $N_L$ LLs and $|\xi\rangle = \sum_I \Lambda_I^\xi |I\rangle_F$ are the eigenvectors of the interaction in the two-fermion basis $|I\rangle_F$ with expansion coefficients $\Lambda_I^\xi$. The index $I$ runs over the entire two-fermion basis in the subspace of $N_L$ LLs. The positive semi-definite property of the Hamiltonian is evident when $E_\xi \geq 0$, as will be demonstrated for the case of four LLs. Putting all of the pieces together, the TK Hamiltonian may be expressed as a sum over angular momentum terms [24],

$$\hat{H}_{\text{int}} = \sum_j \hat{H}_j, \tag{27}$$

where $\hat{H}_j = \sum_\xi E_\xi \mathcal{T}_j^{\xi+} \mathcal{T}_j^{\xi-}$ is a positive semi-definite operator with

$$\mathcal{T}_j^{\xi+} = \sum_I \Lambda_I^\xi T_{j,m_I}^{n_1,n_2+}, \qquad \mathcal{T}_j^{\xi-} = (\mathcal{T}_j^{\xi+})^\dagger. \tag{28}$$

Note that for the Hamiltonian in Eq. (27), in general, $[\hat{H}_j, \hat{H}_{j'}] \neq 0$ for $j \neq j'$. Nevertheless, there can be a common zero-energy state. In the subspace of $N_L$ LLs, $E_\xi \geq 0$, and a zero-energy state may appear if and only if $\hat{H}_j |\Psi_0\rangle = 0$ for all $j$. Whenever such a zero-energy state exists (and as we will explain such states do indeed exist), the projected Hamiltonian is, by definition, a frustration-free Hamiltonian.

Obtaining the projected Hamiltonian for $N_L = 1, 2$ and $3$ LLs was previously explored [14, 24, 28]. This led to the discovery of non-trivial structures for $\nu = 2/5$ and $\nu = 1/2$ FQH ground states and their excitations. In the current paper, we will chiefly focus on $N_L = 4$ LLs.

### 2.3 QH Hamiltonian in the subspace of four LLs

The two-fermion basis, spanning the positive eigenvalue subspace of the TK Hamiltonian, for $N_L = 4$ LLs includes up to 40 vectors $|I\rangle_F$ (see Appendix C for their construction). This cutoff value, 40, includes all those basis vectors having non-vanishing matrix elements of the TK Hamiltonian. Diagonalizing the interaction matrix leads to only 12 nonzero eigenvalues,

$$
\begin{aligned}
E_\xi \in \frac{V_1}{4\pi} \Bigg\{ & \frac{325}{16}, \frac{323 + 47\sqrt{17}}{32}, \frac{69}{8}, \frac{31 + 3\sqrt{33}}{8}, \\
& \frac{323 - 47\sqrt{17}}{32}, \frac{75 + 7\sqrt{57}}{32}, \frac{31 - 3\sqrt{33}}{8}, \frac{13 + \sqrt{89}}{16}, \\
& \frac{13 + \sqrt{129}}{32}, \frac{75 - 7\sqrt{57}}{32}, \frac{13 - \sqrt{89}}{16}, \frac{13 - \sqrt{129}}{32}, 0, \cdots, 0 \Bigg\}.
\end{aligned}
\tag{29}
$$

We note that $E_\xi \geq 0$. Thus, the positive semi-definite Hamiltonian projected onto $N_L = 4$ LLs is given by

$$\hat{H}_{\text{int}} = \sum_j \sum_{\xi=1}^{12} E_\xi \mathcal{T}_j^{\xi+} \mathcal{T}_j^{\xi-}, \tag{30}$$

where in the operators $\mathcal{T}_j^{\xi+}$ each individual operator $T_{j,m}^{n_1,n_2+}$ is specified by a set of numbers $\{n_1, n_2, m\}$ as given in Table 8 of Appendix C. The expansion coefficients $\Lambda_I^\xi$ are also given in Appendix C.

# 3 Ground states of QH Hamiltonians

By its nature, any positive semi-definite Hamiltonian can only have non-negative eigenvalues. Thus, any non-trivial zero-energy eigenstate of Eq. (27), if it exists, will be a ground state which satisfies

$$\hat{H}_j|\Psi_0\rangle = 0 \quad \forall j \Longleftrightarrow \mathcal{T}_j^{\xi^-}|\Psi_0\rangle = 0 \quad \forall \xi, j. \tag{31}$$

These zero-energy states collectively exhaust the ground state manifold. As we will explain, one may indeed precisely find all existing zero-energy states for given number of particles $N$ at filling fractions $\nu = (N-1)/(L-1)$. The filling fraction of the ground state, on the other hand, determines the electron density $\rho = (B/\phi_0)\nu$, where $\phi_0 = hc/|e|$ is the electron's magnetic flux quantum. Therefore, exploring the ground states of the Hamiltonian family considered here leads to candidate incompressible states with different LL filling factors. Assuming that we have determined a set of zero-energy ground states from Eq. (31), an important question is whether adding or removing electrons may increase the ground state energy. The answer to this question establishes the relationship between the electron density and the ground state energy of the FQH state, which we will explore in the next subsection.

## 3.1 Monotonicity of the ground state energy

The kinetic energy in our particle number conserving system is quenched; the system is dominated by interparticle interactions. An interesting question for a general system with k-body interactions is what is the relation between the ground state energies of $N$ and $N-n$ ($n > 0$) particles when the total number of available states is fixed. As demonstrated in Ref. [54], for general k-body interaction positive semi-definite Hamiltonian, the energy of the ground state is monotonically increasing in the number of particles. Reference [54] focused on flavorless and spinless electrons (thus, spinless electrons confined only to the LLL). In what follows, we generalize this earlier result to a broader setting in which the electrons may have several internal degrees of freedom (such as the LL index, spin and angular momentum).

Consider a general k-body Hamiltonian,

$$H_k = \sum_{[\mathbf{n}]} V_{[\mathbf{n}]} \, c_{\mathbf{n}_1}^\dagger c_{\mathbf{n}_2}^\dagger \cdots c_{\mathbf{n}_k}^\dagger c_{\mathbf{n}_{k+1}} c_{\mathbf{n}_{k+2}} \cdots c_{\mathbf{n}_{2k}}, \tag{32}$$

where $\mathbf{n}_l = (n_l^1, n_l^2, \cdots)$, $l = 1, 2, \cdots, 2k$, represents a set of labels such as the band (or LL) index, spin, angular momentum, etc., and $[\mathbf{n}] = \{\mathbf{n}_1, \cdots, \mathbf{n}_{2k}\}$. Note that the Hamiltonian $H_k$ conserves the number of particles,

$$[H_k, \hat{N}] = 0. \tag{33}$$

Here, $\hat{N} = \sum_{\mathbf{n}_q} c_{\mathbf{n}_q}^\dagger c_{\mathbf{n}_q}$. We next consider an $N'$-particle density matrix $\rho_{N'}$ and further define

$$\rho_{N'-1} = \frac{1}{N'} \sum_{\mathbf{n}_q} c_{\mathbf{n}_q} \rho_{N'} c_{\mathbf{n}_q}^\dagger, \tag{34}$$

such that $\hat{N}\rho_{N'} = \rho_{N'}\hat{N} = N'\rho_{N'}$. This implies that $\text{Tr}[\rho_{N'}] = \text{Tr}[\rho_{N'-1}] = 1$. We next establish the following identity

$$\text{Tr}[\rho_{N'-1} H_k] = \frac{N'-k}{N'} \, \text{Tr}[\rho_{N'} H_k]. \tag{35}$$

To show this, we first compute

$$\text{Tr}[\rho_{N'} H_k] = \frac{1}{N'} \text{Tr}[\rho_{N'} \hat{N} H_k], \tag{36}$$

and use the operator identity

$$\hat{N} H_k = k H_k + \sum_{\mathbf{n}_q} c^\dagger_{\mathbf{n}_q} H_k c_{\mathbf{n}_q}, \tag{37}$$

to obtain

$$\mathrm{Tr}[\rho_{N'} H_k] = \frac{k}{N'} \mathrm{Tr}[\rho_{N'} H_k] + \frac{1}{N'} \mathrm{Tr}\Big[ H_k \sum_{\mathbf{n}_q} c_{\mathbf{n}_q} \rho_{N'} c^\dagger_{\mathbf{n}_q} \Big]$$

$$= \frac{k}{N'} \mathrm{Tr}[\rho_{N'} H_k] + \mathrm{Tr}[\rho_{N'-1} H_k]. \tag{38}$$

This indeed establishes the identity in Eq. (35).

Now, if $N' < N$ (setting $N' = N - \mathrm{n}$) then, by induction,

$$\mathrm{Tr}[\rho_{N-\mathrm{n}} H_k] = [N, \mathrm{n}, k] \, \mathrm{Tr}[\rho_N H_k], \tag{39}$$

where

$$[N, \mathrm{n}, k] = \frac{(N - \mathrm{n} + 1 - k)(N - \mathrm{n} + 2 - k)\cdots(N - k)}{(N - \mathrm{n} + 1)(N - \mathrm{n} + 2)\cdots N}.$$

If $\rho_N$ is chosen such that the ground state energy $E_0(N) = \mathrm{Tr}[\rho_N H_k]$, then by the Ritz variational principle [55], we get

$$\mathrm{Tr}[\rho_{N-\mathrm{n}} H_k] \geq E_0(N - \mathrm{n}). \tag{40}$$

Equivalently,

$$E_0(N - \mathrm{n}) \leq [N, \mathrm{n}, k] \, E_0(N). \tag{41}$$

If the Hamiltonian is a positive semi-definite operator then $E_0(N) \geq 0$ for any $N$ and

$$E_0(N - \mathrm{n}) \leq [N, \mathrm{n}, k] \, E_0(N) \leq E_0(N).$$

For the particular case of $\mathrm{n} = 1$, we find that

$$E_0(N - 1) \leq \frac{N - k}{N} E_0(N) \leq E_0(N). \tag{42}$$

This inequality proves the monotonicity of the ground state energy. Equation (42) allows for the inclusion of general LLs and angular momentum ($\mathbf{n}_i = (n, j)$) indices. Thus, if a zero-energy ground state exists for a given density then, for all lower electron densities, the ground state energy must strictly vanish.

The above demonstration of monotonicity may be generalized to a linear combination of k-body interactions [54]

$$H = \sum_{k=k_{\min}}^{k_{\max}} H_k, \tag{43}$$

with $k_{\max} - k_{\min} \geq 0$. From Eq. (39),

$$\mathrm{Tr}[\rho_{N-\mathrm{n}} H] = [N, \mathrm{n}, k_{\min}] \mathrm{Tr}[\rho_N H] + \delta E. \tag{44}$$

Here,

$$\delta E = \sum_{k=k_{\min}}^{k_{\max}} ([N, \mathrm{n}, k] - [N, \mathrm{n}, k_{\min}]) \mathrm{Tr}[\rho_N H_k]. \tag{45}$$

Similar to the above, if $E_0(N) = \mathrm{Tr}[\rho_N H]$ then the Ritz variational principle mandates that

$$E_0(N - \mathrm{n}) \leq [N, \mathrm{n}, k_{\min}] E_0(N) + \delta E. \tag{46}$$

Note that in (45) the term in parenthesis is negative semi-definite since $[N, \mathrm{n}, k_{\min}] \geq [N, \mathrm{n}, k_{\min} + \delta k] \geq 0$ when $0 \leq \delta k \leq k_{\max} - k_{\min}$. Then, whenever $\delta E \leq 0$

$$E_0(N - \mathrm{n}) \leq [N, \mathrm{n}, k_{\min}] E_0(N) \leq E_0(N), \tag{47}$$

which is in particular guaranteed if all $H_k$'s are positive semi-definite.

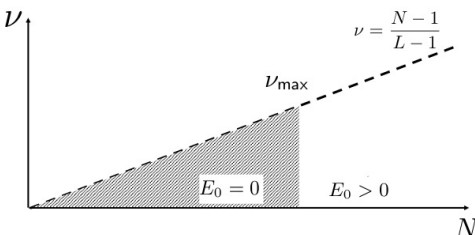

Figure 2: All ground states of positive semi-definite Hamiltonians of the form of Eq. (32), with densities $\nu$ less or equal to the maximal density $\nu_{\text{max}}$ are zero-energy states. For densities exceeding that threshold value, $\nu > \nu_{\text{max}}$, the ground states have positive energies.

## 3.2  Determining the densest zero-energy mode: Entangled Pauli principle (EPP)

Here, we explicitly determine the ground state of the projected Hamiltonian in Eq. (27) for an $N$-particle system. Intuitively, the densest ground state of this type of Hamiltonians corresponds to an incompressible QH liquid. The monotonicity that we established in the previous subsection indeed indicates that if we find a zero-energy ground state with filling fraction $\nu$, then for all the smaller filling fractions (with fixed $L$) the ground states will also be zero-energy eigenstates, i.e., $E_0 = 0$. (This is schematically illustrated in Fig. 2.)

Since the spatial extent of LL orbitals is directly associated with the magnitude of its angular momentum, when there are several zero-energy states with the same bulk filling fraction $\nu$, we will define the one with smallest total angular momentum $J$ to be the densest state. When alluding to "the ground state", we will mainly refer to the densest zero-energy ground state.

The ground state $|\Psi_0\rangle$ can be written as a linear superposition of Slater determinants in the occupation number representation basis,

$$|\Psi_0\rangle = \sum_{\mathfrak{n}} C_{\mathfrak{n}} |\mathfrak{n}\rangle \,, \tag{48}$$

with coefficients $C_{\mathfrak{n}} \in \mathbb{C}$. Each basis state $|\mathfrak{n}\rangle$ (a single Slater determinant),

$$|\mathfrak{n}\rangle = c^{\dagger}_{n_1,j_1} c^{\dagger}_{n_2,j_2} \cdots c^{\dagger}_{n_N,j_N} |0\rangle \,, \qquad j_1 \leq j_2 \leq \cdots \leq j_N \,,$$

is associated with a total angular momentum partition [56] (due to the rotational symmetry of the projected Hamiltonian) $\{\lambda\} = \lambda_{j_1} \lambda_{j_1+1} \cdots \lambda_{j_i} \cdots \lambda_{j_N}$. Here, $0 \leq \lambda_{j_i} \leq N_L$ represents the multiplicity of occupied orbitals with fixed angular momentum $\hbar j_i$, where $j_{\text{min}} = -N_L + 1$ is the lowest possible value, and $j_{\text{max}} = L - N_L$ the largest possible one. A "multiplicity" $\lambda_{j_i} > 1$ implies that electrons occupy orbitals with the same angular momentum $\hbar j_i$ and different LL index $n$. An equivalent alternative notation for the occupation number configuration is afforded by $\{j_1, \cdots, j_N\}$. For instance, the $N = 3$ Slater determinant

$$c^{\dagger}_{n,j_1} c^{\dagger}_{n',j_1} c^{\dagger}_{n'',j_1+2} |0\rangle = |2_{n,n'} 0 1_{n''}\rangle = -|2_{n',n} 0 1_{n''}\rangle \tag{49}$$

has an associated angular momentum partition $\{\lambda\} = 201 \equiv \{j_1, j_1, j_1 + 2\}$.

Any basis state element $|\mathfrak{n}\rangle$ in the expansion above can be classified as being one of two (mutually exclusive) types: (i) an expandable $|\mathfrak{n}'\rangle$ or a (ii) non-expandable state (which with some abuse of notation we will denote by $|\mathfrak{n}\rangle$) [24]. By fiat, expandable states can be obtained by an "inward squeezing" of *other basis states appearing in the zero mode under consideration,*



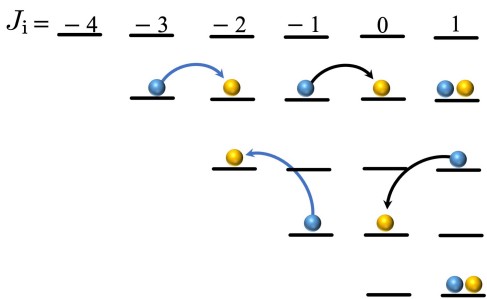

Figure 3: Example of an inward squeezing process for a state consisting of six electrons ($J = -2\hbar$), confined to the four lowest LLs. The electron configuration prior to (after) squeezing is represented by blue (yellow) color.

*Eq. (48), with non-zero coefficient.* If this is not the case, we refer to the basis state as "non-expandable". Here, by "inward squeezing", we refer to an inward pair hoping process in the occupation number basis, i.e.,

$$|\mathfrak{n}'\rangle \propto c^{\dagger}_{n_1,j_1} c^{\dagger}_{n_2,j_2} c_{n_3,j_3} c_{n_4,j_4} |\mathfrak{n}\rangle, \tag{50}$$

where $j_3 < j_1 \leq j_2 < j_4$. For instance, in Fig. 3, the state $|02_{2,3}02_{1,3}2_{0,3}\rangle$ (expandable state in yellow (or light shade)) is obtained from an inward squeezing of $|1_302_{1,3}03_{0,2,3}\rangle$ (the state in blue (darker shade)). We point out that the total angular momentum of $|\mathfrak{n}\rangle$ given by $J = \hbar \sum_{l=j_1}^{j_N} l \lambda_l$ does not change under the inward squeezing process. Thus, the concept of (non-)expandable state is relative to the specific ground state $|\Psi_0\rangle$.

The projection of Eq. (48) onto its non-expandable states is often termed the "root" or "dominant" state. We can schematically write the ground state in Eq. (48) as

$$|\Psi_0\rangle = |\Psi_{\text{root}}\rangle + |\text{Squeezed States}\rangle. \tag{51}$$

Here, $|\Psi_{\text{root}}\rangle$ represents the root state while all expandable Slater determinants are encapsulated in $|\text{Squeezed States}\rangle$. For the LLL, the root state is typically a single Slater determinant [20, 24, 57] obeying a generalized Pauli exclusion principle [20]. For example, in the occupation number basis, such a principle may state that $q$ consecutive states can be occupied by at most $p$ particles. This can gives rise to a QH state at $\nu = p/q$. By contrast, when multiple LLs are present, in account of the degeneracy of the fixed angular momentum orbitals, $0 \leq \lambda_j \leq N_L$, a given root pattern may correspond to various non-expandable Slater determinants. As a result, the root state is a linear superposition of all such non-expandable Slater determinant states,

$$|\Psi_{\text{root}}\rangle = \sum_{\mathfrak{n}_{\text{root}}} C_{\mathfrak{n}_{\text{root}}} |\mathfrak{n}_{\text{root}}\rangle, \tag{52}$$

where Slater determinants $|\mathfrak{n}_{\text{root}}\rangle$ have a common occupation number configuration $\{\lambda\}_{\text{root}}$. This reveals an essential entangled structure associated with the root state, which replaces the generalized Pauli exclusion principles with an EPP as the underlying organizing principle [14, 15]. The EPP encodes the entanglement structure that determines the densest possible root state (associated with the incompressible zero mode state), and various quasihole type and/or edge excitations, which can be thought of as inserting domain walls of various types into the densest root state (see below). Generically, $|\Psi_{\text{root}}\rangle$ contains central information such as density of the QH state, quasiparticle charge and exchange statistics [33, 34], and, in the thin cylinder (Tao-Thouless [58]) limit, it constitutes the exact ground state [58–65]. For these reasons, $|\Psi_{\text{root}}\rangle$ expresses the "DNA" of the QH state [14].

### 3.2.1 Entangled Pauli principle and pseudospin classification

We next study the two-particle ground states for $N_L = 4$ LLs and show that their root states can be understood via its pseudospin structure, i.e., they carry representations of a certain su(2) pseudospin algebra. For pseudospin classification purposes, it is more convenient to work in the pseudofermion basis. (The polynomial part of the corresponding pseudofermion orbital basis states is $\bar{z}_i^{n_i} z_i^{j_i+n_i}$, in contrast to the orthogonal LL orbitals of Section 2.1.) The many-body basis elements are defined as

$$|\mathfrak{n}) = |\{\lambda\}) \equiv \tilde{c}_{n_1,j_1}^* \tilde{c}_{n_2,j_2}^* \cdots \tilde{c}_{n_N,j_N}^* |0\rangle, \tag{53}$$

where $\tilde{c}_{n_i,j_i}^*$ ($\tilde{c}_{n_i,j_i}$) are the pseudofermion creation (annihilation) operators [15] satisfying $\{\tilde{c}_{n_1,j_1}^*, \tilde{c}_{n_2,j_2}\} = \delta_{n_1,n_2}\delta_{j_1,j_2}$. Then, the relevant su(2) pseudospin algebra is defined as $S^\pm = \sum_{j\geq 0} S_j^\pm$ and $S^z = \sum_{j\geq 0} S_j^z$, where

$$\begin{aligned}
S_j^+ &= 3\tilde{c}_{1,j}^* \tilde{c}_{0,j} + 2\tilde{c}_{2,j}^* \tilde{c}_{1,j} + \tilde{c}_{3,j}^* \tilde{c}_{2,j}, \\
S_j^- &= 3\tilde{c}_{2,j}^* \tilde{c}_{3,j} + 2\tilde{c}_{1,j}^* \tilde{c}_{2,j} + \tilde{c}_{0,j}^* \tilde{c}_{1,j}, \\
S_j^z &= \frac{3}{2}\tilde{c}_{3,j}^* \tilde{c}_{3,j} + \frac{1}{2}\tilde{c}_{2,j}^* \tilde{c}_{2,j} - \frac{1}{2}\tilde{c}_{1,j}^* \tilde{c}_{1,j} - \frac{3}{2}\tilde{c}_{0,j}^* \tilde{c}_{0,j}.
\end{aligned} \tag{54}$$

We note that the pseudospin algebra is local in angular momentum space, i.e., for each $j$ the generators satisfy the su(2) algebra. Here, the pseudospin Casimir operator is defined as usual, $\hat{S}^2 = S^+S^- + (S^z)^2 - S^z$, with the eigenvalue of $S(S+1)$. In Section 5, we will expand on the utility of this algebra to locally detect certain degeneracies associated with elementary excitations, emerging from the domain wall structure in the EPP description of the zero mode spectrum.

The pseudospin language is particularly useful to understand the EPP structure. To see this, we study the root states with the following $\{\lambda\}$ patterns: 2, 11, 101, and 1001. We start our discussion with a two-particle root state with multiplicity 2, i.e., two particles with the same angular momentum,

$$|\Psi_0\rangle = \sum_{n,n'} C_{n,n'} |2_{n,n'}). \tag{55}$$

There are 6 coefficients to satisfy the linear constraints defined by Eq. (31). For a single angular momentum $j$, due to the fermionic antisymmetry, only 5 out of 12 constraints, are linearly independent. This leads to 5 linear equations for the coefficients which can be uniquely solved.

Therefore, the unique ground state becomes

$$|\Psi_0\rangle = |2_{3,0}) + 3|2_{1,2}), \tag{56}$$

where both particles occupy orbitals with angular momentum index $j = 0$ (which we suppressed in (56)). One can check that this state is annihilated by both $S^+$ and $S^-$ operators in the pseudospin algebra, and it carries the $S = 0$ representation. We note that the eigenvalues of $S^z$ are determined by the total LL index of the states as $n + n' - 3$. Multiplicity 2 in the root pattern thus forms a singlet and is *generalized* entangled with respect to the u($N = 2$) algebra (single Slater determinants are unentangled with respect to the same algebra) [66, 67].

Consider next the 11 pattern in the root state. The corresponding ground state,

$$|\Psi_0\rangle = \sum_{n,n'} C_{n,n'} |1_n 1_{n'}), \tag{57}$$

has 16 parameters up to a normalization factor to satisfy 12 constraints. We thus get 4 different solutions, which can be expressed as

$$
\begin{aligned}
|\tilde{\Psi}_0^{(1)}\rangle &= |1_0 1_2\rangle + |1_2 1_0\rangle - 2|1_1 1_1\rangle, \\
|\tilde{\Psi}_0^{(2)}\rangle &= |1_0 1_3\rangle - 2|1_1 1_2\rangle + |1_2 1_1\rangle, \\
|\tilde{\Psi}_0^{(3)}\rangle &= |1_3 1_0\rangle - 2|1_2 1_1\rangle + |1_1 1_2\rangle, \\
|\tilde{\Psi}_0^{(4)}\rangle &= |1_1 1_3\rangle + |1_3 1_1\rangle - 2|1_2 1_2\rangle.
\end{aligned}
\tag{58}
$$

In terms of the su(2) pseudospin algebra, $|\tilde{\Psi}_0^{(1)}\rangle$, $|\tilde{\Psi}_0^{(2)}\rangle + |\tilde{\Psi}_0^{(3)}\rangle$ and $|\tilde{\Psi}_0^{(4)}\rangle$ carry a spin triplet representation, while $|\tilde{\Psi}_0^{(2)}\rangle - |\tilde{\Psi}_0^{(3)}\rangle$ is a spin singlet. Hence, 11 is the root pattern realizing pseudospins $S = 0$ and 1.

Now, let us consider the 101 root pattern. The corresponding ground state,

$$
|\Psi_0\rangle = \sum_{n,n'} C_{n,n'} |1_n 0 1_{n'}\rangle + C'_{n,n'} |0 2_{n,n'} 0\rangle,
\tag{59}
$$

has 16 coefficients $C_{n,n'}$ associated with the root state and 6 coefficients $C'_{n,n'}$ associated with inward-squeezed states to satisfy 12 linear constraints and a normalization condition. We thus expect to obtain 10 possible solutions. One of those solutions, however, has already been discussed in Eq. (56), where all the $C_{n,n'}$ are zero (i.e., only the squeezed state contributes). As a result, we obtain 9 independent solutions in the pseudofermion basis, with root states

$$
\begin{aligned}
|\Psi_{\text{root}}^{(1)}\rangle &= |1_1 0 1_0\rangle - |1_0 0 1_1\rangle, \\
|\Psi_{\text{root}}^{(2)}\rangle &= |1_2 0 1_0\rangle - |1_1 0 1_1\rangle, \\
|\Psi_{\text{root}}^{(3)}\rangle &= |1_1 0 1_1\rangle - |1_0 0 1_2\rangle, \\
|\Psi_{\text{root}}^{(4)}\rangle &= |1_2 0 1_1\rangle - |1_1 0 1_2\rangle, \\
|\Psi_{\text{root}}^{(5)}\rangle &= |1_3 0 1_0\rangle - |1_2 0 1_1\rangle, \\
|\Psi_{\text{root}}^{(6)}\rangle &= |1_1 0 1_2\rangle - |1_0 0 1_3\rangle, \\
|\Psi_{\text{root}}^{(7)}\rangle &= |1_3 0 1_1\rangle - |1_2 0 1_2\rangle, \\
|\Psi_{\text{root}}^{(8)}\rangle &= |1_2 0 1_2\rangle - |1_1 0 1_3\rangle, \\
|\Psi_{\text{root}}^{(9)}\rangle &= |1_3 0 1_2\rangle - |1_2 0 1_3\rangle.
\end{aligned}
\tag{60}
$$

In this case, $|\Psi_{\text{root}}^{(.)}\rangle$ is no longer the same as $|\Psi_0\rangle$ as we have excluded inward squeezed terms $C'_{n,n'}$. These root states can be linearly combined to form pseudospins $S = 0$, 1, and 2 representations in the following way,

$$
\begin{aligned}
S = 0,\ S^z = 0 \ &: |\Psi_{\text{root}}^{(6)}\rangle + |\Psi_{\text{root}}^{(5)}\rangle - 2|\Psi_{\text{root}}^{(4)}\rangle, \\
S = 1,\ S^z = 1 \ &: |\Psi_{\text{root}}^{(7)}\rangle - |\Psi_{\text{root}}^{(8)}\rangle, \\
S = 1,\ S^z = 0 \ &: |\Psi_{\text{root}}^{(5)}\rangle - |\Psi_{\text{root}}^{(6)}\rangle, \\
S = 1,\ S^z = -1 &: |\Psi_{\text{root}}^{(2)}\rangle - |\Psi_{\text{root}}^{(3)}\rangle, \\
S = 2,\ S^z = 2 \ &: |\Psi_{\text{root}}^{(9)}\rangle, \\
S = 2,\ S^z = 1 \ &: |\Psi_{\text{root}}^{(8)}\rangle + |\Psi_{\text{root}}^{(7)}\rangle, \\
S = 2,\ S^z = 0 \ &: |\Psi_{\text{root}}^{(6)}\rangle + |\Psi_{\text{root}}^{(5)}\rangle + 4|\Psi_{\text{root}}^{(4)}\rangle, \\
S = 2,\ S^z = -1 &: |\Psi_{\text{root}}^{(2)}\rangle + |\Psi_{\text{root}}^{(3)}\rangle, \\
S = 2,\ S^z = -2 &: |\Psi_{\text{root}}^{(1)}\rangle.
\end{aligned}
\tag{61}
$$

Finally, we consider the pattern 1001 in the root state. The corresponding two-particle ground state,

$$|\Psi_0\rangle = \sum_{n,n'} C_{n,n'}|1_n 001_{n'}\rangle + \tilde{C}_{n,n'}|01_n 1_{n'}0\rangle, \tag{62}$$

has 16 parameters $C_{n,n'}$ associated with the root state and 16 parameters $\tilde{C}_{n,n'}$ associated with inward-squeezed states to satisfy 12 linear constraints and a normalization condition. We thus expect to find 20 possible solutions. Four of those solutions, however, we have already discussed in Eq. (58), where all the $C_{n,n'}$ are zero in Eq. (62). Excluding those 4 solutions, we get 16 independent solutions each of which consists of an unentangled root state (with a single Slater determinant) of the form $|1_n 001_{n'}\rangle$. From these 2-particle considerations, we may now infer/anticipate the following EPP, to be generalized to $N$-particle root states further below:

1. 2 is the highest multiplicity in the allowed ground state root pattern. It can only occur as a pseudospin singlet with $S = 0$.

2. 110 pattern can appear in the root state in pseudospins $S = 0, 1$ representations.

3. 101 pattern can appear in the root state in pseudospins $S = 0, 1, 2$ representations.

4. 1001 pattern can appear in the root state as an unentangled state.

Root states (DNA) consistent with the above EPP rules admit a matrix product state (MPS) representation that highlights its patterns of entanglement. We discuss this next.

### 3.2.2  MPS construction of DNA from the Entangled Pauli principle

We have so far established constraints for the ground state wave function of two particles, formulated as two-particle EPPs for the root states. It remains to show two things. First, that the same constraints apply to the root states of any $N$-particle zero modes. Moreover, that any state that is consistent with these constraints does appear as the root state of some zero mode. Together, this will then allow zero mode counting in terms of all possible "DNAs" of zero modes, namely, the root states consistent with the EPP. It will also allow for the construction of a complete set of zero modes in terms of parton-like wave functions. The latter results will be derived in Sec. 6. In the following several subsections, we focus on elevating the EPP to $N$-particle zero modes and their root states, and on solving for all possible $N$-particle root states consistent with this EPP.

We begin by assuming that rules 1.-3. of the preceding section apply to any $N$-particle root state, and that, in the spirit of rule 4., there are no further constraints. In particular, there are no constraints, at root level, on particles that are separated by more than two orbitals. We will prove this below. For now, let us construct the solutions to these constraints.

It is easy to formulate solutions to the EPP, as formulated above, using a generalized AKLT construction [68]. Note that in our pseudospin description, a pair of particles can be decomposed into $S = 0, 1, 2$ and 3 representations, while each individual particle carries a $S = 3/2$ representation. The latter can be represented as a symmetric rank-3 tensor with indices taking on two distinct values. This, in turn, can be understood as describing three "virtual" spin-1/2 degrees of freedom in a totally symmetric state, as done in the AKLT-construction, giving rise to matrix product or simple tensor network states solving the EPP-constraint. We will represent a symmetric rank-3 tensor $M^I_{I^{(1)}I^{(2)}I^{(3)}}$ by a circle with three legs, as in Fig. 4. The superscript $I$ labels the four possible $S_z$ values such a spin-3/2 state can have (not represented in figure). We now associate each circle with a spin-3/2 state

$$\sum_I M^I_{I^{(1)}I^{(2)}I^{(3)}}|I\rangle. \tag{63}$$

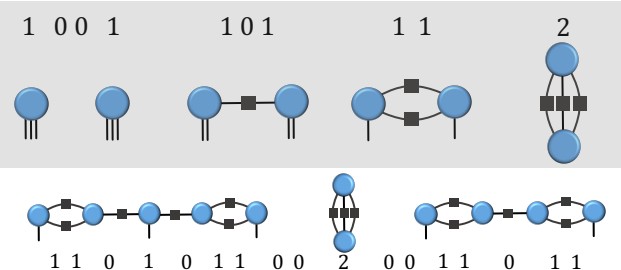

Figure 4: MPS representation of EPPs. Every circle represents a (pseudo-)spin-3/2 degree of freedom formed by three symmetrized spin-1/2, in a generalized AKLT construction. Associated MPSs are then formed by associating a rank-3 symmetric tensor with each circle, whose indices are represented by three emanating lines. Lines between different tensors can form singlet bonds via contraction (with a Levi-Civita tensor $\epsilon_{\alpha\beta}$), represented by a small solid box, as discussed in the text. The various constraints of the EPP can be satisfied by such contractions. The top row represents the various 2-particle root states (labeled by corresponding root patterns) discussed in Sec. 3.2.1. Degeneracies are recovered by considering "dangling bonds" (see main text). The bottom row shows a sequence of "minimum charge" (see Sec. 4) domain walls between 110 and 020 patterns. Such domain walls carry a single spin-1/2 degree of freedom.

We may consider all states obtained by tensoring $N$ copies of these states together. This gives us many "virtual" degrees of freedom encoded in the subscripts, which we may utilize to satisfy the desired constraints, associated with a certain root pattern.

We begin by looking at a situation where a 1 in the pattern is padded by two zeros left and right, i.e., $\ldots 00100\ldots$. In this case, the EPP imposes no constraint on this isolated particle, and the indices $I^{(1)}I^{(2)}I^{(3)}$ in Eq. (63) can be chosen arbitrarily. Note that the choice of $I^{(1)}I^{(2)}I^{(3)}$ reflects the $S^z$ values of the virtual spin-1/2 degrees of freedom, thus, the total $S^z$ of the state. Therefore, only one $I$ in Eq. (63) will contribute for given $I^{(1)}I^{(2)}I^{(3)}$. Note that due to the symmetry of the tensor, this choice of virtual indices only recovers the four-fold (*not* $2 \times 2 \times 2 = 8$-fold) degeneracy associated with a spin-3/2, as it must. In the following, we must always take into account this symmetry when counting degeneracies in terms of free, "dangling spin-1/2 bonds".

Next we consider a 101 configuration in the root pattern. According to the EPP, these cannot be in a spin-3 state, i.e., must be in the subspace formed by the spin-0, 1, and 2 representations. As in the original AKLT-construction, we can achieve this by joining two virtual spin-1/2 degrees of freedom into a singlet. This is done by contracting two indices on the two different tensors representing the two particles with the totally anti-symmetric tensor $\epsilon_{\alpha\beta}$, indicated by a small box in Fig. 4. If the 101 unit is unconstrained on either side (there are at least two 0's on either side), then there will be two pairs of dangling virtual bonds on either side. Owing to symmetry, each such pair is associated with a spin-1 degeneracy, i.e., a 3-fold degeneracy. This recovers the 9-fold degeneracy of the 101-pattern observed in the preceding section.

Similarly, given now a 11-configuration in the root pattern, we would contract two indices on the two different tensors via an $\epsilon_{\alpha\beta}$-tensor, as shown in the figure. This realizes the constraint of the pair being in the spin-0 $\oplus$ spin-1 subspace. Each of the two isolated dangling bonds now represents a two-fold degeneracy, recovering the expected 4-fold degeneracy from the preceding section.

Finally, we can also represent a doubly occupied mode at root level through two tensors with all indices paired into singlets. If now $I$, and $I'$ are the physical (spin-3/2) degrees of freedom of the two fermions, since the latter are now occupying the same "site", i.e., mode, it is important to check that the resulting expression is anti-symmetric under exchange of $I$ and $I'$. This is indeed the case. This leads to the *generalized* entangled [66, 67] two-particle state associated with a "2" discussed above.

Longer units of entangled $\dots 11011 \dots$ are now formed analogously, as shown in the figure. An important special situation are domain walls at root level of the form $\dots 200200\,11011 \dots$ and $\dots 110110\,1\,011011 \dots$, i.e., domain walls representing shifts between the densest possible patterns, 200 and/or 110. These domain walls will play an important role in Sec. 4, in that they represent elementary (charge 1/3, see Sec. 4) excitations. As seen in the figure (bottom row), there is a single dangling bond associated to any such domain wall. The associated elementary excitations thus carry a pseudospin-1/2.

### 3.2.3  The densest $N$-particle ground state

We now formally elevate the EPP to apply to general $N$-particle zero modes and their root states, as already assumed in the preceding subsection. Let us begin by showing that in an $N$-particle root state, when $N_L = 4$, a single angular momentum orbital can have a maximum multiplicity of 2 (here, we follow the method utilized in [14]). To see this, we proceed by assuming multiplicity $p$ for orbitals with angular momentum $\hbar j$. The corresponding root state can then be written as

$$|\Psi_{\text{root}}\rangle = \sum_{n_1, n_2, \dots, n_p} C^j_{n_1 n_2 \dots n_p} c^\dagger_{n_1, j} c^\dagger_{n_2, j} \dots c^\dagger_{n_p, j} |\mathfrak{n}_p\rangle + |\text{rest}\rangle \,,$$

where $|\mathfrak{n}_p\rangle$ is a Slater determinant with $N - p$ particles, and $|\text{rest}\rangle$ includes other Slater determinants in the root state such that $\langle \text{rest}| c^\dagger_{n_1, j} c^\dagger_{n_2, j} \dots c^\dagger_{n_p, j} |\mathfrak{n}_p\rangle = 0$. Generically, there are $\binom{4}{p}$ coefficients $C^j_{n_1 n_2 \dots n_p}$, which determine the pseudospin structure at angular momentum $\hbar j$. Now, contracting Eq. (31) with $|\mathfrak{n}_2\rangle = c^\dagger_{n_3, j} \dots c^\dagger_{n_p, j} |\mathfrak{n}_p\rangle$ and complex conjugating, we obtain

$$\langle \Psi_0 | \mathcal{T}^{\xi\,+}_j | \mathfrak{n}_2 \rangle = \sum_{I=1}^{40} \Lambda^\xi_I \sum_k \eta_{k + \frac{n_2 - n_1}{2}} \left( j + \frac{n_2 + n_1}{2}, m_I \right) \langle \Psi_0 | c^\dagger_{n_1, j-k} c^\dagger_{n_2, j+k} | \mathfrak{n}_2 \rangle = 0 \,. \tag{64}$$

By definition, since the root state consists of the non-expandable states, only $k = 0$ terms can be nonzero in the last line. This gives,

$$\sum_{I=1}^{40} \Lambda^\xi_I \, \eta_{\frac{n_2 - n_1}{2}} \left( j + \frac{n_2 + n_1}{2}, m_I \right) C^j_{n_1 n_2 \dots n_p} = 0 \,, \tag{65}$$

where for each set of particles $(n_3, \dots, n_p)$ we get 5 constraints. It is clear that the number of constraints for $p = 3$ and 4 is larger than the number of coefficients, which leads to $C^j_{n_1 n_2 \dots n_p} = 0$. For $p = 2$, however, we get 6 coefficients and 5 constraints, which uniquely determine the coefficients up to an overall factor. As a result, multiplicity 2 in the root state represents the same singlet state identified above for $N = 2$, irrespective of $j$.

One can follow steps similar to those that led to Eq. (64) to obtain constraints associated to the appearance of Slater determinants of the form $|\mathfrak{n}\rangle = c^\dagger_{n_1, j-k'} c^\dagger_{n_2, j+k'} | \mathfrak{n}_2 \rangle$ in the root state. Here, for all $n_i, k' \geq 0$ and $-k' \leq \tilde{k} \leq k'$, we assume that $c^\dagger_{n_i, j+\tilde{k}} c_{n_i, j+\tilde{k}} | \mathfrak{n}_2 \rangle = 0$. The expansion coefficients $C^{j,k'}_{n_1, n_2}$ of such determinants are found to be subject to the same general constraints already observed for two particles, e.g., the pattern 11 can appear only in the pseudospin 0

and 1 representations (or any linear combination thereof). As a result, in an $N$-particle root state, the local EPP and thus the spin structure of two-particle clusters, 2, 11, 101, 1001, etc., are analogous to the two-particle root states.

It is straightforward to check that 21 and 201 are not allowed in the root pattern, as they would give rise to an over-constrained system of linear constraints. In these cases, each electron in 2 would have to be further entangled with the 1 at the right. For a 111 pattern, the two leftmost 1's would have to be in the subspace of pseudospin 0 or 1 representation, and similarly the two rightmost 1's. An additional constraint would apply to the two outermost 1's. Overall, we will get an over-constrained system. We can thus conclude that the configuration 111 is also not allowed in the ground state root pattern. In contrast, as expected from our 2-particle considerations above, we find no constraint for the pattern 1001. We thus anticipate that this pattern can generally appear at root level, where the sites corresponding to 1's are subject to no other constraints than already mentioned (involving nearest- or next-nearest neighbor occupied sites next to the 1001 pattern). Analogous statements apply for patterns 2002, 2001, 1002, or patterns with more than two 0's separating adjacent sites. We may indeed anticipate that all states satisfying the constraints listed here may appear as root states in *some* zero mode. That this is so, however, will follow from explicit construction in Sec. 6.1.2. Indeed, the results of this section will then lead to a proof that no further zero modes can exist, and that a complete set of zero modes has been found, thus allowing rigorous zero-mode counting in terms of possible root states [14].

As an immediate corollary to the above results, no root state can be "denser" than that corresponding to a pattern with repeated unit cell 200. This pattern corresponds to a filling factor of 2/3, and thus, no zero modes can exist at higher filling factor. We thus get an upper bound $\nu = 2/3$ for our ground state. Note that orbitals with negative angular momentum are somewhat special, as their existence depends on LL index. As as result, we will show in Appendix D that the densest possible root pattern is subject to a left boundary condition of the form 1002002002... . Formally, a root state with bulk pattern ...110110110... is also possible (all constraints can be satisfied, and there is a corresponding zero mode, see Sec. 6.1.2). This pattern also realizes a state with filling factor 2/3 in the thermodynamic limit. However, this pattern and the corresponding zero mode have a slightly higher angular momentum, for a given particle number, than the state associated to the root pattern

$$\{\lambda\}_{\text{root}} = 100200200\ldots2002. \tag{66}$$

We will thus consider the zero mode associated with the latter the "densest" zero mode.

## 4 Braiding Statistics: A case for Fibonacci anyons

A key aspect of our formalism is reduction of parton states to root states. This reduction has some tradition in the literature for single component states [20, 22, 24, 69, 70]. Indeed, one benefit of this reduction is that it grants access to the braiding statistics of the underlying Abelian or non-Abelian QH state. This is so not only by making contact with data predicted by field theory, such as degeneracies, but rather more directly, under the assumptions of the "coherent state Ansatz" [33, 34, 65] the root data self-consistently lead to braiding statistics without invoking or appealing to a field-theoretic formalism. In other words, the assumptions of the coherent-state Ansatz provide an entirely microscopic framework to arrive at braiding. Recently [14], we have demonstrated that this approach does, in principle, generalize to parton states by correctly deriving the braiding statistics [71] for Jain's 221 state. The application to the present case, which we will pursue in full detail in the following, will expose more general features of this formalism.

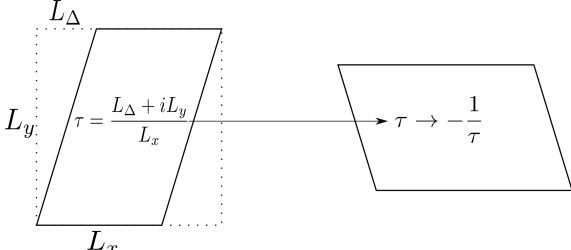

Figure 5: Modular space representation of the torus. S-duality, a simple rotation in the modular space connects two tori of different aspect ratios, namely $\tau \to -\frac{1}{\tau}$.

At its core, the coherent state method assumes the existence on the torus of an *orthonormal* basis of $n_h$-quasihole states labeled by root patterns. This orthogonality is justified by the assumption of adiabatic evolution [21] of quasihole states in the thin torus limit, where these states evolve into (torus versions of) our root states discussed in detail in the preceding sections. Wave functions of localized quasiholes are then naturally described as coherent states in this basis. This Ansatz is then constrained in non-trivial ways by symmetries, notably S-duality on the torus, as well as topological and locality considerations, which strongly constrain braiding.

Before we review this formalism, and extend it to include fermions and/or entanglement at the root level, we observe that our ground state pattern $200200\ldots$ and $110110\ldots$ are formally identical to those of the bosonic Gaffnian state, *if* the underlying entanglement structure is ignored. For these reasons, much of the following calculations can go in parallel with that carried out in Ref. [34] for the Gaffnian, explicitly ignoring the fact that the Gaffnian probably cannot be supplied with a gapped parent Hamiltonian. The present case will differ from the Gaffnian calculation, as both the fermionic nature of the underlying particles *as well as* the entanglement structure of the root state must be taken into account in crucial places (see Section 4.3.4 on mirror symmetry for further details). More generally, within the coherent state formalism, the result we obtain cannot be attributed to any bosonic state, or any *single-component* fermionic state with the given underlying root patterns. Indeed, at the single component level, a given set of root patterns may or may not be consistent [65] with a given statistics of the underlying constituent particles (fermions or bosons). As a direct consequence, entanglement at the root level is necessary to allow for a complete description of certain phases in the FQH regime via root patterns. In the present context, the aforementioned differences with the bosonic Gaffnian case of Ref. [34] are of significant physical importance also since the present case has been linked to a topological phase [72], whereas the Gaffnian is thought to be critical [73]. In particular, the results obtained in the following will be consistent with the effective field theory of Ref. [72].

## 4.1 Symmetry and S-duality on torus

Central to the program described above is the notion of modular S-duality on the torus. We begin by reviewing how this duality is realized by single-particle LL physics and we will later generalize it to our interacting multiple LLs many-body Hamiltonians.

We will start with a torus defined as the infinite complex plane modulo a lattice generated by fundamental periods $\mathbf{L_1} = L_x \hat{x}$, $\mathbf{L_2} = L_\Delta \hat{x} + L_y \hat{y}$: For points on the torus, we may thus choose complex coordinates $z = x + iy$ with the identification $z \equiv z + L_x(r + \tau s)$, where $r$ and $s$ are any integers and the complex aspect ratio $\tau = \tau_1 + i\tau_2 = \frac{L_\Delta}{L_x} + i\frac{L_y}{L_x}$ is called the modular parameter of the torus. Modular transformations on the torus are generated by the realization that the periods $\mathbf{L_1}$ and $\mathbf{L_2}$ are not unique. Specifically, the replacement $\mathbf{L_2} \to \mathbf{L_2} + \mathbf{L_1}$ does

not change the lattice or the underlying torus. The same is true for the replacements $\mathbf{L_1} \to \mathbf{L_2}$, $\mathbf{L_2} \to -\mathbf{L_1}$. These two transformations acting on the lattice are known as modular $T$ and $S$-transformations, respectively, and generate the modular group. These transformations extend trivially to linear transformations of the 2D plane that leave the lattice of periods invariant, and as such, generate non-trivial transformations from the torus to itself. Modular parameters associated to fundamental periods related to each other by a modular transformation describe the same torus. In particular, this leads to the identification of $\tau \to 1 + \tau$ for the modular $T$-transformation, and $\tau \to -\frac{1}{\tau}$ for the modular $S$-transformation. If we insist that the period $\mathbf{L_1}$ is always along the $x$-axis, we may, somewhat loosely speaking, associate to the modular $S$-transformation the active "rotation" depicted in Fig. 5.

Note that, for most values of the parameter $\tau$, the formal replacement $\tau \to -1/\tau$ does not constitute a true symmetry of the Hamiltonian, essentially since the unit cell is not in general invariant under the change shown in Fig. 5, and generates the same lattice only modulo a non-trivial rotation. Rather, therefore, the formal replacement $\tau \to -1/\tau$ is associated to two different descriptions of the same physics. We will now explore the consequences of this duality first at the level of single particle, LL physics on the torus.

At the single particle level, a chief manifestation of S-duality is the existence of two mutually dual choices of basis that we will denote by $\psi_{n,j}$ and $\bar{\psi}_{n,j}$, respectively. Here, $n$ is a LL index, and $j$ will index the eigenvalue of $\psi_{n,j}$ and $\bar{\psi}_{n,j}$ under magnetic translations along the $\mathbf{L_1}$-direction (for $\bar{\psi}_{n,j}$) and the $\mathbf{L_2}$-direction (for $\psi_{n,j}$), respectively. As we will show below, these two basis sets are mutually related by discrete Fourier transform in $j$. This is quite natural, since in the presence of a magnetic field $B$, operators corresponding to $x$ and $y$ coordinates behave as position/momentum conjugate pair upon LL projection. To elaborate this point, we start with the basis $\bar{\psi}_{n,j}$, as with our convention, the magnetic translation in the $\mathbf{L_1}$-direction is simpler ($L_\triangle$-independent)

$$\bar{\psi}_{n,j} = \sum_s \bar{\phi}_{n,j+sN_\phi} \,. \tag{67}$$

The number of flux quanta, $N_\phi = \frac{BL_xL_y}{\phi_0}$, can be identified with the integer $L = N/\nu$, for $N$ electrons on the torus with filling fraction $\nu$, and $\bar{\phi}_{n,j}$ is the $n^{th}$ LL wave function on the cylinder with linear momentum $j$. By construction, $\bar{\phi}_{n,j}$ will satisfy proper magnetic periodic boundary conditions in the $\mathbf{L_1}$-direction, and as we will see, the sum in Eq. (67) will properly periodize it in the $\mathbf{L_2}$-direction. To construct $\bar{\phi}_{n,j}$, we assume a particular gauge $\mathbf{A}_\tau = By(\hat{x} - \frac{\tau_1}{\tau_2}\hat{y})$ which is perpendicular to $\tau_1\hat{x} + \tau_2\hat{y}$ [74]. $\bar{\phi}_{n,j}$ can be readily solved for in terms of Hermite polynomials, as the single particle Hamiltonian $H$ can be expressed in terms of $\hat{a}^\dagger\hat{a}$, where,

$$\hat{a}\bar{\phi}_{0,j} = \frac{1}{\sqrt{2}}\left(\partial_x + i\partial_y + \frac{\tau}{\tau_2}y\right)\bar{\phi}_{0,j} = 0\,,$$

$$\hat{a}^\dagger\bar{\phi}_{n,j} = -\frac{1}{\sqrt{2}}\left(\partial_x - i\partial_y - \frac{\bar{\tau}}{\tau_2}y\right)\bar{\phi}_{n,j} = \bar{\phi}_{n+1,j}\,, \tag{68}$$

with $\bar{\tau} = \tau_1 - i\tau_2$. In the above equations, we have set the magnetic length scale to one, i.e., $\ell = 1$ ($2\pi L = L_xL_y$). Also, we do not require basis states to be normalized, just that their normalization is independent of $j$. We now introduce the magnetic translation operator under the gauge $\mathbf{A}_\tau$,

$$t(\mathbf{l}) = e^{-\mathbf{l}\cdot\nabla - il_y\left(x - \frac{\tau_1}{\tau_2}y\right)}\,, \tag{69}$$

where $\mathbf{l} = l_x\hat{x} + l_y\hat{y}$. Periodic magnetic boundary conditions read

$$t(\mathbf{L_1})\psi = \psi\,,$$
$$t(\mathbf{L_2})\psi = \psi\,. \tag{70}$$

The evaluation of these conditions is somewhat easier in "skewed" coordinates $L_x(x_\Delta + \tau y_\Delta) = x + iy$, where the magnetic translation operator reads

$$t(\mathbf{l}) = e^{-\mathbf{l}\cdot\nabla - il_y L_x x_\Delta}. \tag{71}$$

The orbital $\bar{\phi}_{n,j}$ is fully determined by the requirement that it has $x$-momentum quantum number $j$ and satisfies Eq. (68). The solution, in skewed coordinates reads

$$\bar{\phi}_{n,j} = H_n\left(\sqrt{-i2\pi\tau L}\left(y_\Delta - \frac{j}{L}\right)\right) e^{-i2\pi j x_\Delta + i\pi\tau L\left(y_\Delta - \frac{j}{L}\right)^2}. \tag{72}$$

Having well-defined $x$-momentum $2\pi j/L_x$, $\bar{\phi}_{n,j}$ already satisfies the first of the boundary conditions. The smallest nonzero translations in the $\mathbf{L}_1$-direction and $\mathbf{L}_2$-direction that are consistent with Eqs. (70) are given by $t_1 = t(\mathbf{L}_1/L)$ and $t_2 = t(\mathbf{L}_2/L)$, respectively. One easily checks that $t_2\bar{\phi}_{n,j} = \bar{\phi}_{n,j+1}$, immediately implying $t_2\bar{\psi}_{n,j} = \bar{\psi}_{n,j+1}$, where, at the same time, $\bar{\psi}_{n,j+L} = \bar{\psi}_{n,j}$. Since $t(\mathbf{L}_2) = (t_2)^L$, the second of Eqs. (70) follows for $\bar{\psi}_{n,j}$. We summarize the algebraic properties of $t_1$, $t_2$ and their action on the $\bar{\psi}_{n,j}$ basis as follows

$$[t_1, H] = [t_2, H] = 0, \quad t_1 t_2 = \bar{\omega} t_2 t_1, \quad \bar{\omega} = \omega^{-1} = e^{i\frac{2\pi}{L}},$$
$$t_1\bar{\psi}_{n,j} = \bar{\omega}^j\bar{\psi}_{n,j}, \quad t_2\bar{\psi}_{n,j} = \bar{\psi}_{n,j+1}.$$

In particular $t_1$ and $t_2$ satisfy a Weyl algebra [75], which, as we will see, essentially fixes the change of basis between the $\bar{\psi}_{n,j}$ basis and its dual counterpart, $\psi_{n,j}$.

Before we elaborate further, we wish to construct the $\psi_{n,j}$ basis via continuous deformation of the magnetic lattice. One advantage of the skewed coordinates is that Eq. (72) and the $\bar{\psi}_{n,j}$ derived via Eq. (67) fully retain their meaning if $\mathbf{L}_1$, $\mathbf{L}_2$ are arbitrary and in particular $\mathbf{L}_1$ is *not* necessarily aligned with the $x$-axis. That is, these equations will define a complete set of LL-orbitals for a torus described by any magnetic lattice in the complex plane, for *some* gauge. If we now continuously deform $\mathbf{L}_1$ into the initial $\mathbf{L}_2$ and $\mathbf{L}_2$ into minus the initial $\mathbf{L}_1$, as described in the beginning of this section, the resulting orbitals will again be a valid basis for the original torus. This is, however, a different set of orbitals, as $\tau$ goes to $-1/\tau$ during the transformation, and the skewed coordinates now refer to $(\mathbf{L}_2, -\mathbf{L}_1)$ as opposed to $(\mathbf{L}_1, \mathbf{L}_2)$. Restoring the original skewed coordinates thus amounts to the replacements $x_\Delta \to y_\Delta$, $y_\Delta \to -x_\Delta$ in Eq. (72), on top of the replacement $\tau \to -1/\tau$. The corresponding replacements in Eq. (67) will then define the $\psi_{n,j}$ in some gauge, not equal to the original gauge.

We proceed by finally showing that after gauge fixing, the $\psi_{n,j}$ so defined are related to the $\bar{\psi}_{n,j}$ via discrete Fourier transform. From their characterization in the preceding paragraph, it is straightforward to see that, $t_1$ and $t_2$ act as follows on the $\psi_{n,j}$:

$$t_2\psi_{n,j} = \omega^j\psi_{n,j}, \quad t_1\psi_{n,j} = \psi_{n,j+1}. \tag{73}$$

This actually involved a re-labeling $j \to L - j$, so as to have $t_1$, and not $t_1^\dagger$, increase the $j$-index. With the help of Eqs. (73) one immediately shows that the right-hand side of

$$\psi_{n,j} = \frac{1}{\sqrt{L}}\sum_{j'}\bar{\omega}^{jj'}\bar{\psi}_{n,j'} \tag{74}$$

satisfies Eq. (73). Noting further that the quantum numbers $n$ and $j$ uniquely specify an orbital, by completeness the $\psi_{n,j}$ must be linear combinations of the $\bar{\psi}_{n,j}$ with fixed $n$. Therefore, the first of Eqs. (73) already requires Eq. (74) to be true up to a phase that possibly depends on $n$ and $j$. Requiring also the second of Eqs. (73) renders this phase $j$-independent, and we may

Table 1: Action of symmetry operations and S-duality on single-particle wave functions on a torus without skewness ($L_\Delta = 0$). In the presence of finite skewness, similar relations in particular for S-duality can be defined in skewed coordinates ($x_\Delta, y_\Delta$). For vanishing skewness, the replacement $\tau \to -1/\tau$ and its associated dual descriptions of the torus reduces to an exchange of inverse radii $\kappa = \frac{2\pi}{L_y}, \bar{\kappa} = \frac{2\pi}{L_x}$. The "mirror symmetries" $I_x$ and $I_y$ both involve a time-reversal transformation that we mostly leave understood, and so are anti-unitary operators. Inversion $\bar{I} = I_x I_y = I_y I_x$.

| | | |
|---|---|---|
| $x$-translation $t_1$ | $t_1 \psi_{n,j} = \psi_{n,j+1}$ | $t_1 \bar{\psi}_{n,j} = \bar{\omega}^j \bar{\psi}_{n,j}$ |
| $y$-translation $t_2$ | $t_2 \psi_{n,j} = \omega^j \psi_{n,j}$ | $t_2 \bar{\psi}_{n,j} = \bar{\psi}_{n,j+1}$ |
| Inversion $\bar{I}$ | $\bar{I} \psi_{n,j} = \psi_{n,-j}$ | $\bar{I} \bar{\psi}_{n,j} = \bar{\psi}_{n,-j}$ |
| $x$-mirror $I_x$ | $I_x \psi_{n,j} = \psi_{n,-j}$ | $I_x \bar{\psi}_{n,j} = \bar{\psi}_{n,j}$ |
| $y$-mirror $I_y$ | $I_y \psi_{n,j} = \psi_{n,j}$ | $I_y \bar{\psi}_{n,j} = \bar{\psi}_{n,-j}$ |
| S-duality | $\bar{\psi}_{n,j}(z) = e^{ixy} \psi_{n,j}(-z)_{\kappa \to \bar{\kappa}}$ | |

set it equal to 1 by convention. Indeed, in Appendix E we show in detail that the right-hand side of Eq. (74) evaluates to

$$\psi_{n,j} = e^{-i2\pi L x_\Delta y_\Delta} \sum_s \phi_{n,j+sL},$$

$$\phi_{n,j} = H_n\left(\sqrt{i2\pi \frac{L}{\tau}\left(x_\Delta - \frac{j}{L}\right)}\right) e^{i2\pi j y_\Delta - i\pi \frac{L}{\tau}\left(x_\Delta - \frac{j}{L}\right)^2}. \tag{75}$$

Notice that this is obtained from Eq. (67) via the replacements $\tau \to -1/\tau$, $x_\Delta \to y_\Delta$, $y_\Delta \to -x_\Delta$, $j \to -j$, up to the initial factor $\exp(-i2\pi L x_\Delta y_\Delta)$, which fixes the gauge. Thus, the discrete Fourier transform realizes S-duality at the single-particle level.

In the following, we will usually specialize to tori with $L_\Delta = 0$. In this case, $x_\Delta = \frac{x}{L_x}$ and $y_\Delta = \frac{y}{L_y}$, and the S-duality relation as well as additional symmetries can be simply stated in terms of the original complex $z$ coordinate, as shown in Table 1. We now extend these symmetries/dualities to interacting many-body systems. For magnetic translations, we define many-body operators $T_1 = \prod_{i=1}^N (t_1)_i$ and $T_2 = \prod_{i=1}^N (t_2)_i$, where $(t_{1,2})_i$ acts on the ith particle. While both of these translation operators commutes with $\hat{H}_{\text{int}}$, they inherit non-trivial commutation relations from the single-particle operators via $T_1 T_2 = \bar{\omega}^{\frac{N}{L}} T_2 T_1$. From this, it follows that a ground state with filling fraction $\nu = \frac{p}{q}$ must have ground state degeneracy that is a multiple of $q$. Likewise, one establishes straightforwardly that $\hat{H}_{\text{int}}$ has the inversion symmetry introduced in Table 1. Moreover, for $L_\Delta = 0$ there are *anti*-unitary operators that implement the combination of a mirror symmetry (in $x$ or $y$) with time-reversal symmetry, see Table 1. For simplicity, we will just refer to these symmetries as "mirror symmetries".

Finally, we wish to evaluate the action of S-duality on the interacting Hamiltonian. In most situations, we start with an interaction $V(\mathbf{r}_1 - \mathbf{r}_2)$ defined in the infinite disk that we lift to the torus by periodizing, i.e., defining the following matrix elements on the torus:

$$\hat{V}_{\mathbf{n},\mathbf{j}}^{L_x,L_y} = \frac{1}{2} \int d^2\mathbf{r}_1 d^2\mathbf{r}_2 \, \psi_{n_1,j_1}^*(\mathbf{r}_1) \psi_{n_2,j_2}^*(\mathbf{r}_2) V^{\text{t}}(\mathbf{r}_1 - \mathbf{r}_2) \psi_{n_3,j_3}(\mathbf{r}_2) \psi_{n_4,j_4}(\mathbf{r}_1), \tag{76}$$

with

$$V^{\text{t}}(\mathbf{r}_1 - \mathbf{r}_2) = \sum_{\ell_{1,2}=0,\pm 1,\ldots} V(\mathbf{r}_1 - \mathbf{r}_2 + \ell_1 \mathbf{L}_1 + \ell_2 \mathbf{L}_2),$$

where $\mathbf{j} \equiv (j_1, j_2, j_3, j_4)$, $\mathbf{n} \equiv (n_1, n_2, n_3, n_4)$ are multi-indices. This then defines the following second-quantized two-body interaction on the torus:

$$\hat{H}_{\text{int}} = \sum_{\{\mathbf{n}, \mathbf{j}\}} \hat{V}_{\mathbf{n}, \mathbf{j}}^{L_x, L_y} c_{n_1, j_1}^\dagger c_{n_2, j_2}^\dagger c_{n_3, j_3} c_{n_4, j_4}. \tag{77}$$

Here, the sum is taken over all possible pairs $(n_i, j_i)$ with $i = 1, \cdots, 4$ and $j_1 + j_2 = j_3 + j_4$, the latter being the consequence of translational invariance.

Next, we Fourier transform the fermionic operators,

$$c_{n_i, j_i}^\dagger = \frac{1}{\sqrt{L}} \sum_{l=0}^{L-1} \bar{\omega}^{j_i l} \tilde{c}_{n_i, l}^\dagger, \tag{78}$$

which, according to Eq. (74), is the same as passing to the basis dual to that of the original creation operators via S-duality. This leads to the dual Hamiltonian

$$\hat{H}_{\text{int}}^D = \frac{1}{L^2} \sum_{\mathbf{l}} \sum_{\{\mathbf{n}, \mathbf{j}\}} \left( \bar{\omega}^{j_1 l_1 + j_2 l_2} \omega^{j_3 l_3 + j_4 l_4} \hat{V}_{\mathbf{n}, \mathbf{j}}^{L_x, L_y} \tilde{c}_{n_1, l_1}^\dagger \tilde{c}_{n_2, l_2}^\dagger \tilde{c}_{n_3, l_3} \tilde{c}_{n_4, l_4} \right).$$

For the above, one straightforwardly obtains the matrix elements in the dual basis, which are obtained from the original matrix elements via Fourier transform:

$$\frac{1}{L^2} \sum_{\mathbf{j}} \left( \omega^{j_1 l_1 + j_2 l_2} \bar{\omega}^{j_3 l_3 + j_4 l_4} \hat{V}_{\mathbf{n}, \mathbf{j}}^{L_x, L_y} \right) \equiv \hat{\tilde{V}}_{\mathbf{n}, \mathbf{l}}^{L_x, L_y}. \tag{79}$$

By the single-particle analysis at the beginning of this section, these Fourier transformed, dual matrix elements $\hat{\tilde{V}}_{\mathbf{n}, \mathbf{l}}^{L_x, L_y}$ are obtained from the original ones $\hat{V}_{\mathbf{n}, \mathbf{j}}^{L_x, L_y}$ via the formal replacement, or analytic continuation, effecting $\mathbf{L_1} \rightarrow \mathbf{L_2}$, $\mathbf{L_2} \rightarrow -\mathbf{L_1}$. Again, this is so since this replacement leads to a description of the same magnetic lattice in terms of an alternate basis, effecting precisely the same change of basis as the Fourier transform Eq. (78). (Note that being a density-density interaction, $V(\mathbf{r}_1 - \mathbf{r}_2)$ is gauge invariant). Moreover, if the original interaction $V(\mathbf{r}_1 - \mathbf{r}_2)$ in the infinite plane is rotationally invariant, it is equally legitimate to associate the dual matrix elements $\hat{\tilde{V}}_{\mathbf{n}, \mathbf{l}}^{L_x, L_y}$ to the actively rotated lattice of Fig. 5. It then follows that, assuming now $\mathbf{L_1}$ to be real, $\hat{\tilde{V}}_{\mathbf{n}, \mathbf{l}}^{L_x, L_y}$ is obtained from $\hat{V}_{\mathbf{n}, \mathbf{j}}^{L_x, L_y}$ via a formal replacement/analytic continuation effecting $\mathbf{L_1} \rightarrow |\mathbf{L_2}|$, $\tau \rightarrow -1/\tau$. This is precisely the S-duality that all interactions considered in the work will exhibit. In particular, the TK-Hamiltonian [13] manifestly does so by rotational invariance in the infinite plane.

## 4.2 Quasiholes and domain walls in toroidal geometry, coherent state construction

Braiding statistics in two spatial dimensions are defined as the result of adiabatic transport when two quasiparticles (quasiholes) are exchanging positions. In a topological phase, one expects that the result of such adiabatic transport only depends on the topology of the exchange path, modulo a trivial Aharonov-Bohm (AB) phase. The non-AB part of the adiabatic transport then defines a representation of the braid group. In situations where the quasiparticle (quasiholes) positions and other locally observable quantum numbers do not completely specify the state of the system, one expects this representation to be non-Abelian.

It might seem at first glance hopeless to attempt to describe an intrinsically (2+1)-dimensional phenomenon such as braiding in a language constructed from one-dimensional patterns. For starters, we should establish a faithful representation of quasiholes in root pattern descriptions. According to our earlier results, it must be possible in principle, though.

Indeed, we have established that there exists a one-to-one correspondence between 1D patterns consistent with an EPP, and a complete set of zero modes of the parent Hamiltonian of a 4 LLs-projected Hamiltonian. Therefore, if we limit our discussion to the braiding of quasiholes (as opposed to quasiparticles) injected into the incompressible ground state, any state describing such localized quasiholes is guaranteed to have an expansion in a basis labeled by patterns that correspond to the EPP. Since the states in this basis carry momentum quantum numbers, such an expansion will be non-trivial – or a coherent state. This is so because localized quasiholes break translational invariance in any directions and therefore cannot carry well-defined momentum quantum numbers. As is by now well-known [33,34], the correspondence between the 2D space the braiding takes place in, and the "one-dimensional" coherent states is through a phase-space picture: The coherent states describe wave packets of fractionalized domain walls centered about certain points in the two-dimensional phase space of a one-dimensional quantum system. Indeed, even single-particle physics in a LL can be viewed in similar terms, as a LL has an innate one-dimensional structure. As mentioned before, it is a manifestation of the fact that the $x$- and $y$- position operators satisfy canonical commutation relations after LL projection. While thus the quasihole locations will be encoded in this manner in the coherent state, other quantum numbers are represented by patterns more straightforwardly. Minimum charge domain walls can be created in various ways between the "200200..." and "110110..." patterns of our ground state, respectively. By a Su-Schrieffer-type counting argument, these domain walls will have a charge of 1/3, and so do the associated quasiholes. To further illustrate this point, we consider two wave functions with root patterns of equal length

$$200200200200200200200200200200200200,$$
$$200\ 11011\ 00200200\ 11011\ 00200200\ 11011\ 00200.$$

While the first pattern has 26 particles, the second has 24 particles with six domain walls (indicated by a larger spacing), and same number of single-particle orbitals. Arbitrariness in the exact position of the domain walls will be discussed in Section 4.3.1. Hence these six domain walls have a total charge equal to 2. As all domain walls are related by translation and/or mirror symmetry, each of them must have a 1/3 quasihole charge. The 1/3 quasihole charge can also be derived from domain walls between a 110110... and a 101101... ground state pattern, which we see as follows:

$$200200200200200200200,$$
$$110110110110110110110,$$
$$110\ 1\ 0110\ 1\ 0110110\ 1\ 0110.$$

Both the "200" and "110" patterns in the first two lines have 14 particles and represent the densest (ground state) patterns. The last pattern has three domain walls of "110 1 011" type. By the same counting argument, each carries a 1/3 charge. Any pattern consistent with our EPP can be decomposed in an arrangement of (possibly fused) charge 1/3 domain walls of the types discussed.

At the heart of the formalism is the existence of a basis of quasihole states, within each sector of given charge and/or angular momentum, that is associated to patterns satisfying the EPP. Such patterns were discussed in the preceding paragraph. So far, we have elaborated in detail on the existence of such a basis for the disk geometry. Analogous statements hold for the cylinder and sphere geometries, where essentially the same polynomial wave functions apply. Here, however, we will be working on the torus. On the torus, there are some fundamental differences, chiefly because wave functions are not polynomial, and there is no well-defined notion of "inward-squeezing" or "dominance". We must therefore first elaborate how this basis is realized on the torus.

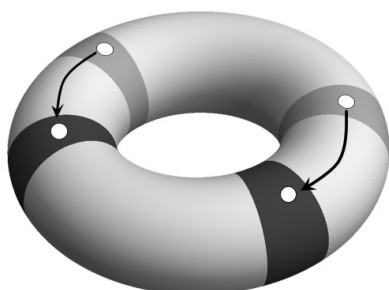

Figure 6: Left- and right-most quasiholes (denoted by white circles) are moved from gray bands to black bands. Their movement can be faithfully represented using $x$-direction movement of corresponding domain walls. This picture breaks down as soon as quasiholes cross along the $x$-direction. Such operation is thus topologically prohibited in the coherent state method. Different ordering of quasiholes belong to different configurations $\sigma$s. All configurations for two and three quasiholes are displayed in Figs. 7 and 8, respectively.

The construction of such a basis rests on the assumption that the quasihole states on the torus can be adiabatically evolved into the "thin torus limit" that we next briefly discuss. Henceforth, we will work with purely imaginary $\tau$. A thin torus is one where $\tau \to 0$ *or* $\tau \to \infty$ (for fixed number of flux quanta). The assumption of adiabatic continuity has been extensively investigated [69]. It is generally assumed to hold for all frustration-free positive semi-definite Hamiltonians of the kind discussed here. While torus wave functions and their Hamiltonians are harder to study directly, the thin torus limit is locally indistinguishable from a thin cylinder limit. In the latter case, we know that not only does the EPP apply to all zero modes, but that zero modes must also reduce to the very root states satisfying the EPP [14, 28]. The same must then hold true on the torus. In the following, we will use the round ket notation $|a, \alpha)$ for states satisfying the EPP on the torus. The notation will be expounded on below and in Tables 3, 4. For now, we will take $\{a, \alpha\}$ to be an abstract label to encode the pattern. Then $|a, \alpha)$ is a complete set of zero modes in the thin torus limit $\tau \to 0$, where patterns refer to the $\psi_{n,j}$ single particle basis constructed above. Likewise, we can construct a complete set of zero modes in the dual thin torus limit $-i\tau \to \infty$, denoted by $\overline{|a, \alpha)}$. The only difference between the $\overline{|a, \alpha)}$ and the $|a, \alpha)$ is that the former refer to patterns ("root states") constructed from the single particle basis $\bar{\psi}_{n,j}$, as opposed to $\psi_{n,j}$. Both kinds of round kets represent thin torus zero modes, but in opposite thin torus limits.

The zero-mode bases we work with at arbitrary given (imaginary) $\tau$ will be defined via adiabatic evolution from the thin torus bases sets as a function of modular parameter. We will denote the basis obtained in this way via adiabatic evolution from the $|a, \alpha)$ by $|a, \alpha\rangle = U(\tau, 0)|a, \alpha)$, where $U(\tau, \tau')$ is a unitary operator associated with the adiabatic evolution from modular parameter $\tau'$ to modular parameter $\tau$. The $|a, \alpha\rangle$ thus depend on $\tau$, but we will mostly leave this understood. Likewise, we define $\overline{|a, \alpha\rangle} = U(\tau, i\infty)\overline{|a, \alpha)}$. An important property of the $|a, \alpha\rangle$, and likewise the $\overline{|a, \alpha\rangle}$, is that by virtue of the unitarity of adiabatic evolution, and the fact that the $|a, \alpha)$ are manifestly orthogonal, they, too, are orthogonal. Moreover, we will take the $|a, \alpha)$, $\overline{|a, \alpha)}$, and thus the $|a, \alpha\rangle$, $\overline{|a, \alpha\rangle}$, to be normalized. An important difference between the $|a, \alpha)$, $\overline{|a, \alpha)}$ and their adiabatically evolved counterparts $|a, \alpha\rangle$, $\overline{|a, \alpha\rangle}$ is the fact that $|a, \alpha)$, $\overline{|a, \alpha)}$ represent torus zero modes at very different modular parameters, whereas the $|a, \alpha\rangle$, $\overline{|a, \alpha\rangle}$ are each complete sets of torus zero modes at *the same* modular parameter $\tau$. As a consequence, the $|a, \alpha\rangle$, $\overline{|a, \alpha\rangle}$ are related to one another by a unitary transformation. This is the manifestation of S-duality within the zero-mode space on the torus.

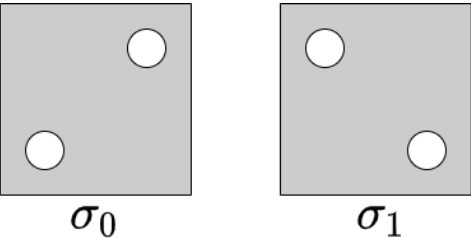

Figure 7: Configurations $\sigma_0, \sigma_1$ for two quasiholes. Braid matrix can be expressed as overlap of coherent states in configurations $\sigma_0$ and $\sigma_1$. Application of global path symmetry $F_x$, $F_y$ and mirror symmetry $I_x$, $I_y$ takes one configuration to another.

In the $\psi_{n,j}$ ($\bar{\psi}_{n,j}$) basis, domain walls are localized along $x$ ($y$) direction. The associated charge depletion is likewise localized in the $x$ ($y$) direction, but delocalized along $y$ ($x$) direction. The latter follows from the fact that these states are adiabatically evolved from states that are eigenstates of translation in $y$ ($x$), and the adiabatic evolution preserves these quantum number. For a description of braiding statistics, we desire to have a description of quasiholes that are localized in both $x$ and $y$ coordinates. Following Ref. [33], we can construct a coherent state Ansatz, $|\psi_\alpha(h)\rangle$,

$$|\psi_\alpha(h)\rangle = \sum_a \prod_{i=1}^{n_h} \phi_i^\alpha(h_i, a_i) |a, \alpha\rangle \,, \tag{80a}$$

$$\phi_i^\alpha(h_i, a_i) = e^{i\beta(\kappa h_{i_y} + \delta_i^\alpha)a_i - \gamma(h_{i_x} - \kappa a_i)^2} \,, \tag{80b}$$

where $h = \{h_1, h_2, ..., h_{n_h}\}$ is the set of complex coordinates for the locations of $n_h$ quasiholes such that the position of the $i^{th}$ quasihole is given by $h_i = h_{i_x} + ih_{i_y}$. $a = (a_1, a_2, ..., a_{n_h})$ is an ordered set of numbers, to be further specified below, s.t, $1 \le a_1 < a_2 < ... < a_{n_h} \le L$ determining the orbital locations of the domain walls inserted into a topological sector identified by the label $\alpha$, such that $a, \alpha$ together completely determine the thin torus state $|a, \alpha\rangle$ adiabatically evolved from. For given $n$, $\alpha$ thus identifies a certain sequence of patterns 200 and 110 that are separated by the domain walls. Tables 3 and 4 show our conventions for $n_h = 2$ and $n_h = 3$. As we will elaborate in later sections, there is a two-fold degeneracy associated to any minimum charge domain wall, corresponding to a local pseudo-spin 1/2 degree of freedom. We will ignore this degree of freedom here and assume that all quasiholes are in the same pseudo-spin state, rendering them locally indistinguishable. The Gaussian form factor $\phi_i^\alpha(h_i, a_i)$ then localizes the $i$th quasihole in $x$ near $h_{i_x}$, since $\kappa a_i$ is the $x$-location of the $i$th domain wall in position space, with $\kappa = 2\pi/L_y$. $\gamma$ determines the shape of the quasihole, and is assumed to be chosen such that it is circular. The precise value of $\gamma$ will not be needed. The $y$-location of the quasihole is determined by the $x$-momentum phase factor of the Gaussian. One can determine the parameter $\beta$ to be 1/3 from S-duality and symmetries on the torus with the methods of Ref. [33]. Thus, the Gaussian form factor $\phi_i^\alpha(h_i, a_i)$ when viewed as a function of $h_i$ exactly has the form of the LLL wave function of a charge 1/3-particle. The parameters $\delta_i^\alpha$ represent an offset between a quasihole's $x$-momentum and $y$-position. These offsets are not arbitrary, since they can, in principle, depend on the type of domain wall. Moreover, are not free to shift the origin of our coordinate system arbitrarily, as below we will argue that by duality, an analogous expression holds in the dual basis. Such complete formal analogy does not, however, survive arbitrary changes in origin. One may indeed see that a certain origin is naturally made special in our description of a LL once the gauge, and equally importantly, certain arbitrary phases in the magnetic translation operators

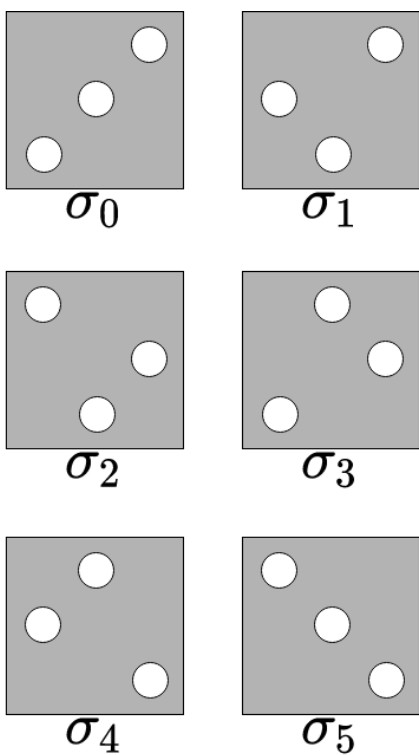

Figure 8: Six configurations for three quasiholes. Braid matrix can be expressed as overlap of coherent states in configurations $\sigma_{2n}$ and $\sigma_{2n+1}$, for $n = 0, 1, 2$. Application of $F_x$, $F_y$ and $I_x$, $I_y$ changes configurations non-trivially (see Table 2).

have been fixed.[1] By symmetry arguments [33], the shifts $\delta_i^\alpha$ may be restricted to zero and $\pi$. Moreover, they must be the same for the same types of domain wall (between the same ground state patterns on either side), and also for domain wall types related by inversion. The last important observation about the coherent state Ansatz Eq. (80) is that since the domain wall positions $a_i$ are ordered, $1 \leq a_1 < a_2 < ... < a_{n_h} \leq L$, the Ansatz is justifiable only as long as the $x$-positions $h_{i_x}$ of the quasiholes are ordered similarly, and are moreover well-separated (compared to a magnetic length) in $x$. In is only then that the $\phi_i^\alpha$ describe well-separated, non-overlapping wave packets (see Fig. 6). We now employ our dual basis construct. We will argue that local operators like the density $\hat{\rho}(x, y)$ are represented by the same matrix when passing to the dual basis if $x$ and $y$ are rotated with the replacement $\kappa \to \bar{\kappa}$ where $\bar{\kappa} = 2\pi/L_x$. From this it follows that an analogous coherent state Ansatz exists in the dual basis,

$$\overline{|\psi_\alpha(h)\rangle} = \sum_a \prod_{i=1}^{n_h} \bar{\phi}_i^\alpha(h_i, a_i)\overline{|a, \alpha\rangle}, \tag{81a}$$

$$\bar{\phi}_i^\alpha(h_i, a_i) = e^{-i\beta(\bar{\kappa}h_{i_x} - \delta_i^\alpha)a_i - \gamma(h_{i_y} - \bar{\kappa}a_i)^2}, \tag{81b}$$

where $\bar{\phi}_i^\alpha(h_i, a_i) = \phi_i^\alpha(-ih_i, a_i)_{\kappa \to \bar{\kappa}}$ according to the S-duality relation in Table 1, and quasiholes must now be separated well in the $y$ direction. Thus, different conditions dictate the validity of Eqs. (80) and (81), respectively. In the following, we will pay special attention to configurations where *both* expressions are valid. Understanding that both the $h_{i_x}$ and the $h_{i_y}$ must be pairwise distinct (by [much] more than a magnetic length), we will usually follow

---

[1]The basis orbitals can be imagined as "rings" traversing the torus in $x$ or $y$ directions for the two bases, respectively. Take the origin to be the point where the zero-momentum orbitals in the two respective bases intersect.

Table 2: Transmutation of $\sigma$s as a result of global path $F_{x,y}$ and mirror $I_{x,y}$ operations for three quasiholes. $F_x$ moves rightmost quasihole from $(h_x, h_y) \rightarrow (h_x - L_x, h_y)$. $F_y$ moves topmost quasihole from $(h_x, h_y) \rightarrow (h_x, h_y - L_y)$. Mirror $I_x$ moves each quasihole at $(h_x, h_y)$ to $(L_x - h_x, h_y)$, while $I_y$ moves each quasihole at $(h_x, h_y)$ to $(h_x, L_y - h_y)$. For two quasiholes, $F_x(\sigma) = F_y(\sigma) = I_x(\sigma) = I_y(\sigma) = \sigma' \neq \sigma$.

| $\sigma$ | $F_x(\sigma)$ | $F_y(\sigma)$ | $I_x(\sigma)$ | $I_y(\sigma)$ |
|---|---|---|---|---|
| $\sigma_0$ | $\sigma_2$ | $\sigma_4$ | $\sigma_5$ | $\sigma_5$ |
| $\sigma_1$ | $\sigma_5$ | $\sigma_5$ | $\sigma_4$ | $\sigma_2$ |
| $\sigma_2$ | $\sigma_4$ | $\sigma_0$ | $\sigma_3$ | $\sigma_1$ |
| $\sigma_3$ | $\sigma_1$ | $\sigma_1$ | $\sigma_2$ | $\sigma_4$ |
| $\sigma_4$ | $\sigma_0$ | $\sigma_2$ | $\sigma_1$ | $\sigma_3$ |
| $\sigma_5$ | $\sigma_3$ | $\sigma_3$ | $\sigma_0$ | $\sigma_0$ |

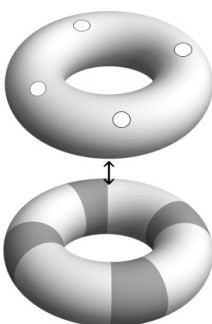

Figure 9: In each configuration $\sigma$, corresponding to quasihole ordering, quasiholes can be faithfully represented as domain walls. In the 4 LLs projected ground state of the $\hat{H}_{\text{int}}$ of Eq. (30), there are 3 types (up to translation symmetry) of domain walls, namely 200 110, 110 200 and 110 1 011. A sequence of these domain walls, $\alpha$, is topologically distinct from another sequence $\alpha'$. Each $\alpha$ defines a topological sector within each configuration $\sigma$. For 2 and 3 quasiholes configurations, we have 9 and 12 topological sectors, respectively. (see Tables 3 and 4)

the convention $h_{1_x} < h_{2_x} < \dots$ . This is assumed in Eq. (80), because the domain walls $a_i$ are generally ordered in the same manner. For the same reason, however, the right-hand side of Eq. (81a) assumes that $h_{1_y} < h_{2_y} < \dots$ . Strictly speaking, in general these two ways of ordering quasiholes need not be the same, but differ by an (implicit) permutation $\sigma$. Thus, as long as we stick with the former convention, Eq. (81a) must thus be replaced with

$$\overline{|\psi_\alpha(h)\rangle} = \sum_a \prod_{i=1}^{n_h} \bar{\phi}_i^\alpha(h_{\sigma(i)}, a_i) \overline{|a, \alpha\rangle} . \tag{82}$$

In essence, $\sigma$ labels different configurations of quasiholes, as shown in Figs. 7 and 8 for two and three quasiholes, respectively. It is not possible to traverse from one configuration to another without violating one of the two conditions that render both Eqs. (80) and (81) valid, *or* by crossing the boundaries of the unit cell of our lattice defining the torus in the extended zone-scheme infinite plane. The latter process, however, also changes the topological sector (see below).

Consider now a quasihole configuration labeled by $\sigma$, such that both the "original" coherent state expression (80) and its dual (81) are valid. Then, as the quasihole locations $h$ identify a $d$-dimensional subspace in the $n_h$-quasihole zero-mode space, where $d$ is the ($n_h$-dependent) number of topological sectors (see Fig. 9). By assumption, both the original and the dual

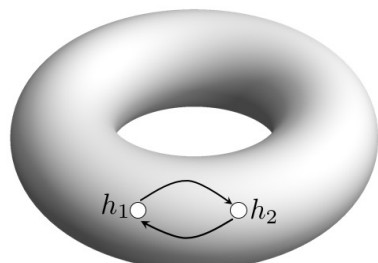

Figure 10: Braiding, an exchange operation of two consecutive quasiholes, can be thought of in terms of the overlap matrix between $|\Psi^\sigma\rangle$ and $|\Psi^{\sigma'}\rangle$. $|\Psi^\sigma\rangle$ is a column matrix of $|\psi_\alpha^\sigma\rangle$s for all topological sectors $\alpha$s. Configurations $\sigma$ and $\sigma'$ are identical except $h_{1_x} < h_{2_x}$ in $\sigma$ gets changed to $h_{1_x} > h_{2_x}$.

coherent states constitute orthonormal bases for this subspace. We thus have the general relation

$$|\psi_\alpha(h)\rangle = \sum_{\alpha'} u_{\alpha\alpha'}^\sigma(h)\overline{|\psi_{\alpha'}(h)\rangle}, \tag{83}$$

where $u^\sigma(h)$ is a unitary matrix that depends smoothly on hole positions within each component of configurations space characterized by a well-defined $\sigma$. Indeed, the $h$-dependence of these matrix-valued transitions functions can be determined [33] from adiabatic transport (holonomy) as follows: $u^\sigma(h) = u(h)\xi^\sigma$, with $u(h) = \exp[i\beta \sum_{i=1}^{n_h} h_{i_x} h_{i_y}]$. From now on, we will refer to $\xi^\sigma$ as the transition matrix.

## 4.3 Braiding in coherent state language

The goal is now to work out the result of adiabatic transport along an exchange path such as shown in Fig. 10, using the coherent state description. To understand how non-trivial braiding comes about in this language, we first observe that we introduced not one but two well distinct methods of defining what a topological sector is. One is to say that a quasihole state lie in the topological sector $\alpha$ if its $L_y \to 0$ limit under adiabatic evolution consists of a sequence of patterns identified with $\alpha$. The alternative definition is analogous, except utilizing the opposite limit $L_x \to 0$. The relation between these two notions of a topological sector is non-trivial. Hence, we generally expect the transitions functions to have off-diagonal matrix elements. The assumption that justifies the term "topological sector" is the following: We assume that no local operator has matrix elements between $|a, \alpha\rangle$ and $|a', \alpha'\rangle$ as long as the same is already true in the associated thin torus limit, i.e., for the states $|a, \alpha\rangle$ and $|a', \alpha'\rangle$. For the latter to be true, it is clearly sufficient that $\alpha \neq \alpha'$ *and* all domain walls are well-separated. In particular, under the same conditions that the coherent state expression Eq. (80) holds, i.e., that all quasiholes are well separated in $x$, no local operator has matrix elements between $|\psi_\alpha(h)\rangle$ and $|\psi_{\alpha'}(h')\rangle$. This includes the local density operator. It is for this reason that localized quasiholes can be formed from $|a, \alpha\rangle$ with a fixed $\alpha$. Moreover, adiabatic transport where local quasiholes are dragged along some path, where the dragging can be thought of as being facilitated via slowly changing local potentials, does not lead to changes in the topological sector, *as long as* the coherent state $|\psi_\alpha(h)\rangle$ remains well defined. Analogous statements hold for the dual coherent states, $\overline{|\psi_\alpha(h)\rangle}$. Luckily, the above does not rule out transitions into a different topological sector along the exchange path shown in Fig. 10. This is so, because along such a path, the conditions for the validity of either coherent state certainly becomes violated somewhere. In particular, the condition that both quasiholes are well separated in $x$ becomes

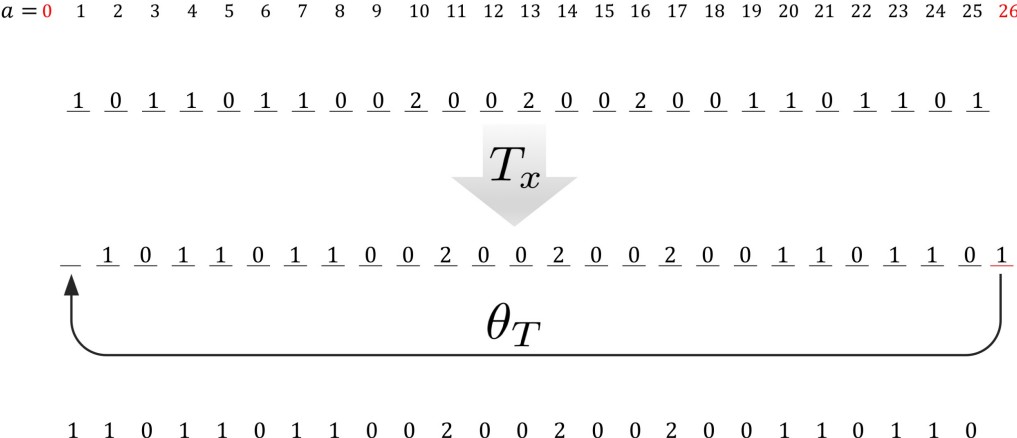

Figure 11: Translation moves each particle by one unit in linear momentum space, $|a\rangle \to |a+1\rangle$. If the last orbital $L = 25$ is occupied, it gets moved to $L+1$, outside our modular coordinates. We have to move this last particle from $L+1 \to 1$ by commuting through other particles. For a system with two quasiholes the total number of particles are always even, thus we get an extra sign $\theta_T(\alpha)$ in fermionic systems. Moreover, resulting pattern belongs to a different topological sector $\alpha' = T(\alpha)$. $\theta_T(\alpha)$, $T(\alpha)$s are tabulated for all $\alpha$s in the table for two and three quasiholes, respectively.

violated. Related to that, the configuration label $\sigma$, where well-defined, assumes multiple values during the path. On the other hand, everywhere along the exchange path, at least one of the two coherent state expressions holds. We may thus evaluate the result of the adiabatic exchange in Fig. 10 using the following strategy. Close to the initial/final positions, we use the coherent state (80) to work out the result of adiabatic transport. At appropriate locations of well-defined $\sigma$, we change between the original and the dual coherent state descriptions by means of Eq. (83), and so in between those locations, we describe the adiabatic transport using the dual coherent state description. It follows from the above discussion that locally, i.e., in the coherent state description appropriate to the respective segment of the path, no transitions between topological sectors happen, and all the information about the braid matrix $\chi$ that describes the result of the adiabatic exchange is contained in the transition matrices $\xi^\sigma$, evaluated in the two configurations where a change of basis is performed. Details are given in Ref. [33]. The result is, with a trivial Aharonov-Bohm phase dropped,

$$|\Psi^{\text{final}}\rangle = \xi^{\sigma_0} (\xi^{\sigma_1})^\dagger |\Psi^{\text{initial}}\rangle \Rightarrow \chi = \xi^{\sigma_0} (\xi^{\sigma_1})^\dagger . \tag{84}$$

Here, $|\Psi\rangle$ denotes a column vector, respectively for the initial/final state, with the coefficients of the different $\psi_\alpha(h)$ with different $\alpha$ as columns. One can proceed analogously for more than two quasiholes, where one neighboring pair is exchanges with the other quasiholes staying fixed.

Via Eq. (84), that task of calculating braiding has been reduced to the evaluation of the transition matrices $\xi^\sigma$. In the following, we will show that these matrices are sufficiently constrained by various symmetry and locality considerations. To this end, we find it educational to discuss some concrete examples of how translation and mirror symmetry as well as certain processes involving "global paths" act on root patterns (see Figs. 11-13).

Table 3: Topological table for 2 quasiholes (even fermion number): Sectors $\alpha$ with $h_{i_x} \sim \kappa a_i$. Position of domain walls are chosen while maintaining the inversion symmetry. Moreover, $a_i + 3$ can be identified with $a_i$ due to the torus degeneracy of the wave function. Domain wall 200200 110110 is related to 011011 002002 by inversion symmetry. Domain wall 110 1 0110 maps to itself under inversion. For bosonic case, $\theta_T = \theta_F = \theta_I = 1$ for all $\alpha$. In [34], the topological sector $\alpha$ is denoted by $(c, \tilde{\alpha})$.

| $(c,\tilde{\alpha})$ | $\alpha$ | patterns | $a_1$ | $a_2$ | $\sum_j jn_j$ | $T(\alpha)$ | $\theta_T(\alpha)$ | $F(\alpha)$ | $\theta_F(\alpha)$ | $I(\alpha)$ | $\theta_I(\alpha)$ |
|---|---|---|---|---|---|---|---|---|---|---|---|
| $(-1,1)$ | 1 | 0200200 11011011 0020020020 | $8-s$ | $15+s$ | 209 | 2 | 1 | 5 | 1 | 3 | $-1$ |
| $(0,1)$ | 2 | 00200200 11011011 002002002 | $9-s$ | $16+s$ | 225 | 3 | 1 | 6 | 1 | 2 | $-1$ |
| $(1,1)$ | 3 | 200200200 11011011 00200200 | $10-s$ | $17+s$ | 191 | 1 | 1 | 4 | $-1$ | 1 | $-1$ |
| $(-1,2)$ | 4 | 1011011 00200200200 1101101 | $7+s$ | $19-s$ | 208 | 5 | $-1$ | 2 | $-1$ | 6 | 1 |
| $(0,2)$ | 5 | 11011011 00200200200 110110 | $8+s$ | $20-s$ | 199 | 6 | 1 | 3 | 1 | 5 | $-1$ |
| $(1,2)$ | 6 | 011011011 00200200200 11011 | $9+s$ | $21-s$ | 215 | 4 | $-1$ | 1 | 1 | 4 | 1 |
| $(-1,3)$ | 7 | 10110110 1 0110110 1 01101101 | 9 | 17 | 208 | 8 | $-1$ | 8 | 1 | 9 | 1 |
| $(0,3)$ | 8 | 110110110 1 0110110 1 0110110 | 10 | 18 | 199 | 9 | 1 | 9 | $-1$ | 8 | $-1$ |
| $(1,3)$ | 9 | 0110110110 1 0110110 1 011011 | 11 | 19 | 215 | 7 | $-1$ | 7 | 1 | 7 | 1 |

Table 4: Topological table for 3 quasiholes (odd fermion number): Sectors $\alpha$ with $h_{i_x} \sim \kappa a_i$. Position of domain walls are chosen while maintaining the inversion symmetry. Moreover, $a_i + 3$ can be identified with $a_i$ due to the torus degeneracy of the wave function. Domain wall 200200 110110 is related to 011011 002002 by inversion symmetry. Domain wall 110 1 0110 maps to itself under inversion. For the bosonic case, $\theta_T = \theta_F = \theta_I = 1$ for all $\alpha$. In [34], the topological sector $\alpha$ is denoted by $(c, \tilde{\alpha})$.

| $(c,\tilde{\alpha})$ | $\alpha$ | patterns | $a_1$ | $a_2$ | $a_3$ | $\sum_j jn_j$ | $T(\alpha)$ | $\theta_T(\alpha)$ | $F(\alpha)$ | $\theta_F(\alpha)$ | $I(\alpha)$ | $\theta_I(\alpha)$ |
|---|---|---|---|---|---|---|---|---|---|---|---|---|
| $(-1,1)$ | 1 | 0200 110110110 1 011011011 0020020 | $5-s$ | 14 | $23+s$ | 296 | 2 | 1 | 8 | 1 | 3 | 1 |
| $(0,1)$ | 2 | 00200 110110110 1 011011011 002002 | $6-s$ | 15 | $24+s$ | 315 | 3 | 1 | 9 | 1 | 2 | 1 |
| $(1,1)$ | 3 | 200200 110110110 1 011011011 00200 | $7-s$ | 16 | $25+s$ | 275 | 1 | 1 | 7 | 1 | 1 | 1 |
| $(-1,2)$ | 4 | 10110 1 011011 00200200200 1101101 | 6 | $12+s$ | $24-s$ | 296 | 5 | 1 | 2 | 1 | 9 | 1 |
| $(0,2)$ | 5 | 110110 1 011011 00200200200 110110 | 7 | $13+s$ | $25-s$ | 285 | 6 | 1 | 3 | 1 | 8 | 1 |
| $(1,2)$ | 6 | 0110110 1 011011 00200200200 11011 | 8 | $14+s$ | $26-s$ | 304 | 4 | 1 | 1 | 1 | 7 | 1 |
| $(-1,3)$ | 7 | 1011 00200200200 110110 1 01101101 | $4+s$ | $16-s$ | 22 | 296 | 8 | 1 | 5 | 1 | 6 | 1 |
| $(0,3)$ | 8 | 11011 00200200200 110110 1 0110110 | $5+s$ | $17-s$ | 23 | 285 | 9 | 1 | 6 | 1 | 5 | 1 |
| $(1,3)$ | 9 | 011011 00200200200 110110 1 011011 | $6+s$ | $18-s$ | 24 | 304 | 7 | 1 | 4 | 1 | 4 | 1 |
| $(-1,4)$ | 10 | 10110 1 0110110 1 0110110 1 01101101 | 6 | 14 | 22 | 296 | 11 | 1 | 11 | 1 | 12 | 1 |
| $(0,4)$ | 11 | 110110 1 0110110 1 0110110 1 0110110 | 7 | 15 | 23 | 285 | 12 | 1 | 12 | 1 | 11 | 1 |
| $(1,4)$ | 12 | 0110110 1 0110110 1 0110110 1 011011 | 8 | 16 | 24 | 304 | 10 | 1 | 10 | 1 | 10 | 1 |

### 4.3.1 Inversion symmetry

In Table 1, we defined inversion symmetry with respect to a rather arbitrary center. In combination with magnetic translations, we can, of course, fix any point on the torus to be the center of our inversion symmetry (note that on the torus, inversion always fixes two points). This can be used to constrain or fix a number of parameters we so far introduced explicitly or implicitly. Consider the domain wall positions $a_i$, which we think of as the "orbital positions" of our domain walls. Our different types of domain walls have different symmetry character when it comes to inversion. If we regard a symmetric domain wall of the type . . . 110110 1 011011 . . . , symmetry dictates that its domain wall position should coincide with the orbital index of the central 1 in this pattern. This can be made rigorous as follows. One consequence of its symmetry is that it is possible to have a single domain wall of this type on the torus, with no other domain walls present, if we appropriately choose the number of flux quanta. We may then write a single quasihole coherent state of the form Eq. (80) for the topological sector associated to the pattern . . . 110110 1 011011 . . . . Now choose an inversion center that preserves this topological sector. We may then demand that applying this inversion to the coherent state produces, up to a phase, the coherent state in the same topological sector with the hole sent

from position $h$ to $2h_I - h$, with $h_I$ the inversion center. From the completeness of our coherent states, the operation of inversion as described will produce a zero mode in the same topological sector with a quasihole localized at $2h_I - h$. One may show that to be consistent with these observations, the domain wall position $a_1$ entering the ($n_h = 1$) coherent state Eq. (80) must indeed coincide with the orbital index of the unpaired 1 in the pattern associated to $|a_1, \alpha\rangle$. For details, we again refer the reader to Ref. [33].

The situation is rather different for the other type of domain wall. Consider the pattern 200200 11011 00200200. It is easy to see that due to lack of inversion symmetry, on the torus such domain wall must always come in pairs. There is then no argument that the "correct" way to choose the domain wall positions $a_1$ and $a_2$ in the pattern is for them to be chosen integers. Here, "correct" again means that the coherent state expression (80) succeeds at localizing the two quasiholes precisely at the complex coordinates given by the parameters $h_1$, $h_2$. Clearly, the first domain wall is localized somewhere between the terminal zero of the first 200-string and the leading 1 of the 110-string. However, there is no immediately obvious way to make this more precise. However, we must plug in some real numbers $a_1$ into the coherent state Ansatz Eq. (80). Hence we must make a choice. The only way to avoid bias is to introduce a parameter $s$ and say that the domain wall position is of the form integer$-s$ for the first domain wall. Inversion symmetry arguments of the flavor discussed for single, inversion-symmetric domain wall then still imply that the second domain-wall position must be of the form integer$+s$, as shown in the following schematic: 200200 ⌷ 11011 ⌷ 00200200,

where ⌷ represents domain walls. This shows, in particular, that the parameter $s$ cannot be absorbed into a coordinate shift (which would in any case also adversely affect conventions for the 110 1 011-type domain walls). We will subsequently constrain $s$, and our solution for the braid matrix will crucially depend on it.

Similar arguments can be made about the parameters $\delta_i^\alpha$. In the case of a 200200 11011 00200200-type topological sector, one can similarly show that $\delta_1^\alpha = -\delta_2^\alpha \bmod 2\pi$. Anticipating that mirror symmetries, which we will discuss in more detail below, lead to similar constraints, we note that analogous requirements with respect to $I_x$-symmetry imply $\delta_1^\alpha = \delta_2^\alpha \bmod 2\pi$. Together, these two constraints fix all $\delta_i^\alpha$-parameters to be 0 or $\pi$ modulo $2\pi$. Since a shift of any $\delta_i^\alpha$ by $2\pi$ only changes coherent state by overall phases, we can simply take $\delta_i^\alpha = 0, \pi$ for all $i$ and $\alpha$. Furthermore, all $\delta_i^\alpha$ referring to domain walls related by mirror/inversion symmetry must be the same, and similarly, using translational symmetry, all $\delta_i^\alpha$ referring to domain walls related by translational symmetry must be the same. It follows that there are only two independent $\delta_i^\alpha$, one for 200200 110110-type domain walls (and their mirror images), and one for 110110 1 011011-type domain walls. Lastly, for reasons related to the fact that the 110110 1 011011-type domain walls can exist as single domain walls on the torus, combining the above symmetries with duality turns out to fix the $\delta_i^\alpha$ for such domain walls completely (mod $2\pi$). In the following subsection, we will show the associated $\delta_i^\alpha$ to be 0.

### 4.3.2 Translation symmetry

The Hamiltonian commutes with magnetic translations. Thus, under adiabatic evolution in the thin torus limit, the action of magnetic translations on the basis $|a, \alpha\rangle$ is the same as that on the "bare", thin torus states $|a, \alpha\rangle$. This is straightforward to work out. Analogous statements

hold for the dual basis, giving:

$$T_x |a, \alpha\rangle = \theta_T(\alpha) |a + 1, T(\alpha)\rangle \,, \tag{85a}$$

$$T_y |a, \alpha\rangle = e^{-i\kappa\bar{\kappa} \sum_j j n_j} |a, \alpha\rangle$$

$$\Rightarrow T_y |a, \alpha\rangle = f(\alpha) e^{i\beta\kappa\bar{\kappa} \sum_i a_i} |a, \alpha\rangle \,, \tag{85b}$$

$$T_y \overline{|a, \alpha\rangle} = \theta_T(\alpha) \overline{|a + 1, T(\alpha)\rangle} \,, \tag{85c}$$

$$T_x \overline{|a, \alpha\rangle} = e^{i\kappa\bar{\kappa} \sum_j j n_j} \overline{|a, \alpha\rangle}$$

$$\Rightarrow T_x \overline{|a, \alpha\rangle} = f^*(\alpha) e^{-i\beta\kappa\bar{\kappa} \sum_i a_i} \overline{|a, \alpha\rangle} \,. \tag{85d}$$

Here, $T(\alpha)$ and $\theta_T(\alpha)$ are tabulated in Tables 3-4 for two and three quasiholes, respectively. In Eqs. (85b),(85d), we have used $\beta = 1/3$ to recast the phase factor appearing in terms of domain-wall positions. The factor

$$e^{-i\kappa\bar{\kappa} \left( \sum_j j n_j + \beta \sum_i a_i \right)} = f(\alpha) \,, \tag{86}$$

with $\kappa\bar{\kappa} = 2\pi/L$, does not depend on the positions $a_i$ themselves, but only on the topological sector (in Tables 3 and 4 it only depends on $c$). Keep in mind that within a fixed topological sector, each domain wall has a stride of 3, i.e., the value $a_i$ is fixed modulo 3. Related, the choice $\beta = 1/3$ is crucial in rendering Eq. (86) dependent on the topological sector only.

The above equations crucially differ from those in Ref. [34] by a fermionic sign $\theta_T(\alpha)$. The application of $T_x$ on $|a, \alpha\rangle$ moves every particle to one site to the right ($a \to a + 1$). In this operation if a particle on the rightmost orbital crosses the boundary used in our fermionic ordering conventions (which may be taken to be increasing in orbital index), it reappears as the leftmost particle, thanks to periodic boundary conditions, thus $\theta_T(\alpha) = (-1)^{(\#\text{of fermionic permutations})}$. Here (#of fermionic permutations) is the number of permutations needed to reorder fermion operators in the root state according to increasing orbital index.

As an example, consider Table 3 for two quasiholes. In the $\alpha = 1$ case, we get no particle moving from left to right, hence, $\theta_T(1) = 1$. For $\alpha = 2$, two particles simultaneously cross the boundary, hence, $\theta_T(2) = 1$. For $\alpha = 4$, one particle crosses the boundary, hence, $\theta_T(4) = -1$. In the three quasihole case, due to odd total particle number, the number of permutations is always even, hence, $\theta_T(\alpha) = 1$ for all $\alpha$s and there is no difference with the case studied in Ref. [34].

Using Eqs. (85) in Eqs. (80), (81) we obtain the effect of the translation operators on coherent states:

$$T_x |\psi_\alpha(h)\rangle = \theta_T(\alpha) e^{-i\beta \sum_i (\kappa h_{i_y} + \delta_i^\alpha)} |\psi_{T(\alpha)}(h + \kappa)\rangle \,, \tag{87a}$$

$$T_y |\psi_\alpha(h)\rangle = f(\alpha) |\psi_\alpha(h + i\bar{\kappa})\rangle \,, \tag{87b}$$

$$T_y \overline{|\psi_\alpha(h)\rangle} = \theta_T(\alpha) e^{i\beta \sum_i (\bar{\kappa} h_{i_x} - \delta_i^\alpha)} \overline{|\psi_{T(\alpha)}(h + i\bar{\kappa})\rangle} \,, \tag{87c}$$

$$T_x \overline{|\psi_\alpha(h)\rangle} = f^*(\alpha) \overline{|\psi_\alpha(h + \kappa)\rangle} \,. \tag{87d}$$

One sees that this has the expected effect, namely, up to phase, to shift the position variables by $\kappa$ and $\bar{\kappa}$, respectively, for $T_x$ and $T_y$. While the first and third of these equations follow straightforwardly from the definition of the coherent states, the remaining two crucially depend on Eq. (86) and thus the fact that $\beta = 1/3$. $\beta$ can thus be uniquely determined from the requirement that $T_x$ and $T_y$ act consistently on the two mutually dual versions of the coherent states [33].

Now we are in a position to apply these operations directly in the S-duality relation (83), in order to obtain a first crucial set of constraints on the transition matrices:

$$|\psi_\alpha(h)\rangle = u(h) \sum_{\alpha'} \xi^\sigma_{\alpha\alpha'} \overline{|\psi_{\alpha'}(h)\rangle}$$

$$\Rightarrow T_x |\psi_\alpha(h)\rangle = \theta_T(\alpha) e^{-i\beta \sum_i (\kappa h_{i_y} + \delta^\alpha_i)} |\psi_{T(\alpha)}(h+\kappa)\rangle$$

$$= u(h+\kappa) e^{-i\beta \sum_i \kappa h_{i_y}} \theta_T(\alpha) e^{-i\beta \sum_i \delta^\alpha_i} \sum_{\alpha'} \xi^\sigma_{T(\alpha)\alpha'} \overline{|\psi_{\alpha'}(h+\kappa)\rangle}$$

$$= u(h) \sum_{\alpha'} \xi^\sigma_{\alpha\alpha'} f^*(\alpha') \overline{|\psi_{\alpha'}(h+\kappa)\rangle}$$

$$\Rightarrow \theta_T(\alpha) e^{-i\beta \sum_i \delta^\alpha_i} \xi^\sigma_{T(\alpha)\alpha'} = \xi^\sigma_{\alpha\alpha'} f^*(\alpha'), \tag{88a}$$

where we have used Eq. (83) (first line) on the right hand side of Eq. (87a) (second line), and compared this to the effect of applying $T_x$ to the first line and evaluating the right hand side via Eq. (87d). The last line is obtained by comparing coefficients in the two lines preceding it.

It is advantageous to cast this as a matrix equation. Let us define the following matrices using the Kronecker delta $\delta_{\alpha\alpha'}$,

$$e^{-i\beta\delta_T}{}_{\alpha\alpha'} = \delta_{\alpha\alpha'} e^{-i\beta \sum_i \delta^\alpha_i}, \tag{88b}$$

$$B_{T\alpha\alpha'} = \theta_T(\alpha)\delta_{T(\alpha)\alpha'}, \quad f_{\alpha\alpha'} = \delta_{\alpha\alpha'} f(\alpha). \tag{88c}$$

With this we can condense Eq. (88a) into matrix form,

$$e^{-i\beta\delta_T} B_T \xi^\sigma f = \xi^\sigma. \tag{89}$$

Similarly, while the above was obtained from the action of $T_x$ along with S-duality, we can get analogous equations by using $T_y$ instead:

$$f(\alpha)\xi^\sigma_{\alpha\alpha''} = \xi^\sigma_{\alpha\alpha'} e^{-i\beta \sum_i \delta^\alpha_i} \theta_T(\alpha')\delta_{T(\alpha')\alpha''} \tag{90a}$$

$$\Rightarrow f\xi^\sigma = \xi^\sigma e^{-i\beta\delta_T} B_T. \tag{90b}$$

The effect of these equations is the following. Following [33], one may group the topological sectors in Tables 3-4 into "supersectors" of three sectors each, related by local lattice translations ($T_x$ or $T_y$ in the mutually dual cases, respectively). Using the above equations utilizing translational symmetry, all matrix elements of $\xi^\sigma$ between any two given supersectors are linearly related. Thus, the number of independent variables in the $\xi^\sigma$-matrix, for $n_h = 2$ quasiholes, is reduced from 27 to 9. These equations also further constrain the $\delta^\alpha_i$-parameters. To see this, let us focus again on $n_h = 2$ quasiholes for the moment. Iterating Eq. (89) three times, we obtain

$$\xi^\sigma = (e^{-i\beta\delta_T} B_T)^3 \xi^\sigma (f)^3. \tag{91}$$

One easily finds that $f^3$ is the identity, while $(e^{-i\beta\delta_T} B_T)^3 = (e^{-i\beta\delta_T})^3$. This equation thus reduces to

$$\xi^\sigma = e^{-3i\beta\delta_T} \xi^\sigma. \tag{92}$$

$e^{-3i\beta\delta_T}$ is a diagonal matrix, and any of its entries that is not equal to 1 would, by the above equation, force an entire row of $\xi^\sigma$ to vanish. This cannot happen, since $\xi^\sigma$ is unitary. Hence, $e^{-3i\beta\delta_T}$ is the identity. Since $3\beta = 1$, this gives

$$\sum_i \delta^\alpha_i = 0 \mod 2\pi. \tag{93}$$

For two domain walls, this is just the familiar fact that $\delta^\alpha_i$ for two mutually inverted domain walls are either both 0 or both $\pi$, also already concluded from mirror and inversion symmetry. However, when the above argument is repeated for a single or for three domain walls, one finds that $\delta^\alpha_i = 0$ for the 110110 1 011011-type domain walls.

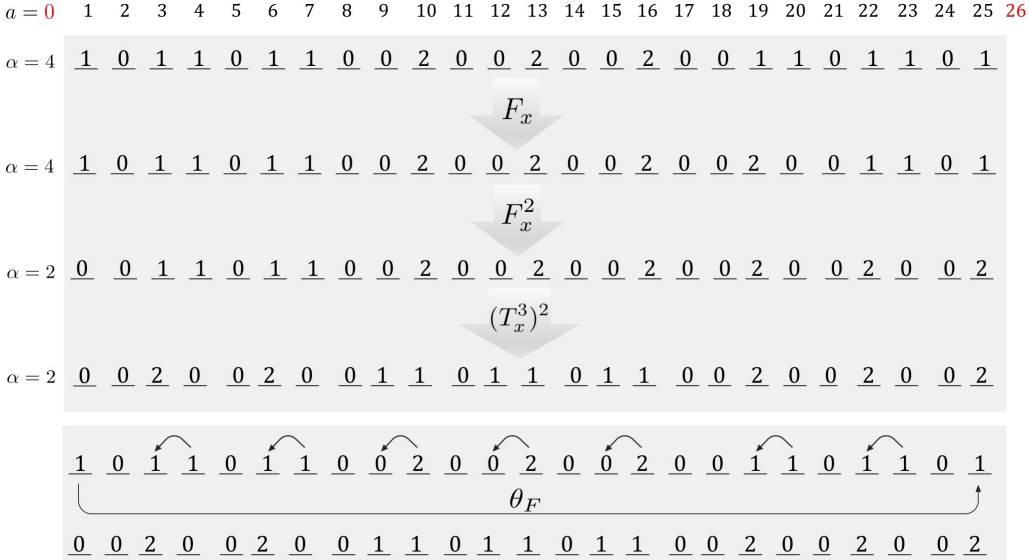

Figure 12: Global path operation $F_x$ moves rightmost domain walls to the further right by three lattice sites at a time. Application of $F_x$ twice more moves the domain wall across the right boundary (in the modular space), which in turn, changes topological sector $\alpha(=4)$ to $F(\alpha)$. To determine $F(\alpha)(=2)$, we next apply $T_x^3$ twice. Notice that $T_x^3$ does not change the topological sector. In this example, 11 00200-type domain wall finally gets moved to the left. The resulting state can also be constructed by moving one particle for each arrow. In this case, the leftmost particle has to cross the boundary to go to rightmost position. This process introduces a fermionic sign factor $\theta_F(\alpha)$ as well as a change in topological sector, $\alpha \to F(\alpha)$.

### 4.3.3 Global path ($F$) operation

So far, we have one separate transition matrix $\xi^\sigma$ for every configuration $\sigma$. To eliminate enough parameters in order to evaluate the braid matrix Eq. (84), it will be necessary to establish relations between the transition matrices for different configurations. As we hinted when introducing the different configurations $\sigma$, it is not possible to move quasiholes from one configuration $\sigma$ to another configuration $\sigma'$ while keeping both our expressions for $|\psi_\alpha(h)\rangle$ and $\overline{|\psi_\alpha(h)\rangle}$ well defined, unless we move across the boundary of our unit cell defining the torus in the magnetic "extended zone scheme". Now we wish to make use of this feature. To do so, we define two operations that cross boundaries in the extended zone scheme. Let $F_x$ be the operation of analytically continuing[2] the expressions for $|\psi_\alpha(h)\rangle$ and $\overline{|\psi_\alpha(h)\rangle}$ in $h_{n_h}$, i.e., the "righmost" particle coordinate, into the region $h_{n_{hx}} > L_x$. In the case of $|\psi_\alpha(h)\rangle$, one thereby transitions into a different topological sector $F(\alpha)$, but not so for $\overline{|\psi_\alpha(h)\rangle}$. The analytically continued state $|\psi_\alpha(h)\rangle$ now describes a zero mode in the topological sector $F(\alpha)$ with quasiholes at positions $h'$, which is the same as $h$, except the rightmost position $h_n$ has become the leftmost position at $h_n - L_x > 0$. This also changes the configuration $\sigma$ associated with $h$ to $\sigma' = F_x(\sigma)$ associated to $h'$. By the usual completeness argument, the analytically continued state $|\psi_\alpha(h)\rangle$ must be equal up to a phase to the coherent state $|\psi_{F(\alpha)}(h')\rangle$. Indeed, it may be checked directly that this is so. The analytically continued state, however, by means of the duality relations Eq. (83), is still related to the dual coherent states via the transition matrix elements $\xi^\sigma_{\alpha\alpha'}$, whereas the state $|\psi_{F(\alpha)}(h')\rangle$ is, by means of the same equation, connected to the dual states via the matrix elements $\xi^{F_x(\sigma)}_{F(\alpha)\alpha'}$. In this way, we establish a matrix relation

---

[2]Alternatively, we may define this via adiabatic transport.

between $\xi^\sigma$ and $\xi^{F_x(\sigma)}$. In a completely analogous manner, we define an operation $F_y$ that takes the topmost particle and moves it over the boundary in the extended zone scheme, affecting now the sector of the dual coherent state $\overline{|\psi_\alpha(h)\rangle}$ via the function $F(\alpha)$, and sending the configuration $\sigma$ to $F_y(\sigma)$. In this way, we obtain a matrix relation between $\xi^\sigma$ and $\xi^{F_y(\sigma)}$.

For two quasiholes, there are only two configurations $\sigma$ (Fig. 7), and the above will suffice to express the transition matrices for one in terms of that of the other, a crucial step in evaluating the braid matrix from Eq. (84). For three particles, all configurations $\sigma$ (Fig. 8) can still be related to each other via the actions of the $F_x$ and $F_y$ moves *and* the mirror symmetries to be discussed in the following subsection. These actions are summarized in Table 2.

Beyond relating transition matrices for different $\sigma$, the $F_{x/y}$-moves also lead to additional constraints on any one $\xi^\sigma$. To see this, we will focus in $\sigma_0 = \text{id}$, i.e., where particles are ascending in $h_x$ as well as $h_y$. For $\sigma = \text{id}$, one easily verifies the relation,

$$F_x(F_y(\text{id})) = \text{id}\,, \tag{94}$$

which constraints $\xi^{\text{id}}$, and, via the aforementioned relations, all other $\xi^\sigma$.

At the level of the basis $|a, \alpha\rangle$, the $F_x$ operation can be given an interpretation that manifests continuity under periodic boundary conditions. Let $a = (a_1, \ldots, a_{n_h})$ be a set of domain wall positions in the topological sector $\alpha$. Suppose $a_{n_h} + 3 \le L$. In this case, $F_x|a, \alpha\rangle = |a', \alpha\rangle$ is simply the state in the same topological sector where the "last" domain wall has hopped to the right by 3 units, such that $a' = (a_1, \ldots, a_{n_h} + 3)$. On the other hand, if $a_{n_h} + 3 > L$, we move into a different topological sector. In this case, $a' = (a_{n_h} + 3 - L, a_1, \ldots a_{n_h} - 1)$, and

$$F_x|a, \alpha\rangle = \theta_F(\alpha)|a', F(\alpha)\rangle\,. \tag{95}$$

Here, $\theta_F(\alpha)$ is a fermionic factor as a result of restoring the order of fermionic orbitals after hopping.[3] It is given in Tables 3 and 4. This operation now allows us to continuously evolve the coherent state $|\psi_\alpha(h)\rangle$ as the $x$-coordinate of the rightmost particle changes from $h_{n_{hx}} < L_x$ to $h_{n_{hx}} > L_x$: In the coherent state expression Eq. (80), we usually assume that all quasihole coordinates are well away from the boundaries of our extended zone scheme unit cell. In this manner, we need not worry about the limits of the sums over domains wall positions, due to exponential localization. This changes when, $h_{n_{hx}} \approx L_x$. In this case, many basis states $|a', \alpha\rangle$ may enter the coherent state with appreciable weight such that $a' = (a_1, \ldots, a_{n_h} + 3q)$, where $q$, and $a_{n_h} \le L$, but $a_{n_h} + 3q > L$. Then, since $a = (a_1, \ldots, a_{n_h})$ is still a proper set of domain wall positions in the topological sector $\alpha$, we can make the identification

$$|a', \alpha\rangle \equiv F_x^q|a, \alpha\rangle\,. \tag{96}$$

Here, $F_x^q$ is the operation that applies the action defined for $F_x$ to the *last* domain wall in $a$ $q$-times (even *after* this domain wall possibly becomes the "first" during this process, thus strictly, $F_x^q \ne (F_x)^q$). With this, the coherent state $|\psi_\alpha(h)\rangle$ evolves smoothly as $h_{n_{hx}} \approx L_x$ and even as $h_{n_{hx}} \gg L_x$ (where $\gg$ signifies multiple magnetic lengths). In the latter case, the identification (96) straightforwardly leads to the following identification of coherent states:

$$|\psi_\alpha(h)\rangle \equiv_{F_x} \theta_F(\alpha)e^{i\beta L(\kappa h_{i_y} + \delta_i^\alpha)}|\psi_{F(\alpha)}(h')\rangle\,, \tag{97a}$$

where $h'$ is obtained from $h = (h_1, \ldots, h_{n_h})$ via $h' = (h_{n_h} - L_x, h_1, \ldots, h_{n_h-1})$ (we tacitly assume $h_{n_h} - L_x < h_1$).

On the other hand, $|\psi_\alpha(h)\rangle$ already evolves smoothly as $h_{i_y}$ is increased beyond $L_y$. It is straightforward to verify that

$$|\psi_\alpha(h)\rangle = e^{i2\pi\beta a_i(\alpha)}|\psi_\alpha(h'')\rangle\,, \tag{97b}$$

---

[3]Again, this can be worked out from the "basis" $|a, \alpha\rangle$, as a result of translational invariance and the fact that translations commute with adiabatic evolution.

where $i$ is the index of the quasihole with largest $h_{i_y}$, $h'' = (h_1, \ldots, h_i - iL_y, \ldots, h_{n_h})$, and we have defined $a_i(\alpha)$ the domain wall position of the $i$th particle in the topological sector $\alpha$, which is well-defined modulo 3. The above two equations then also hold, mutatis mutandis, for the dual coherent states $\overline{|\psi_\alpha(h)\rangle}$:

$$\overline{|\psi_\alpha(h)\rangle} \equiv_{F_y} \theta_F(\alpha) e^{-i\beta L(\kappa h_{i_y'} - \delta_{i'}^\alpha)} \overline{|\psi_{F(\alpha)}(h'')\rangle}, \tag{97c}$$

$$\overline{|\psi_\alpha(h)\rangle} = e^{-i2\pi\beta a_i(\alpha)} \overline{|\psi_\alpha(h')\rangle}. \tag{97d}$$

Using these relations now in the usual manner inside the S-duality relation (83), we get, for $F_x$:

$$|\psi_\alpha(h)\rangle = u(h) \sum_{\alpha'} \xi_{\alpha\alpha'}^\sigma \overline{|\psi_{\alpha'}(h)\rangle}$$
$$\Rightarrow \theta_F(\alpha) e^{i\beta L\delta_i^\alpha} \delta_{F(\alpha)\alpha''} \xi_{\alpha''\alpha'}^{F_x(\sigma)} = \xi_{\alpha\alpha'}^\sigma e^{-i2\pi\beta a_i(\alpha')},$$

from which we read off the corresponding matrix equation:

$$e^{i\beta\delta_i} B_F \xi^{F_x(\sigma)} e^{i2\pi\beta a_i} = \xi^\sigma. \tag{98a}$$

Similarly, using $F_y$ and the S-duality, (83), gives

$$e^{i2\pi\beta a_i} \xi^{F_y(\sigma)} (e^{i\beta\delta_{F_{i'}}} B_F)^{-1} = \xi^\sigma, \tag{98b}$$

where, in the above, we have defined the following matrices:

$$e^{i\beta\delta_{F_i}}{}_{\alpha\alpha'} = \delta_{\alpha\alpha'} e^{i\beta L\delta_i^\alpha}, \quad B_{F\alpha\alpha'} = \theta_F(\alpha)\delta_{F(\alpha)\alpha'},$$
$$e^{i2\pi\beta a_i}{}_{\alpha\alpha'} = \delta_{\alpha\alpha'} e^{i2\pi\beta a_i(\alpha')}.$$

Finally, using Eqs. (98) together with the observation (94) gives the following matrix equation constraining $\xi^{\text{id}}$:

$$e^{i2\pi\beta a_i} e^{i\beta\delta_i} B_F \xi^{\text{id}} e^{i2\pi\beta a_i} (e^{i\beta\delta_{F_{i'}}} B_F)^{-1} = \xi^{\text{id}}. \tag{99}$$

### 4.3.4 Mirror symmetry

We summarized the representation of mirror symmetry in Table 1. Since mirror symmetries commute with the Hamiltonian and adiabatic evolutions, their actors on the basis $|a, \alpha\rangle$ can be worked out from the bare thin torus limits $|a, \alpha\rangle$, and similarly for $\overline{|a, \alpha\rangle}$. We stress again, that $I_x$ and $I_y$ are both anti-linear, in fact, anti-unitary operators, where the basis $|a, \alpha\rangle$ is invariant under $I_y$, and the basis $\overline{|a, \alpha\rangle}$ is invariant under $I_x$, but the two other pairings are non-trivial, and change, in general, the topological sector. Detailed actions are as follows:

$$I_x |a, \alpha\rangle = \theta_I(\alpha) |L - a, I(\alpha)\rangle, \tag{100a}$$

$$I_y |a, \alpha\rangle = |a, \alpha\rangle, \tag{100b}$$

$$I_y \overline{|a, \alpha\rangle} = \theta_I(\alpha) \overline{|L - a, I(\alpha)\rangle}, \tag{100c}$$

$$I_x \overline{|a, \alpha\rangle} = \overline{|a, \alpha\rangle}. \tag{100d}$$

In the above, we used the shorthand notation $L - a$ for $(L - a_{n_h}, L - a_{n_h-1}, \ldots, L - a_1)$. The map $I(\alpha)$ is given in Tables 3 and 4. A sign $\theta_I(\alpha)$ is generated in the non-trivial ones of these operations, also shown in these tables, similar to the sign generated in translation and

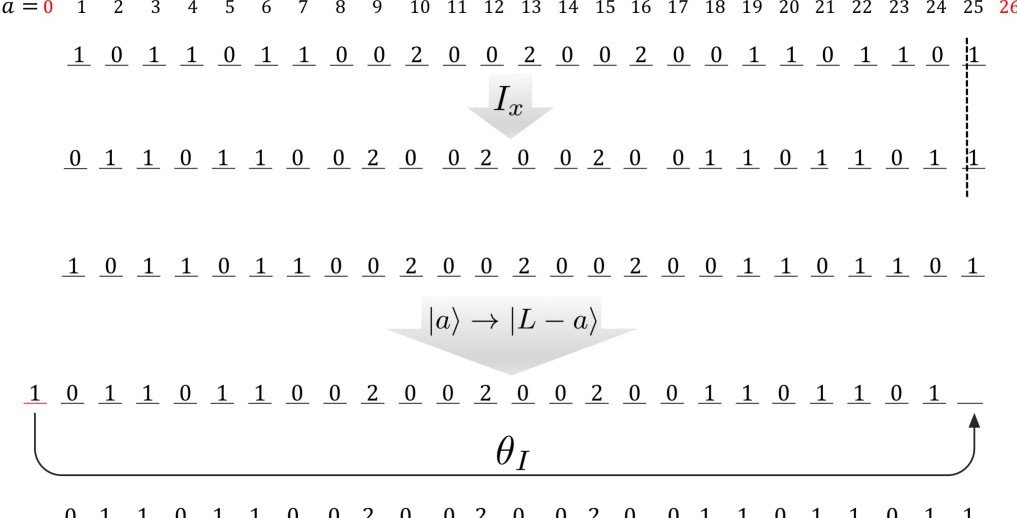

Figure 13: A mirror symmetry operation is defined here with respect to a mirror positioned at $a = L = 25$. A mirror symmetry operation can be viewed as composition of two operations. First operation, that needs the explicit form of the root state, moves $|a\rangle \rightarrow |L - a\rangle$ and adds an overall minus sign due to the entanglement structure of the root state, see Section 3.2.2 and Eq. (212). In the starting configuration, $a$ takes value from 1 to $L$. Any particle at angular momentum $L$ will go to 0 after this operation. In order to keep the original modular coordinates, we have to commute the leftmost particle from 0 to $L$. This process generate extra fermionic phase $\theta_I(\alpha) = -\theta_T(\alpha)$ and a change in topological sector, $\alpha \rightarrow I(\alpha)$.

*F*-moves. However, unlike for translations and *F*-moves, this sign does not only depend on the fermionic nature of their underlying particles, *but also* receives non-trivial contributions from the reversal of bonds connecting these particles. To understand the origin of these extra minus signs, one must consider the entanglement structure of the root states, constructed later in Eq. (212). Each bond in the MPS representation of the root state carries a Levi-Civita tensor. Under mirror symmetry they will acquire an extra $-1$ sign. At this point, our "topological tables" crucially differ from both single-component bosons (the Gaffnian case of Ref. [34]) as well as single component fermions (there would be no consistent solution for reasons related to results of Ref. [65]).

Using the above, it is straightforward to work out the action of mirror (anti-unitary) operations on coherent states:

$$I_x |\psi_\alpha(h)\rangle = \theta_I(\alpha) e^{-i\beta L \sum_i (\kappa h_{i_y} + \delta_i^\alpha)} |\psi_{I(\alpha)}(L_x - h^*)\rangle , \tag{101a}$$

$$I_y |\psi_\alpha(h)\rangle = e^{-i2\beta \sum_i (\pi + \delta_i^\alpha) a_i(\alpha)} |\psi_\alpha(h^* + iL_y)\rangle , \tag{101b}$$

$$I_y \overline{|\psi_\alpha(h)\rangle} = \theta_I(\alpha) e^{i\beta L \sum_i (\bar\kappa h_{i_x} - \delta_i^\alpha)} \overline{|\psi_{I(\alpha)}(h^* + iL_y)\rangle} , \tag{101c}$$

$$I_x \overline{|\psi_\alpha(h)\rangle} = e^{i2\beta \sum_i (\pi - \delta_i^\alpha) a_i(\alpha)} \overline{|\psi_\alpha(L_x - h^*)\rangle} , \tag{101d}$$

where, again, expressions like $L_x - h^*$ are shorthand notations for the implicated action on all quasihole coordinates. This also changes the configuration from $\sigma$ to $I_x(\sigma)$ or $I_y(\sigma)$, as shown in Table 2. Just as with the other symmetries, we will use the above in the S-duality

relation Eq. (83):

$$|\psi_\alpha(h)\rangle = u(h) \sum_{\alpha'} \xi^\sigma_{\alpha\alpha'} \overline{|\psi_{\alpha'}(h)\rangle}$$

$$\Rightarrow I_{x(y)}|\psi_\alpha(h)\rangle = u^*(h) \sum_{\alpha'} \left(\xi^\sigma_{\alpha\alpha'}\right)^* I_{x(y)} \overline{|\psi_{\alpha'}(h)\rangle}.$$

Simplifying above, just as we did for the other symmetries and operations, we obtain two matrix equations,

$$e^{-i\beta L \sum_i \delta^\alpha_i} \theta_I(\alpha) \delta_{I(\alpha)\alpha'} \xi^{I_x(\sigma)}_{\alpha'\alpha''} = (\xi^\sigma_{\alpha\alpha''})^* e^{i2\beta \sum_i (\pi - \delta^{\alpha''}_i) a_i(\alpha'')}$$

$$\Rightarrow e^{-i\beta\delta_I} B_I \xi^{I_x(\sigma)} e^{-i\beta^- a_I} = (\xi^\sigma)^*, \tag{102a}$$

$$e^{-i2\beta \sum_i (\pi + \delta^\alpha_i) a_i(\alpha)} \xi^{I_y(\sigma)}_{\alpha\alpha''} = (\xi^\sigma_{\alpha\alpha'})^* e^{-i\beta L \sum_i \delta^{\alpha'}_i} \theta_I(\alpha') \delta_{I(\alpha')\alpha''}$$

$$\Rightarrow e^{-i\beta^+ a_I} \xi^{I_y(\sigma)} \left(e^{-i\beta\delta_I} B_I\right)^{-1} = (\xi^\sigma)^*. \tag{102b}$$

Here, we have again introduced following matrices:

$$e^{-i\beta\delta_I}{}_{\alpha\alpha'} = \delta_{\alpha\alpha'} e^{-i\beta L \sum_i \delta^\alpha_i}, \quad B_{I\alpha\alpha'} = \theta_I(\alpha) \delta_{I(\alpha)\alpha'},$$

$$e^{i\beta^\pm a_I}{}_{\alpha\alpha'} = \delta_{\alpha\alpha'} e^{i2\beta \sum_i (\pi \pm \delta^{\alpha'}_i) a_i(\alpha')},$$

which defines the parameter $\beta^\pm$.

If we combine the above two equations, we arrive at every possible constraint on transition matrices from inversion symmetry alone. In particular, this can be used, in a manner similar to the one observed for $F$-moves, to constrain $\xi^\sigma$ for one given $\sigma$. Since, again, all $\sigma$'s are related by $F$-moves and mirror symmetry (for two and three particles), it suffices to focus on $\sigma = \text{id}$.

$$I_x(I_y(\text{id})) = \text{id}. \tag{103}$$

This then leads to the following constraint on $\xi^{\text{id}}$:

$$(e^{-i\beta^+ a_I})^* e^{-i\beta\delta_I} B_I \xi^{\text{id}} e^{-i\beta^- a_I} \left((e^{-i\beta\delta_I})^* B_I\right)^{-1} = \xi^{\text{id}}. \tag{104}$$

### 4.3.5 Locality constraints

We have so far determined symmetry/operations constraints on the transition matrix $\xi^\sigma$s. Further constraints can be derived considering locality constraints on the braid matrix $\chi$ itself. In Section 4.2, we already commented on the way locality factors into the coherent state formalism: Matrix elements of local operators between basis states $|a, \alpha\rangle$, $|a', \alpha'\rangle$ can be non-zero only if the underlying root states $|a, \alpha\rangle$, $|a', \alpha'\rangle$ differ from one another only locally. This is in particular true for matrix elements of the identity operator, i.e, the inner products $\langle a', \alpha'|a, \alpha\rangle$. In particular, we argued in this way that the Berry connection matrix $\langle \psi_\alpha|\nabla_{ij}\psi_{\alpha'}(h)\rangle$, where $\nabla_{ij}$ contains derivatives with respect to the coordinates of the moving quasiholes $i$ and $j$, is diagonal in $\alpha$, $\alpha'$ *as long as* quasiholes are well separated in $h_x$. It is useful to contemplate a calculation of the Berry matrix along the whole exchange path using the $|\psi_\alpha(h)\rangle$ coherent state, even for segments where the $h_x$-separation of the braided quasiholes is small. This should be possible in principle, even though we avoid technicalities by using the dual states $\overline{|\psi_\alpha(h)\rangle}$ along those segments.

Let's contemplate a pair of quasiholes that initially, for well separated $h_x$, is in the first of the following two topological sectors:

transition is possible between:
1011011 00200200200 1101101 ,
10110110 1 0110110 1 01101101 .

The pair will remain in the first of these two topological sectors while well separated in $x$; however, at some point along the exchange path, the intermediate 200-string of the pattern will become small. By the above argument, off-diagonal matrix elements in the Berry connection matrix between the first and the second sector are then possible. Hence, the transition between these two sectors as a result of the exchange path is possible. Note that we regard the "outer" 110-strings as essentially infinitely long during the process, as we consider the braided pair is well removed in $x$ from all other quasiholes. In particular, then no transition is possible during which these outer strings change. An example is the following:

$$\text{transition is not possible between:}$$
$$0200200\ 11011011\ 0020020020\,,$$
$$1011011\ 00200200200\ 1101101\,.$$

Indeed, a stronger statement is possible. Consider the first of the two sectors above. It is not possible to replace the inner 110-string with any other string such that two charge 1/3 domain walls remain between strings. Thus, given the outer 200-strings, by locality (and charge conservation), we cannot make a transition from the first of these two sectors into *any* other sector.

The above considerations impose strong constraints on the braid matrix. Let us write the topological sector label $\alpha$ as $\alpha = (c, \tilde{\alpha})$, as shown in Tables 3 and 4. Here, $\tilde{\alpha}$ is thought of as labeling a "supersector" of translationally related sub-sectors $c$. This leads to the following structure of the braid matrix:

$$\chi_{\alpha\alpha'} = \delta_{cc'} \tilde{\chi}_{\tilde{\alpha}\tilde{\alpha}'}\,. \tag{105}$$

Indeed, the labeling is such that identical "outer" strings only happen for identical $c$. Moreover, sectors with different $c$ but same $\alpha$ are related by translation, justifying the above factorization. Further constraints apply to the super-sector factor $\tilde{\chi}$. For two quasiholes, the above arguments imply:

$$\tilde{\chi}^{(2)} = \begin{pmatrix} \times & 0 & 0 \\ 0 & \times & \times \\ 0 & \times & \times \end{pmatrix}, \tag{106}$$

where $\times$ stands for elements that are not necessarily zero. The zeros, on the other hand, are required precisely by the arguments made for the two cases studied above for two domain walls. Similar arguments imply the following structure for $\tilde{\chi}$ for three quasiholes, where we assume that the two leftmost quasiholes are being braided:

$$\tilde{\chi}^{(3)} = \begin{pmatrix} + & 0 & 0 & 0 \\ 0 & + & 0 & 0 \\ 0 & 0 & + & + \\ 0 & 0 & + & + \end{pmatrix}, \tag{107}$$

where we used a different symbol, $+$, for matrix elements not necessarily zero, for reasons that will become apparent shortly. The study of three quasiholes is necessary in this formalism, among other things, because certain pairings of domain walls require a third domain wall on the torus. This is true for the leftmost domain wall with $\tilde{\alpha} = 1$, and $\tilde{\alpha} = 2$. Our locality arguments then immediately imply that the braiding in these sectors, again for the two leftmost quasiholes, is diagonal, as shown above. For $\tilde{\alpha} = 3$ and $\tilde{\alpha} = 4$, however, the braiding of the two leftmost quasiholes involve pairs of domains walls that were already resent in the two-quasihole cases. In those cases, locality implies that the result does not depend on the presence or absence of a third, far removed, quasihole. For these reasons, the lower right $2 \times 2$ blocks of $\tilde{\chi}^{(2)}$ and $\tilde{\chi}^{(3)}$ must be same:

$$\begin{pmatrix} \times & \times \\ \times & \times \end{pmatrix} = \begin{pmatrix} + & + \\ + & + \end{pmatrix}. \tag{108}$$

In the remainder of this section, we will use the symmetry and locality constraints discussed above, respectively, on the transition matrices and the braid matrix for two and three quasiholes to determine braiding statistics.

## 4.4 Braid matrix for two quasiholes

We begin by considering the linear constraints on the transition matrix $\xi^{\text{id}}$ from translation symmetries. Using Eqs. (89), (90b), along with the inversion symmetry $\delta_1^\alpha = -\delta_2^\alpha \mod 2\pi$, $\xi^{\text{id}}$ is reduced to the following form,

$$\xi^{\text{id}} = \begin{pmatrix} \tilde{\xi}_{11}V & \tilde{\xi}_{12}VI & \tilde{\xi}_{13}VI \\ \tilde{\xi}_{21}IV & \tilde{\xi}_{22}IVI & \tilde{\xi}_{23}IVI \\ \tilde{\xi}_{31}IV & \tilde{\xi}_{32}IVI & \tilde{\xi}_{33}IVI \end{pmatrix}, \tag{109}$$

with

$$V = \begin{pmatrix} 1 & \Omega^2 & \Omega \\ \Omega^2 & 1 & \Omega \\ \Omega & \Omega & \Omega \end{pmatrix}, \qquad I = \begin{pmatrix} 1 & 0 & 0 \\ 0 & -1 & 0 \\ 0 & 0 & -1 \end{pmatrix}, \tag{110}$$

in the $\alpha$ basis of Table 3, where, to simplify expressions, we introduced $\Omega = e^{i2\pi/3}$. From $F$-moves, Eq. (99), it turns out that $\tilde{\xi}_{22} = \Delta^2 p^2 \tilde{\xi}_{11}$, $\tilde{\xi}_{23} = -\Delta p \tilde{\xi}_{13}$, and $\tilde{\xi}_{32} = -\Delta p \tilde{\xi}_{31}$ with $p = -\Omega^{-1-s}$ and $\Delta = \Omega^{-\text{La}}$, where a is related to the one unknown $\delta$-parameter related to the ...200200 110110... type domain wall via $\delta_i^\alpha = 2\pi a$, $a = 0, \frac{1}{2}$. Note that for two quasiholes on the torus, $L = 1 \mod 3$. Using, finally, the inversion symmetry constraint Eq. (104), most of the parameters $\tilde{\xi}_{ij}$ are forced to vanish if $a = 1/2$. The only solution consistent with $a = 1/2$ can only have $4s = -1 \mod 3$ with $\tilde{\xi}_{12}$, $\tilde{\xi}_{21} = \Delta^2 p^2 \tilde{\xi}_{12}$, and $\tilde{\xi}_{33}$ non-zero, and in Eq. (84), gives a diagonal braid matrix. $a = 1/2$, $4s = -1 \mod 3$ solution, however, can be shown to be inconsistent while considering mirror symmetry for the three quasiholes case. We will thus proceed with $a = 0$, thus fixing the last remaining $\delta$-parameter. In summary, we have reduced $\xi^{\text{id}}$, Eq. (109), to the following:

$$\xi^{\text{id}} = \begin{pmatrix} \tilde{\xi}_{11}V & \tilde{\xi}_{12}VI & \tilde{\xi}_{13}VI \\ \tilde{\xi}_{12}IV & p^2\tilde{\xi}_{11}IVI & -\tilde{\xi}_{13}IVI \\ \tilde{\xi}_{31}IV & -p\tilde{\xi}_{31}IVI & \tilde{\xi}_{33}IVI \end{pmatrix}. \tag{111}$$

We may now use the above form for $\xi^{\text{id}}$ to continue the program described above. For determining $\xi^{\sigma_1}$, where $\sigma_1$ denotes the only other configuration for two particles, we may use either $F$-moves, Eq. (98), or mirror symmetry Eq. (102). Since one involves complex conjugation, and the other does not, by comparison we may express all complex conjugated remaining $\tilde{\xi}_{ij}$ parameters through their un-conjugated counterparts in the following. We then obtain the braid matrix from Eq. (84). Comparing this braid matrix with the locality constraint (106) yields two quadratic equations in the $\tilde{\xi}_{ij}$-parameters. Furthermore, one obtains four more quadratic equations from the requirement that the braid matrix is unitary. This yields the following six non-linear equations,

$$2p\tilde{\xi}_{11}\tilde{\xi}_{12} + p\tilde{\xi}_{13}^2 = -\Omega^2,$$
$$2p\tilde{\xi}_{31}^2 - \tilde{\xi}_{33}^2 = -\Omega^2,$$
$$p^2\tilde{\xi}_{11}^2 + \tilde{\xi}_{12}^2 - p\tilde{\xi}_{13}^2 = 0,$$

$$\tilde{\xi}_{31}(-p\tilde{\xi}_{11}+p\tilde{\xi}_{12})+\tilde{\xi}_{13}\tilde{\xi}_{33}=0,$$
$$(1+p^2)\tilde{\xi}_{11}\tilde{\xi}_{12}-p\tilde{\xi}_{13}^2=0,$$
$$\tilde{\xi}_{31}(\tilde{\xi}_{11}-p\tilde{\xi}_{12})+\tilde{\xi}_{13}\tilde{\xi}_{33}=0, \tag{112}$$

where the first four express unitarity, and the last two locality. From these equations, all the $\tilde{\xi}_{ij}$ can be determined when the parameter $p$, is known.

$$\tilde{\xi}_{11}^2=-\frac{\Omega^2}{(1+p)^2},$$
$$\tilde{\xi}_{12}=\tilde{\xi}_{11},$$
$$\tilde{\xi}_{13}^2=\xi_{31}^2=(p+p^{-1})\tilde{\xi}_{11}^2,$$
$$\tilde{\xi}_{33}^2=(1-p)^2\tilde{\xi}_{11}^2. \tag{113}$$

One then obtains for the braid matrix:

$$\tilde{\chi}^{(2)}=e^{-i\beta\pi}\begin{pmatrix} p^{-1} & 0 & 0 \\ 0 & p(p+p^{-1}-1) & \pm(1-p)\sqrt{p+p^{-1}} \\ 0 & \pm(1-p)\sqrt{p+p^{-1}} & p+p^{-1}-1 \end{pmatrix}. \tag{114}$$

Not yet having enough information to determine the remaining parameter gives us another reason to proceed to three particles. Indeed, the remaining parameters can ultimately be determined from Eq. (108). For completeness, we mention that in writing Eq. (114), we have tacitly assumed $p\neq\pm i$. For $p=\pm i$ one finds additional solutions that lead to a diagonal braid matrix. These solutions turn out to be inconsistent when compared with the three quasihole result below, as already remarked in a similar context above. For details, we refer to Ref. [33], where equations differ in detail, but the procedure is similar.

## 4.5 Braid matrix for three quasiholes

For three quasiholes, we may proceed in a manner that is perfectly analogous to that for two quasiholes in the preceding section. As opposed to two quasiholes (see Table 3), in this case the total number of fermions is always odd for all topological sectors (see Table 4). Hence, all the fermionic sign-factors $\theta_T$, $\theta_F$, and $\theta_I$ are identical to the bosonic ones [34], since in those topological sectors permutations do not distinguish bosons from fermions. Therefore, the formulas in this section will be identical to corresponding formulas in Ref. [34]. We will nonetheless reproduce them here for self-containedness.

Again, there are a great multitude of simple linear constraint rendering many of the elements of $\xi^{\text{id}}$ proportional to one another. These are the constraints ended in translational-, $F$-move, and inversion symmetry, $\xi^{\text{id}}$ by itself, to wit, Eqs. (89), (90b), (99), and (104). Indeed, in the present case, translational symmetry by itself leads to a major simplification of the $\xi^\sigma$, in that they factorize via

$$\xi^\sigma=\tilde{\xi}^\sigma\otimes U, \tag{115}$$

where $\tilde{\xi}^\sigma$ acts on supersectors $\tilde{\alpha}$, and $U$ acts on subsectors $c$, and where

$$U=\begin{pmatrix} 1 & \Omega & \Omega^2 \\ \Omega & \Omega & \Omega \\ \Omega^2 & \Omega & 1 \end{pmatrix}, \qquad \tilde{\xi}^{\text{id}}=\begin{pmatrix} \tilde{\xi}_{11} & \tilde{\xi}_{12} & \tilde{\xi}_{13} & \tilde{\xi}_{14} \\ \tilde{\xi}_{21} & \tilde{\xi}_{22} & \tilde{\xi}_{23} & \tilde{\xi}_{24} \\ \tilde{\xi}_{31} & \tilde{\xi}_{32} & \tilde{\xi}_{33} & \tilde{\xi}_{34} \\ \tilde{\xi}_{41} & \tilde{\xi}_{42} & \tilde{\xi}_{43} & \tilde{\xi}_{44} \end{pmatrix}.$$

This factorization happens here, and not for two quasiholes, because of the aforementioned absence of non-trivial fermionic-phase factors. The elements of $\tilde{\xi}^{\text{id}}$, which we will denote by

$\tilde{\xi}_{ij}$, can then be further determined using the global path operation and mirror symmetry, Eqs. (99), (104), yielding the following additional relations:

$$\tilde{\xi}_{22} = \tilde{\xi}_{33} = p^2 \tilde{\xi}_{11}, \tag{116a}$$

$$\tilde{\xi}_{31} = \tilde{\xi}_{13} = \tilde{\xi}_{21} = \tilde{\xi}_{12}, \tag{116b}$$

$$\tilde{\xi}_{32} = \tilde{\xi}_{23} = -p\tilde{\xi}_{12}, \tag{116c}$$

$$\tilde{\xi}_{34} = \tilde{\xi}_{24} = -p\tilde{\xi}_{14}, \tag{116d}$$

$$\tilde{\xi}_{43} = \tilde{\xi}_{42} = -p\tilde{\xi}_{41}. \tag{116e}$$

Again, we may now evaluate the braid matrix by plugging in the above into Eq. (84), by first obtaining $(\xi^{\sigma_2})^*$ for the other configuration $(\sigma_2)$ that appears when the leftmost pair is braided, starting in the configuration $\sigma_0 = \mathrm{id}$. This can be done by subsequently applying first $I_y$ via Eq. (102b), and then $F_x$ via Eq. (98a), to the transition matrix $\xi^{\mathrm{id}}$. For the resulting braid matrix we then obtain

$$\tilde{\chi}^{(3)} = \begin{pmatrix} \tilde{\xi}_{11} & \tilde{\xi}_{12} & \tilde{\xi}_{12} & \tilde{\xi}_{14} \\ \tilde{\xi}_{12} & \tilde{\xi}_{11}p^2 & -\tilde{\xi}_{12}p & -\tilde{\xi}_{14}p \\ \tilde{\xi}_{12} & -\tilde{\xi}_{12}p & \tilde{\xi}_{11}p^2 & -\tilde{\xi}_{14}p \\ \tilde{\xi}_{41} & -\tilde{\xi}_{41}p & -\tilde{\xi}_{41}p & \tilde{\xi}_{44} \end{pmatrix} \cdot \begin{pmatrix} -\tilde{\xi}_{12}p\Omega & \tilde{\xi}_{11}p^2\Omega & \tilde{\xi}_{12}\Omega & -\tilde{\xi}_{41}p\Omega \\ -\tilde{\xi}_{11}p\Omega & \tilde{\xi}_{12}\Omega & -\frac{\tilde{\xi}_{12}\Omega}{p} & \tilde{\xi}_{41}\Omega \\ -\tilde{\xi}_{12}p\Omega & -\tilde{\xi}_{12}p\Omega & -\tilde{\xi}_{11}p\Omega & -\tilde{\xi}_{41}p\Omega \\ -\tilde{\xi}_{14}p\Omega & -\tilde{\xi}_{14}p\Omega & \tilde{\xi}_{14}\Omega & \tilde{\xi}_{44}\Omega \end{pmatrix}, \tag{117}$$

where again we only display the "supersector" factor in Eq. (105), and where the zeros come from the locality constraint Eq. (107). These matrix elements are not automatically zero, but rather, enforcing their vanishing gives us the following three constraints:

$$p^2\tilde{\xi}_{11}^2 - (p-1)\tilde{\xi}_{12}^2 - p\tilde{\xi}_{14}^2 = 0, \tag{118a}$$

$$(p^2 - p)\tilde{\xi}_{11}\tilde{\xi}_{12} + \tilde{\xi}_{12}^2 - p\tilde{\xi}_{14}^2 = 0, \tag{118b}$$

$$-p\tilde{\xi}_{11}\tilde{\xi}_{41} - (p-1)\tilde{\xi}_{12}\tilde{\xi}_{41} + \tilde{\xi}_{14}\tilde{\xi}_{44} = 0. \tag{118c}$$

Finally, by imposing the locality of $\xi^{\mathrm{id}}$ one imposes that of $\tilde{\xi}^{\mathrm{id}}$, as $U$ is already unitary. This yields the following four additional equations:

$$|\tilde{\xi}_{11}|^2 + 2|\tilde{\xi}_{12}|^2 + |\tilde{\xi}_{14}|^2 = 1, \tag{119a}$$

$$3|\tilde{\xi}_{41}|^2 + |\tilde{\xi}_{44}|^2 = 1, \tag{119b}$$

$$\tilde{\xi}_{12}\tilde{\xi}_{11}^* + p^2\tilde{\xi}_{11}\tilde{\xi}_{12}^* - p|\tilde{\xi}_{12}|^2 - p|\tilde{\xi}_{14}|^2 = 0, \tag{119c}$$

$$\tilde{\xi}_{41}\tilde{\xi}_{11}^* - 2p\tilde{\xi}_{41}\tilde{\xi}_{12}^* + \tilde{\xi}_{44}\tilde{\xi}_{14}^* = 0. \tag{119d}$$

The non-linear equations (116-118) have the following [34] solution:

$$\tilde{\xi}_{11} = -\frac{e^{i\theta_1}}{(1+p)^2} \quad \tilde{\xi}_{12} = \tilde{\xi}_{11},$$

$$\tilde{\xi}_{14}^2 = e^{-i2\theta_2}\tilde{\xi}_{41}^2 = (p + p^{-1} - 1)\tilde{\xi}_{11}^2,$$

$$\tilde{\xi}_{44} = e^{i\theta_2}\tilde{\xi}_{11}, \tag{120}$$

in terms of two additional unknown phases $\theta_1$ and $\theta_2$. In Eq. (117), this gives the following result for the braid matrix:

$$\tilde{\chi}^{(3)} = e^{-i\beta\pi}e^{i\theta_1} \begin{pmatrix} 1 & 0 & 0 & 0 \\ 0 & 1 & 0 & 0 \\ 0 & 0 & p(1-p) & \pm e^{i\theta_2}p\sqrt{p + p^{-1} - 1} \\ 0 & 0 & \pm e^{i\theta_2}p\sqrt{p + p^{-1} - 1} & e^{i2\theta_2}(1-p) \end{pmatrix}. \tag{121}$$

As a final step, it turns out that the remaining unknowns are largely determined by the locality argument requiring consistency between the two quasihole and the three quasihole braid matrix, Eq. (108):

$$\begin{pmatrix} p(p+p^{-1}-1) & \pm(1-p)\sqrt{p+p^{-1}} \\ \pm(1-p)\sqrt{p+p^{-1}} & p+p^{-1}-1 \end{pmatrix} = e^{i\theta_1}\begin{pmatrix} p(1-p) & \pm e^{i\theta_2}p\sqrt{p+p^{-1}-1} \\ \pm e^{i\theta_2}p\sqrt{p+p^{-1}-1} & e^{i2\theta_2}(1-p) \end{pmatrix}.$$
(122)

Comparing matrix elements yields the following equations:

$$e^{i\theta_1} = p^2, \quad e^{i\theta_2} = 1, \tag{123a}$$

$$p + p^{-1} = \varphi = \frac{1+\sqrt{5}}{2} \Rightarrow p = e^{\pm i\pi/5}, \tag{123b}$$

and from $p = -\Omega^{-1-s}$, $s = \frac{1}{2} \pm \frac{3}{10}$.

As we will now explain, this determines all braiding processes in terms of two possible and closely related non-Abelian solutions. Eqs. (123) in the two quasihole and three quasibole braid matrices, Eq. (114) and Eq. (121), respectively, then give the following

$$\tilde{\chi}^2 = \Omega\begin{pmatrix} p & 0 & 0 \\ 0 & p^{-1}\varphi^{-1} & p^2\varphi^{-1/2} \\ 0 & p^2\varphi^{-1/2} & \varphi^{-1} \end{pmatrix}, \quad \tilde{\chi}^3 = \Omega\begin{pmatrix} p^{-2} & 0 & 0 & 0 \\ 0 & p^{-2} & 0 & 0 \\ 0 & 0 & p^{-1}\varphi^{-1} & p^2\varphi^{-1/2} \\ 0 & 0 & p^2\varphi^{-1/2} & \varphi^{-1} \end{pmatrix}. \tag{124}$$

Together, these equations imply the following when applied to the braiding of any pair of quasiholes, in a pattern with $n_h$ quasiholes: If the pair was linked by a 110-string bounded by two 200-strings (as for two quasiholes, $\alpha = 1, 2, 3$), the state picks up a phase $e^{-i\beta\pi}p$. If the pair was linked by a 110-string bounded by one 110-string and one 200-string (as for three quasiholes, $\alpha = 1$–6), the state picks up a phase $e^{-i\beta\pi}p^{-2}$. Finally, if the linking string is either 200 or 110, and is bounded by 110-strings on both sides (as is is for the last six $\alpha$'s for both two and three quasiholes), the state stays in the same topological sector with an amplitude $e^{-i\beta\pi}p^{-1}\varphi^{-1}$ if the linking string is 200, and an amplitude $e^{-i\beta\pi}p^{-1}\varphi^{-1}$ if the linking string is 110. Furthermore, there is an amplitude for transitioning between these two respective sectors of $e^{-i\beta\pi}p^2\varphi^{-\frac{1}{2}}$. It is easy to see that these off-diagonal blocks, which we just described, have the same eigenvalues as those appearing in the diagonal blocks described above. In the topological sector basis, one can thus determine the result of any braiding process, for each of the two solutions associated to the two ways to resolve the $\pm$ sign in these expressions. One may easily see, though, that these two solutions are related by an Abelian phase plus complex conjugation. Moreover, they share the same Abelian phase with the bosonic case of Ref. [34]. Just as in this reference, one may therefore show that these solutions describe Fibonacci-type anyons. However, we stress once more that both the assumptions of fermionic constituent particles, *as well as* that of root state entanglement of a certain type have been essential to arrive at this solution using the coherent state method.

# 5 Partons as the densest zero modes

In the previous sections, we have developed a general framework and an organizing principle (the EPP) for determining the densest zero-energy state of frustration-free QH Hamiltonians. While our second quantized technique is applicable to any k-body Hamiltonian with LL mixing, it is often the case that QH physics is studied in the first quantization language. In this section, we make connections to the theory of symmetric (and antisymmetric) polynomials in holomorphic and anti-holomorphic variables, which correspond to the first-quantized description

of the QH problem. For simplicity, we work in the symmetric gauge. In the LLL (the holomorphic case), many tools exist to uniquely identify subsets of these polynomials as determined by various clustering conditions enforced by frustration-free Hamiltonians. For multiple LLs, these tools generally do not work. The present section is devoted to the development of alternative methods. As we will show, the parton states have the fundamental polynomial property of being the densest zero modes of certain frustration-free QH Hamiltonians in presence of multiple LL mixing.

## 5.1 Multivariate polynomials with the $M$-clustering property

Let $\mathcal{A}_N$ be the algebra of multivariate polynomials $P(Z, \bar{Z})$ in variables $(Z, \bar{Z})$, where $Z = \{z_1, z_2, \cdots, z_N\}$ and $\bar{Z} = \{\bar{z}_1, \bar{z}_2, \cdots, \bar{z}_N\}$, with $z_i = x_i + iy_i$ and $\bar{z}_i = x_i - iy_i$, $i = 1, \cdots, N$. Polynomials $P(Z, \bar{Z})$ consist of sums of monomials, which are products of (not normalized and without Gaussian factors) non-orthogonal single-particle orbitals (already used to define the pseudofermion basis in Eq. (53)),

$$\phi_{\alpha_i}(z_i, \bar{z}_i) = \phi_{\alpha_i}(i) = \bar{z}_i^{n_i} z_i^{s_i}, \qquad s_i = j_i + n_i. \tag{125}$$

Here, $\alpha_i = (n_i, s_i)$ represents a pair of non-negative integers. We will be interested in working within linear subspaces satisfying $0 \le n_i \le N_L - 1$ (i.e., those restricted to $N_L$ LLs). Moreover, as we will now discuss, finite dimensional subspaces may be obtained by placing additional restrictions on (the number operator $\hat{n}_i^b = b_i^\dagger b_i$ eigenvalues of Eq. (5)) $s_i$, via $0 \le s_i \le s_{\max}$ and/or restrictions on the total angular momentum of the polynomial. Finally, we will further restrict our linear subspaces of interest by the condition that their elements are either symmetric or antisymmetric under the exchange operations

$$(z_i, \bar{z}_i) \longleftrightarrow (z_j, \bar{z}_j), \quad i, j = 1, \cdots, N, i \ne j, \tag{126}$$

of all pairs of variables. Let the total angular momentum operator be defined as

$$\hat{J} = \hbar \sum_{i=1}^N (z_i \partial_{z_i} - \bar{z}_i \partial_{\bar{z}_i}). \tag{127}$$

The application of the total angular momentum operator on any monomial in the variables $(Z, \bar{Z})$ leaves the monomial invariant up to a multiplicative (angular momentum) factor $J = \hbar \sum_{i=1}^N (s_i - n_i)$. It is clear that the total angular momentum operator defines a linear map $\hat{J} : \mathcal{A}_N \to \mathcal{A}_N$ that also preserves all linear subspaces defined above. $\hat{J}$ has the natural (infinite) basis of eigenstates (125). However, we shall now consider the finite-dimensional linear subspaces $\mathcal{H}_{N,J,n}$ of $\mathcal{A}_N$ of polynomials of angular momentum less than or equal to $J$ and maximum degree $n = N_L - 1$ in each $\bar{z}_i$, and their (anti-)symmetrized subspaces $(\hat{A})\hat{\mathcal{S}}\mathcal{H}_{N,J,n}$. Note that, moreover, all such polynomials automatically have bounded $s_i$, i.e., satisfy $0 \le s_i \le s_{\max}$ with an appropriately chosen $s_{\max}$ depending on $N$, $J$, and $n$. The subspaces $\mathcal{H}_{N,J,n}$ form finite dimensional Hilbert spaces having an inner product ($\ell = 1/\sqrt{2}$)

$$\langle P | P' \rangle = \int dZ d\bar{Z} \, \bar{P}(Z, \bar{Z}) P'(Z, \bar{Z}) \, e^{-\frac{1}{2} \sum_{i=1}^N z_i \bar{z}_i}. \tag{128}$$

From now on, we will be working in these finite dimensional Hilbert spaces $\mathcal{H}_{N,J,n}$.

Within the space of polynomials $\mathcal{H}_{N,J,n}$, there are families of polynomials that have special properties. A polynomial $P(Z, \bar{Z}) \in \mathcal{H}_{N,J,n}$ has the $M$-clustering property, with $M$ a positive integer, in the pair $(i, j)$ if

$$P(Z, \bar{Z}) = \sum_{q=0}^M z_{ij}^q \bar{z}_{ij}^{M-q} P_q(Z, \bar{Z}) = P^{(M)}(Z, \bar{Z}), \tag{129}$$

where $z_{ij} = z_i - z_j$, $\bar{z}_{ij} = \bar{z}_i - \bar{z}_j$, and $P_q(Z, \bar{Z}) \in \mathcal{H}_{N, J+M-2q, n}$. If furthermore $P(Z, \bar{Z}) \in (\hat{\mathcal{A}})\hat{\mathcal{S}}\mathcal{H}_{N,J,n}$, then $P(Z, \bar{Z})$ is a polynomial (anti-)symmetric with respect to variables exchanges $(z_i, \bar{z}_i) \longleftrightarrow (z_j, \bar{z}_j)$. Clearly, polynomials with the $M$-clustering property can only exist if $s_{\max} \geq M$ or $n \geq M$. Those polynomials with the $M$-clustering property in all pairs $(i, j)$ form a subspace $\mathcal{H}_{N,J,n,M} \subset \mathcal{H}_{N,J,n}$. Moreover, $P(Z, \bar{Z}) \in \mathcal{H}_{N,J,n}$ has the $M$-clustering property in the pair $(i, j)$, iff $\forall s + t < M$,

$$Q_{ij}^{st} P(Z, \bar{Z}) \equiv \partial_{z_{ij}}^s \partial_{\bar{z}_{ij}}^t P(Z, \bar{Z}) \Big|_{z_i = z_j, \bar{z}_i = \bar{z}_j} = 0. \tag{130}$$

A little reflection shows that for $N$ even and $P(Z, \bar{Z}) \in (\hat{\mathcal{A}})\hat{\mathcal{S}}\mathcal{H}_{N,J,n,M}$, Eq. (129) can be written as

$$P^{(M)}(Z, \bar{Z}) = \sum_{\mathbf{q}=0}^{M} z_{12}^{q_1} \bar{z}_{12}^{M-q_1} \ldots z_{N-1N}^{q_{N/2}} \bar{z}_{N-1N}^{M-q_{N/2}} P_{\mathbf{q}}(Z, \bar{Z}), \tag{131}$$

where $\mathbf{q} \equiv (q_1, q_2, \ldots, q_{N/2})$, and $P_{\mathbf{q}}(Z, \bar{Z})$ is a polynomial symmetric under the exchange of all pair of coordinates $(z_{2i-1}, \bar{z}_{2i-1}) \longleftrightarrow (z_{2i}, \bar{z}_{2i})$, $i = 1, \cdots, N/2$.

We will mostly be interested in the antisymmetric subspace $\hat{\mathcal{A}}\mathcal{H}_{N,J,n,M}$ of polynomials with the $M$-clustering property. Slater determinants

$$\chi_p(Z, \bar{Z}) = \begin{vmatrix} \phi_{\alpha_1}(1) & \phi_{\alpha_1}(2) & \cdots & \phi_{\alpha_1}(N) \\ \phi_{\alpha_2}(1) & \phi_{\alpha_2}(2) & \cdots & \phi_{\alpha_2}(N) \\ \vdots & \vdots & \cdots & \vdots \\ \phi_{\alpha_N}(1) & \phi_{\alpha_N}(2) & \cdots & \phi_{\alpha_N}(N) \end{vmatrix} \tag{132}$$

represent the simplest examples of those polynomials with an $M = 1$ clustering property, since they do have a linear behavior as two-particles approach each other.[4] Specifically, any Slater

---

[4]This is due to the fact that any Slater determinant for $N$-particles can be expressed as

$$\chi^{\text{even}} = \frac{1}{\mathcal{N}} \sum_{\sigma \in S_N} \text{sign}(\sigma) D_{\alpha_{\sigma_1} \alpha_{\sigma_2}}(1, 2) D_{\alpha_{\sigma_3} \alpha_{\sigma_4}}(3, 4) \ldots D_{\alpha_{\sigma_{N-1}} \alpha_{\sigma_N}}(N-1, N),$$

where $\sigma \in S_N$ are permutations of particle indices

$$D_{\alpha_{\sigma_1} \alpha_{\sigma_2}}(1, 2) = \begin{vmatrix} \phi_{\alpha_{\sigma_1}}(1) & \phi_{\alpha_{\sigma_1}}(2) \\ \phi_{\alpha_{\sigma_2}}(1) & \phi_{\alpha_{\sigma_2}}(2) \end{vmatrix},$$

when $N \in$ even, while

$$\chi^{\text{odd}} = \frac{1}{\mathcal{N}} \sum_{\sigma \in S_N} \text{sign}(\sigma) D_{\alpha_{\sigma_1} \alpha_{\sigma_2} \alpha_{\sigma_3}}(1, 2, 3) D_{\alpha_{\sigma_4} \alpha_{\sigma_5}}(4, 5) \ldots D_{\alpha_{\sigma_{N-1}} \alpha_{\sigma_N}}(N-1, N),$$

when $N \in$ odd, where $\mathcal{N}$ is a normalization factor. In the vicinity of a coincidence hyperplane for particles $1, 2$ for example, one gets as particle coordinate $1$ approaches particle coordinate $2$, i.e., $\mathbf{r}_1 \rightarrow \mathbf{r}_2$,

$$D_{\alpha_1 \alpha_2}(1, 2) \approx \left( \phi_{\alpha_1}(1) \nabla \phi_{\alpha_2}(1) - \phi_{\alpha_2}(1) \nabla \phi_{\alpha_1}(1) \right) \cdot (\mathbf{r}_2 - \mathbf{r}_1),$$

while for the $3 \times 3$ determinant

$$D_{\alpha_1 \alpha_2 \alpha_3}(1, 2, 3) \approx \frac{1}{2} \sum_{\sigma \in S_3} \text{sign}(\sigma) \phi_{\alpha_{\sigma_1}}(3) \left( \phi_{\alpha_{\sigma_2}}(1) \nabla \phi_{\alpha_{\sigma_3}}(1) - \phi_{\alpha_{\sigma_3}}(1) \nabla \phi_{\alpha_{\sigma_2}}(1) \right) \cdot (\mathbf{r}_2 - \mathbf{r}_1).$$

Since the orbitals are linearly independent and have well-defined derivatives, the expression inside the parenthesis (the Wronskian) is nonzero. Hence,

$$|\chi(\mathbf{r}_i \rightarrow \mathbf{r}_j)| \propto |\mathbf{r}_i - \mathbf{r}_j|.$$

Applied to our orbital basis $\phi_{\alpha_{\sigma_i}}(i) = \bar{z}_i^{n_{\sigma_i}} z_i^{s_{\sigma_i}}$,

$$D_{\alpha_{\sigma_1} \alpha_{\sigma_2}}(1, 2) = \bar{z}_1^{n_{\sigma_2}} z_1^{s_{\sigma_2}} \bar{z}_2^{n_{\sigma_2}} z_2^{s_{\sigma_2}} (\bar{z}_1^{\delta n} z_1^{\delta s} - \bar{z}_2^{\delta n} z_2^{\delta s}),$$

determinant satisfies the following identity

$$\chi_p(Z,\bar{Z}) = S_p(Z,\bar{Z})z_{ij} + \tilde{S}_p(Z,\bar{Z})\bar{z}_{ij}, \tag{133}$$

where $S_p$ and $\tilde{S}_p$ are symmetric polynomials with respect to the coordinate exchange $(z_i, \bar{z}_i) \leftrightarrow (z_j, \bar{z}_j)$. Another example of polynomials with the $M$-clustering property are parton-like states $\Psi_p(Z,\bar{Z})$, defined as a product of $M$ Slater determinants

$$\Psi_p(Z,\bar{Z}) = \prod_{\mu=1}^{M} \chi_{p_\mu}(Z,\bar{Z}), \tag{134}$$

where $M \in$ odd for fermions and $M \in$ even for bosons. Using Eq. (133) for each Slater determinant in $\Psi_p(Z,\bar{Z})$, it is straightforward to show that $\Psi_p(Z,\bar{Z})$ is an element of $\mathcal{H}_{N,J,n,M}$ and can be written as the $P(Z,\bar{Z})$ of Eq. (129). Although, $\Psi_p(Z,\bar{Z})$ is an element of $(\hat{A})\hat{S}\mathcal{H}_{N,J,n,M}$, i.e., the (anti-)symmetric subspace of $\mathcal{H}_{N,J,n,M}$, it is not clear whether parton-like states linearly generate this subspace. While in the following, we will be mostly concerned with the antisymmetric case, our reasoning and results carry, without difficulty, to the symmetric case.

## 5.2 Schmidt decomposition of M-clustering polynomials

The Schmidt decomposition of a many-body state can be used to study the non-trivial properties of the system such as entanglement entropy. Entanglement properties are often employed to determine the particular topological phase of matter that a given many-body state may belong to [76]. In this subsection, we will not emphasize the entanglement properties but rather demonstrate an analog of the Schmidt decomposition to polynomials $P(Z,\bar{Z})$ that satisfy the $M$-clustering property.

**Lemma 1:** Let $P \in \hat{A}\mathcal{H}_{N,J,n,M}$ and $1 \le n < N$. Then

$$P = \hat{A} \sum_{\lambda} c_\lambda P_n^\lambda(1,2,\ldots,n)\tilde{P}_{N-n}^\lambda(n+1,\ldots,N), \tag{135}$$

where $\lambda$ runs over a finite index set, and $P_n^\lambda \in \hat{A}\mathcal{H}_{n,J,n,M}$, $\tilde{P}_{N-n}^\lambda \in \hat{A}\mathcal{H}_{N-n,J,n,M}$.

**Proof:** Note that, so far, we have been considering abstract polynomials. Two such abstract polynomials are identical if and only if they are identical as maps from $\mathbb{C}^N$ to $\mathbb{C}$, since all the coefficients are encoded in the associated maps via differential operations. We will now identify polynomials with their associated evaluation maps. For $P_n^\lambda$ we now choose an orthonormal basis $\{P_n^\lambda\}$ of $\hat{A}\mathcal{H}_{n,J,n,M}$. If we now fix $z_i, \bar{z}_i$ to arbitrary complex numbers $a_i, \bar{a}_i$ for $i > n$, then $P(z_1, \bar{z}_1, \ldots, z_n, \bar{z}_n, a_{n+1}, \bar{a}_{n+1}, \ldots, a_N, \bar{a}_N)$ is an element of $\hat{A}\mathcal{H}_{n,J,n,M}$. As a result

$$P(z_1, \bar{z}_1, \ldots, z_n, \bar{z}_n, a_{n+1}, \bar{a}_{n+1}, \ldots, a_N, \bar{a}_N) = \sum_{\lambda} c_\lambda P_n^\lambda(z_1, \bar{z}_1, \ldots, z_n, \bar{z}_n)\tilde{P}_{N-n}^\lambda(a_{n+1}, \bar{a}_{n+1}, \ldots, a_N, \bar{a}_N), \tag{136}$$

where, without loss of generality, we assumed $\delta n = n_{\sigma_1} - n_{\sigma_2} \ge 0$ and $\delta s = s_{\sigma_1} - s_{\sigma_2} \ge 0$. Since

$$2(\bar{z}_1^{\delta n} z_1^{\delta s} - \bar{z}_2^{\delta n} z_2^{\delta s}) = (z_1^{\delta s} + z_2^{\delta s})(\bar{z}_1^{\delta n} - \bar{z}_2^{\delta n}) + (\bar{z}_1^{\delta n} + \bar{z}_2^{\delta n})(z_1^{\delta s} - z_2^{\delta s}),$$

and,

$$\frac{z_1^{\delta s} - z_2^{\delta s}}{z_1 - z_2} = \sum_{\ell=0}^{\delta s-1} z_1^{\delta s-\ell-1} z_2^\ell,$$

with a similar expression for $\bar{z}$, then

$$D_{\alpha_{\sigma_1}\alpha_{\sigma_2}}(1,2) = S(1,2)(\bar{z}_1 - \bar{z}_2) + \tilde{S}(1,2)(z_1 - z_2),$$

where $S(1,2)$, $\tilde{S}(1,2)$ are symmetric polynomials with no coincidence plane zeroes.

where

$$c_\lambda \tilde{P}^\lambda_{N-n} = \int dz_1 d\bar{z}_1 \cdots dz_n d\bar{z}_n \, \bar{P}^\lambda_n P \, e^{-\frac{1}{2}\sum_{i=1}^n z_i \bar{z}_i}. \tag{137}$$

It is clear that as a function of $a_i, \bar{a}_i$, the righthand side defines a polynomial in the variables $a_{n+1}, \ldots, \bar{a}_N$, which we will argue to be an element of $\hat{A}\mathcal{H}_{N-n,J,n,M}$. Since Eq. (136) holds as an identity for fixed but arbitrary $z_1, \bar{z}_1, \ldots, z_n, \bar{z}_n$ and $a_{n+1}, \ldots, \bar{a}_N$, the two sides are identical as polynomial maps and therefore as elements of $\hat{A}\mathcal{H}_{N,J,n}$. Furthermore, since the lefthand side is in $\hat{A}\mathcal{H}_{N,J,n,M}$ so is the righthand side (though individual terms are not). We can thus introduce the (anti-)symmetrizer $\hat{A}$ on the righthand side as it is in Eq. (135) without changing the polynomial.

We finally show that the polynomials $\tilde{P}^\lambda_{N-n}$ also enjoy the $M$-clustering property. This is easy: In Eq. (137) change $(a_i, \bar{a}_i) \to (z_i, \bar{z}_i)$, then apply $Q^{st}_{ij}$ for $i, j > n$ on both sides. On the righthand side, this results in 0 for $s + t < M$. Thus, $\tilde{P}^\lambda_{N-n} \in \hat{A}\mathcal{H}_{N-n,J,n,M}$. This completes the proof of the Lemma.

Indeed, a Slater determinant is the simplest example illustrating the Lemma for $M = 1$. The following identity

$$P(Z, \bar{Z}) = n!(N-n)! \begin{vmatrix} \phi_{\alpha_1}(1) & \phi_{\alpha_1}(2) & \cdots & \phi_{\alpha_1}(N) \\ \phi_{\alpha_2}(1) & \phi_{\alpha_2}(2) & \cdots & \phi_{\alpha_2}(N) \\ \vdots & \vdots & \cdots & \vdots \\ \phi_{\alpha_N}(1) & \phi_{\alpha_N}(2) & \cdots & \phi_{\alpha_N}(N) \end{vmatrix}$$
$$= \hat{A}[D_n(1, \cdots, n) D_{N-n}(n+1, \cdots, N)] \tag{138}$$

explicitly realizes a Schmidt decomposition. Here, the determinants in the argument of the antisymmetrizer are

$$D_n(1, \cdots, n) = \begin{vmatrix} \phi_{\alpha_1}(1) & \cdots & \phi_{\alpha_1}(n) \\ \vdots & \cdots & \vdots \\ \phi_{\alpha_n}(1) & \cdots & \phi_{\alpha_n}(n) \end{vmatrix}, \tag{139}$$

$$D_{N-n}(n+1, \cdots, N) = \begin{vmatrix} \phi_{\alpha_{n+1}}(n+1) & \cdots & \phi_{\alpha_{n+1}}(N) \\ \vdots & \cdots & \vdots \\ \phi_{\alpha_N}(n+1) & \cdots & \phi_{\alpha_N}(N) \end{vmatrix}.$$

More sophisticated relations appear for $M \geq 3$. The first non-trivial example is a product of an odd number of Slater determinants, i.e., the parton-like state,

$$\Psi_p(Z, \bar{Z}) = \prod_{\mu=1}^M \chi_{p_\mu}(Z, \bar{Z}) = \prod_{\mu=1}^M \hat{A}\left[ D_n^{(p_\mu)}(1, \cdots, n) D_{N-n}^{(p_\mu)}(n+1, \cdots, N) \right]. \tag{140}$$

If $M \in$ odd,

$$S(Z, \bar{Z}) = \prod_{\mu=2}^M \chi_{p_\mu}(Z, \bar{Z}) \tag{141}$$

is a totally symmetric function of all particle coordinates, and

$$\Psi_p(Z, \bar{Z}) = \hat{A}\left[ D_n^{(p_1)} D_{N-n}^{(p_1)} S(Z, \bar{Z}) \right]$$
$$= \hat{A} \sum_\lambda c_\lambda P_n^\lambda \tilde{P}^\lambda_{N-n}. \tag{142}$$

Here, we employed the property that $D_n^{(p_1)} S(Z, \bar{Z}) = \sum_\lambda c_\lambda P_n^\lambda S^\lambda_{N-n}$, where $S^\lambda_{N-n}(n+1, \ldots, N)$ is totally symmetric in its arguments, holds by arguments similar to those used in the proof of Eq. (135). Moreover, $D_{N-n}^{(p_1)} S^\lambda_{N-n} = \tilde{P}^\lambda_{N-n}$.



Figure 14: The state $\chi_4(Z,\bar{Z})$, with $L=6, N_L=4$, and $N_{orb}=18$, filled with $N=17$ and 18 particles. For the state with 17 particles, the largest angular momentum orbitals are not completely filled, i.e., the "shell" is not closed. For $N=18$ particles, $\chi_4$ is a closed-shell Slater determinant.

## 5.3 Closed-shell parton states

Consider a parton-like state $\Psi_p(Z,\bar{Z})$. When its Slater determinant components are constructed out of single-particle orbitals $\phi_{\alpha_i}(i)$ of $\nu_\mu$ LLs (maximum degree of $\bar{z}_i$ is $\nu_\mu-1$), it is easy to verify that the number of single-particle orbitals for $\nu_\mu$ LLs, with distinct $L_\mu = L(N_L = \nu_\mu)$ defining the highest available angular momenta for each Slater determinant, is given by

$$N_{shell} = \nu_\mu L_\mu - \frac{\nu_\mu(\nu_\mu-1)}{2}. \tag{143}$$

We define a Slater determinant to be **closed-shell** whenever $N=N_{shell}$. Equivalently, a closed-shell Slater determinant is obtained when all the orbitals with certain angular momentum and less are filled, which results in a unique and densest possible configuration (Fig. 14).

We will define a **parton state** to be given by

$$\Phi_\nu(Z,\bar{Z}) = \prod_{\mu=1}^{M} \chi_{\nu_\mu}(Z,\bar{Z}), \tag{144}$$

when all Slater determinants $\chi_{\nu_\mu}$ are **closed-shell**. That is, the parton states are a special subset of all possible parton-like states (Eq. (134)).

One can associate with any such parton state, a string $[\nu_1, \nu_2, \cdots, \nu_M]$ of positive integers $\nu_\mu$, $\mu=1,2,\cdots,M$, such that $\nu_1 \le \nu_2 \le \cdots \le \nu_M$, and show that [8] $\Phi_\nu(Z,\bar{Z})$ represents a state of filling fraction $\nu = (\sum_{\mu=1}^{M} \nu_\mu^{-1})^{-1}$. Restricting the single-particle orbitals to be confined to the subspace generated by the $N_L$ LLs imposes the constraint

$$\sum_{\mu=1}^{M} \nu_\mu = N_L + M - 1. \tag{145}$$

A natural question concerns the possible filling fractions $\nu$ compatible with this constraint. To answer this question, we write down the generating function of partitions of the integer $N_L + M - 1$ into $M$ elements

$$\prod_{t=1}^{\infty} \frac{1}{1-u^t v w^{t-1}} = \sum_{t_1,t_2} u^{t_1} v^{t_2} \sum_\nu w^{\nu^{-1}}, \tag{146}$$

from which one can extract all possible $\nu$'s by inspection of the coefficient of the term with $t_1 = N_L + M - 1$, and $t_2 = M$. For $M=3$, up to $N_L = 6$, we obtain

$$u^3 v^3 (w^3),$$
$$u^4 v^3 (w^{5/2}),$$
$$u^5 v^3 (w^2 + w^{7/3}),$$
$$u^6 v^3 (w^{3/2} + w^{11/6} + w^{9/4}),$$
$$u^7 v^3 (w^{4/3} + w^{5/3} + w^{7/4} + w^{11/5}),$$
$$u^8 v^3 (w^{7/6} + w^{5/4} + w^{19/12} + w^{17/10} + w^{13/6}). \tag{147}$$

For instance, scanning the third line in Eq. (147), we see that when $N_L = 3$, one could get parton states of filling fractions $\nu = 1/2$ and $\nu = 3/7$.

Both the smallest, $\nu_{\min}$, and the largest, $\nu_{\max}$, possible values of $\nu$ carry a special physical meaning. The minimum corresponds to the Jain sequence $\nu_{\min} = \frac{N_L}{2N_L+1}$. The maximum, on the other hand, plays a role in the determination of incompressible (highest density) zero modes of TK type frustration-free QH Hamiltonians, Eq. (25). For fixed $N_L$ and $M$, we next obtain the largest possible filling fraction $\nu_{\max}$ in a systematic manner.

The filling fraction $\nu_{\max}$ can be computed by maximizing $\nu$ (over integers) subject to the constraint of Eq. (145). This *integer optimization* procedure leads to the following condition

$$\begin{cases} \nu_\mu(\nu_\mu + 1) = \nu_{\mu'}(\nu_{\mu'} + 1), \text{ or} \\ \nu_\mu(\nu_\mu + 1) = \nu_{\mu'}(\nu_{\mu'} - 1), \end{cases} \tag{148}$$

for all pairs $\mu, \mu' = 1, 2, \ldots, M$. This associates the *unique* ordered string

$$[\underbrace{\nu_1, \ldots, \nu_1}_{M-n_\nu}, \underbrace{\nu_1 + 1, \cdots, \nu_1 + 1}_{n_\nu}] \tag{149}$$

to $\nu_{\max}$, with $M \nu_1 = M + N_L - 1 - n_\nu$. This results in

$$\nu_{\max} = \frac{\nu_1(\nu_1 + 1)}{2M \nu_1 - N_L + 1}. \tag{150}$$

Table 5 displays various examples of parton states $[\nu_1, \nu_2, \cdots, \nu_M]$ corresponding to maximun filling fraction. It is clear, from Eq. (149), that a unique string of numbers $[\nu_1, \nu_2, \cdots, \nu_M]$ is associated to a maximum filling-fraction parton state. However, this does not imply that there is a unique parton state associated with this unique string. Indeed, there are, in general, several parton-like states (with different total angular momentum) that are associated with a given ordered string. We next study the conditions for the existence of a unique parton-like state.

Since each closed-shell Slater determinant $\chi_{\nu_\mu}(Z, \bar{Z})$ is an eigenstate of total angular momentum

$$\hat{J}\chi_{\nu_\mu}(Z, \bar{Z}) = J_\mu \chi_{\nu_\mu}(Z, \bar{Z}), \tag{151}$$

with

$$\begin{aligned} J_\mu &= \frac{\hbar}{6}(\nu_\mu + 3L_\mu \nu_\mu + 3L_\mu^2 \nu_\mu - 3\nu_\mu^2 - 6L_\mu \nu_\mu^2 + 2\nu_\mu^3) \\ &= \frac{\hbar}{24}\left(\frac{12N^2}{\nu_\mu} - (12N - 1)\nu_\mu - \nu_\mu^3\right), \end{aligned} \tag{152}$$

this implies that parton states are also eigenstates of total angular momentum

$$\hat{J}\Phi_\nu(Z, \bar{Z}) = J_{\min} \Phi_\nu(Z, \bar{Z}), \tag{153}$$

Table 5: Parton states $\Phi_\nu(Z,\bar{Z}) = [\nu_1, \nu_2, \ldots, \nu_M] = \prod_{\mu=1}^{M} \chi_{\nu_\mu}(Z,\bar{Z})$ corresponding to maximum filling fraction $\nu_{\max}$, given $M$ and $N_L$.

| $M$ | $N_L$ | $[\nu_1, \nu_2, \ldots, \nu_M]$ | $\nu_{\max}$ |
|---|---|---|---|
| 3 | 1 | $[1,1,1]$ | 1/3 |
| 3 | 2 | $[1,1,2]$ | 2/5 |
| 3 | 3 | $[1,2,2]$ | 1/2 |
| 3 | 4 | $[2,2,2]$ | 2/3 |
| 3 | 5 | $[2,2,3]$ | 3/4 |
| 3 | 6 | $[2,3,3]$ | 6/7 |
| 3 | 7 | $[3,3,3]$ | 1 |
| 3 | 8 | $[3,3,4]$ | 12/11 |
| 3 | 9 | $[3,4,4]$ | 6/5 |
| 5 | 1 | $[1,1,1,1,1]$ | 1/5 |
| 5 | 2 | $[1,1,1,1,2]$ | 2/9 |
| 5 | 3 | $[1,1,1,2,2]$ | 1/4 |
| 5 | 4 | $[1,1,2,2,2]$ | 2/7 |
| 5 | 5 | $[1,2,2,2,2]$ | 1/3 |
| 5 | 6 | $[2,2,2,2,2]$ | 2/5 |
| 5 | 7 | $[2,2,2,2,3]$ | 3/7 |
| 5 | 8 | $[2,2,2,3,3]$ | 6/13 |
| 5 | 9 | $[2,2,3,3,3]$ | 1/2 |
| 7 | 1 | $[1,1,1,1,1,1,1]$ | 1/7 |
| 7 | 2 | $[1,1,1,1,1,1,2]$ | 2/13 |
| 7 | 3 | $[1,1,1,1,1,2,2]$ | 1/6 |
| 7 | 4 | $[1,1,1,1,2,2,2]$ | 2/11 |
| 7 | 5 | $[1,1,1,2,2,2,2]$ | 1/5 |
| 7 | 6 | $[1,1,2,2,2,2,2]$ | 2/9 |
| 7 | 7 | $[1,2,2,2,2,2,2]$ | 1/4 |
| 7 | 8 | $[2,2,2,2,2,2,2]$ | 2/7 |
| 7 | 9 | $[2,2,2,2,2,2,3]$ | 3/10 |

with eigenvalue $J_{\min}$

$$\frac{J_{\min}}{\hbar} = \frac{N^2}{2\nu} - \frac{\sum_{\mu=1}^{M} \nu_\mu^3 + (12N-1)(N_L + M - 1)}{24}. \tag{154}$$

For a fixed filling fraction $\nu$, it is possible to have several parton states with different values of $J$. This is also true for the minimum total angular momentum $J = J_{\min}$. As we showed previously, the constraint that makes the parton-like state with $\nu = \nu_{\max}$ and $J_{\min}$ to be unique, is the closed-shell condition. In conclusion, a closed-shell parton state projected onto $N_L$ LLs is the unique and densest (in the sense of angular momentum) possible parton-like state. As will be demonstrated in the next section, the *unique* closed-shell parton state with $\nu = \nu_{\max}$ and $J_{\min}$ will be related to our *densest zero mode* of previous sections.

## 5.4 Parton-like states as a basis

The set of Slater determinants forms a basis for the entire antisymmetric Hilbert subspace $\hat{A}\mathcal{H}_{N,J,n}$. In other words, any polynomial in $\hat{A}\mathcal{H}_{N,J,n}$ can be written as a linear superposition of Slater determinants. Given a single-particle orbital basis $\mathcal{B} = \{\phi_{\alpha_i}(z,\bar{z})\}$ with $0 \leq n_i \leq N_L - 1$ and $0 \leq s_i \leq s_{\max}$, the total number of orbitals is $N_{\text{orb}} = N_L(s_{\max}+1) \geq N$. Then, the dimension

of the Hilbert subspace $\hat{\mathcal{A}}\mathcal{H}_{N,J,n}$ is given by $d_{\mathcal{H}} = \binom{N_{\text{orb}}}{N}$. Any polynomial $P(Z,\bar{Z}) \in \hat{\mathcal{A}}\mathcal{H}_{N,J,n}$ can be written as

$$P(Z,\bar{Z}) = \sum_{\mu=1}^{d_{\mathcal{H}}} c_\mu \chi_{P_\mu}(Z,\bar{Z}). \tag{155}$$

Obviously, this expansion also applies for the subspaces $\hat{\mathcal{A}}\mathcal{H}_{N,J,n,M}$, whose dimension $d_{\mathcal{H}_M} < d_{\mathcal{H}}$, but it does not apply for the symmetric subspaces $\hat{\mathcal{S}}\mathcal{H}_{N,J,n,M}$. Then, given a polynomial with the $M$-clustering property, it seems reasonable (and resource efficient) to look for an expansion in terms of elements of $\mathcal{H}_{N,J,n,M}$.

*Do parton-like states form a basis for the symmetric and antisymmetric polynomials with the $M$-clustering property?* Consider the simple case of $N = 2$ particles,

$$P^{(M)}(Z,\bar{Z}) = \sum_{q=0}^{M} z_{12}^q \bar{z}_{12}^{M-q} P_q(Z,\bar{Z}). \tag{156}$$

Since the Slater determinants $\chi_{P_\mu}(Z,\bar{Z})$ form a complete basis of $\hat{\mathcal{A}}\mathcal{H}_{2,J,n}$

$$z_{12} P_q(Z,\bar{Z}) = \sum_\mu c_\mu \chi_{P_\mu}(Z,\bar{Z}), \tag{157}$$

$$\bar{z}_{12} P_q(Z,\bar{Z}) = \sum_\mu \tilde{c}_\mu \chi_{P_\mu}(Z,\bar{Z}). \tag{158}$$

It thus follows that

$$\begin{aligned} P^{(M)}(Z,\bar{Z}) &= \sum_\mu \Big( \sum_{q=1}^{M} c_\mu \chi_{P_\mu} z_{12}^{q-1} \bar{z}_{12}^{M-q} + \tilde{c}_\mu \chi_{P_\mu} \bar{z}_{12}^{M-1} \Big) \\ &= \sum_\mu d_\mu \Psi_{P_\mu}(Z,\bar{Z}) \end{aligned} \tag{159}$$

can be expanded in terms of parton-like states $\Psi_{P_\mu}(Z,\bar{Z})$.

This simple line of reasoning cannot be straightforwardly generalized to $N > 2$. For polynomials depending only on variables (holomorphic coordinates) $Z$ ($N_L = 1$), one can use an alternative proof: consider the simple case of polynomials with the $M$-clustering property depending only on variables $Z$. It is a well-known result from commutative algebra, that the ring of multivariate polynomials over the complex field is a unique factorization domain (UFD), or factorial [77]. In our case, this implies the factorization

$$P^{(M)}(Z) = S(Z)\,\chi_1(Z)^M\,, \tag{160}$$

with $S(Z)$ a totally symmetric polynomial under the exchange of arbitrary indices i and j, and

$$\chi_1(Z) = \prod_{i<j}(z_i - z_j) \tag{161}$$

a Vandermonde determinant, i.e., a totally antisymmetric polynomial under the exchange of arbitrary indices i and j. This factorization is valid for $M$ even or odd (i.e., bosons or fermions, respectively). Expanding the totally antisymmetric polynomial

$$S(Z)\,\chi_1(Z) = \sum_\mu c_\mu \chi_{P_\mu}(Z), \tag{162}$$

in terms of Slater determinants $\chi_{P_\mu}(Z)$, one obtains

$$P^{(M)}(Z) = \sum_\mu c_\mu \Psi_{P_\mu}(Z), \tag{163}$$

with parton-like states

$$\Psi_{p_\mu}(Z) = \chi_{p_\mu}(Z)\chi_1(Z)^{M-1}. \tag{164}$$

The proof of the expansion in Eq. (159) for any $N > 2$ and arbitrary $N_L$ is beyond the scope of this paper. If one conjectured that *all* elements of the space of polynomials with $M$-clustering exponent, $P^{(M)}(Z,\bar{Z}) \in \mathcal{H}_{N,J,n,M}$, can always be written as

$$P^{(M)}(Z,\bar{Z}) = \sum_\mu c_\mu P_\mu^{(M-1)}(Z,\bar{Z})\tilde{P}_\mu^{(1)}(Z,\bar{Z}), \tag{165}$$

then, it is straightforward to show by induction that those same elements can always be written as linear superpositions of parton-like states, i.e., Eq. (159). We will further elaborate on the completeness of parton-like states in the $M$-clustering subspace in the following section(s).

## 5.5  Generating algebras of polynomials $P(Z,\bar{Z})$

We are interested in determining a generating algebra of the elements of $\bigoplus_J \hat{\mathcal{A}}\mathcal{H}_{N,J,n,M}$. Concretely, we mean by that the idea of understanding $\bigoplus_J \hat{\mathcal{A}}\mathcal{H}_{N,J,n,M}$ as a cyclic module of some symmetry algebra. Here, a cyclic module is a representation that is generated by one particular element (a "vacuum") via the action of the algebra in question. Since, for given $N$, $n$, $M$, $\bigoplus_J \hat{\mathcal{A}}\mathcal{H}_{N,J,n,M}$ is the zero-mode space of an associated frustration-free TK Hamiltonian (25), we can think of the algebras in question as symmetry algebras preserving the ground state subspace of this Hamiltonian. The goal of this section is thus to define algebras of operators acting on polynomials that are as rich as possible while preserving the number of LLs $N_L = n + 1$ as well as the (anti-)symmetry and the $M$-clustering property of these polynomials. At first, we will let $n \to \infty$, so as to remove the restriction on the number of LLs. We will subsequently identify sub-algebras that preserve a given maximum $n$.

Define the following symmetric linear operators,

$$A_{\boldsymbol{\alpha}}^1 = \hat{\mathcal{S}}\big[\varphi_{\boldsymbol{\alpha}}(Z,\bar{Z})\partial_{\bar{z}_N}\big], \quad A_{\boldsymbol{\alpha}}^{-1} = \hat{\mathcal{S}}\big[\varphi_{\boldsymbol{\alpha}}(Z,\bar{Z})\partial_{z_N}\big], \quad A_{\boldsymbol{\alpha}}^0 = \hat{\mathcal{S}}\big[\varphi_{\boldsymbol{\alpha}}(Z,\bar{Z})\big], \tag{166}$$

where $\hat{\mathcal{S}}$ is the symmetrizer with respect to variable indices $i = 1, \cdots, N$, and

$$\varphi_{\boldsymbol{\alpha}}(Z,\bar{Z}) \equiv \prod_{i=1}^N \phi_{\alpha_i}(i). \tag{167}$$

These operators form a Lie algebra,

$$[A_{\boldsymbol{\alpha}}^\varepsilon, A_{\boldsymbol{\alpha}'}^{\varepsilon'}] = \sum_{\boldsymbol{\beta}} C_{\boldsymbol{\alpha}\boldsymbol{\alpha}'}^{\boldsymbol{\beta}} A_{\boldsymbol{\beta}}^\varepsilon + \sum_{\boldsymbol{\beta}'} C_{\boldsymbol{\alpha}\boldsymbol{\alpha}'}^{\boldsymbol{\beta}'} A_{\boldsymbol{\beta}'}^{\varepsilon'}, \tag{168}$$

where $C_{\boldsymbol{\alpha}\boldsymbol{\alpha}'}^{\boldsymbol{\beta}}$ are integers, and $\varepsilon, \varepsilon' = 0, \pm 1$. They satisfy

$$\big[\hat{J}, A_{\boldsymbol{\alpha}}^\varepsilon\big] = (J_{\boldsymbol{\alpha}} + \hbar\varepsilon)A_{\boldsymbol{\alpha}}^\varepsilon. \tag{169}$$

The action of these symmetric operators on elements of $\hat{\mathcal{A}}\mathcal{H}_{N,J,n,M}$, preserves their (anti-)symmetry. As for the invariance of the $M$-clustering property, it is evident that the action of the symmetric operator $A_{\boldsymbol{\alpha}}^0$ on any polynomial does not change that property since its action is multiplicative. It remains to analyze the action of $A_{\boldsymbol{\alpha}}^{\pm 1}$. Without loss of generality, we single out a pair $(i, j)$ of indices and write the action of $A_{\boldsymbol{\alpha}}^{-1}$ on $P(Z,\bar{Z})$ as

$$A_{\boldsymbol{\alpha}}^{-1} P(Z,\bar{Z}) = \sum_q (A_{\boldsymbol{\alpha}}^{-1} z_{ij}^q \bar{z}_{ij}^{M-q})P_q(Z,\bar{Z}) + z_{ij}^q \bar{z}_{ij}^{M-q} A_{\boldsymbol{\alpha}}^{-1} P_q(Z,\bar{Z}). \tag{170}$$

The last term in (170) preserves the $M$-clustering property in the pair (i, j). Our last task is thus to show that $A_{\boldsymbol{\alpha}}^{-1} z_{ij}^q \bar{z}_{ij}^{M-q}$ also preserves the $M$-clustering property in (i, j), since we know the expression (170) to be totally (anti-)symmetric, so the pair (i, j) is arbitrary. We may rewrite $A_{\boldsymbol{\alpha}}^{-1} = \hat{\mathcal{S}}\big[S_i(Z, \bar{Z}) \phi_{\alpha_N}(i) \partial_{z_i}\big] = \hat{\mathcal{S}}\big[S_j(Z, \bar{Z}) \phi_{\alpha_N}(j) \partial_{z_j}\big]$, where $S_{i(j)}(Z, \bar{Z})$ is a symmetric polynomial of $N-1$ variables that does not contain particle index i(j) and orbital $\alpha_N$. As a result,

$$A_{\boldsymbol{\alpha}}^{-1} z_{ij}^q \bar{z}_{ij}^{M-q} = \big[S_i(Z, \bar{Z}) \phi_{\alpha_N}(i) - S_j(Z, \bar{Z}) \phi_{\alpha_N}(j)\big] q z_{ij}^{q-1} \bar{z}_{ij}^{M-q} . \tag{171}$$

It is clear that in this expression, the bracketed term is antisymmetric with respect to exchanging i and j. The latter antisymmetry restores the overall clustering exponent $M$ in the expression. This concludes the proof that $A_{\boldsymbol{\alpha}}^{-1} z_{ij}^q \bar{z}_{ij}^{M-q}$ preserves the $M$-clustering property. Finally, since the steps above generalize to $\varepsilon = \pm 1$, if $P(Z, \bar{Z}) \in \hat{\mathcal{A}}\mathcal{H}_{N,J,n,M}$ one gets

$$A_{\boldsymbol{\alpha}}^{\varepsilon} P(Z, \bar{Z}) = \tilde{P}(Z, \bar{Z}) \in \hat{\mathcal{A}}\mathcal{H}_{N,J+J_{\boldsymbol{\alpha}}+\hbar\varepsilon,\tilde{n},M} , \tag{172}$$

where $\tilde{n} = n - \varepsilon(\varepsilon + 1)/2 + \max(n_i)$. Moreover, it is easy to see that the action of $A_{\boldsymbol{\alpha}}^{\varepsilon}$ on any parton-like state results in a linear superposition of parton-like states,

$$A_{\boldsymbol{\alpha}}^{\varepsilon} \Psi_p(Z, \bar{Z}) = \sum_{\mu} d_{\mu} \Psi_{p_{\mu}}(Z, \bar{Z}) . \tag{173}$$

Finally, all of the above clearly carries over in straightforward ways to any element of the algebra generated by the $A_{\boldsymbol{\alpha}}^{\varepsilon}$.

The Lie algebra (168) has several interesting sub-algebras that are noteworthy for their preservation of a maximum number of LLs $N_L$. Their action is graphically depicted in Table 6, with definitions given as follows:

- **Affine Kac-Moody algebra**. For $m > 0$, and $n = N_L - 1$ non-negative integers, the generators

$$S_m^+ = \sum_i z_i^{m+1}(n\bar{z}_i - \bar{z}_i^2 \partial_{\bar{z}_i}) ,$$

$$S_m^- = \sum_i z_i^{m-1} \partial_{\bar{z}_i} ,$$

$$S_m^z = \sum_i z_i^m(\bar{z}_i \partial_{\bar{z}_i} - n/2) \tag{174}$$

define an untwisted affine Kac-Moody [78] algebra

$$[S_m^+, S_{m'}^-] = 2 S_{m+m'}^z , \qquad [S_m^z, S_{m'}^{\pm}] = \pm S_{m+m'}^{\pm} . \tag{175}$$

The action of these operators on $P(Z, \bar{Z})$ changes its total angular momentum via

$$[\hat{J}, S_m^{\pm,z}] = m\hbar S_m^{\pm,z} . \tag{176}$$

- **su(2) algebras**. For given $s_{\max}$ and $n = N_L - 1$, we define generators of three independent su(2) algebras:

$$S^+ = \sum_i z_i \bar{z}_i(n - \bar{z}_i \partial_{\bar{z}_i}) , \quad S^- = \sum_i z_i^{-1} \partial_{\bar{z}_i} ,$$

$$S^z = \sum_i (\bar{z}_i \partial_{\bar{z}_i} - n/2) , \tag{177}$$

Table 6: Action of sub-algebra generators. Arrows represent direction of change in a $(J, \bar{L}_z)$ plane, where right direction corresponds to increasing angular momentum while up refers to increasing LL index.

| $S_m^+$ | $S_m^-$ | $S^+$ | $S^-$ | $L_-$ | $L_+$ | $\bar{L}_-$ | $\bar{L}_+$ |
|---|---|---|---|---|---|---|---|
| $\nearrow^m$ | $\searrow_m$ | $\uparrow$ | $\downarrow$ | $\leftarrow$ | $\rightarrow$ | $\searrow$ | $\nwarrow$ |

where $[S^+, S^-] = 2S^z$, $[S^z, S^\pm] = \pm S^\pm$,

$$L_+ = \sum_i (s_{\max} z_i - z_i^2 \partial_{z_i}), \quad L_- = \sum_i \partial_{z_i},$$

$$L_z = \sum_i (z_i \partial_{z_i} - s_{\max}/2), \tag{178}$$

such that $[L_+, L_-] = 2L_z$, $[L_z, L_\pm] = \pm L_\pm$, and

$$\bar{L}_+ = \sum_i (n \bar{z}_i - \bar{z}_i^2 \partial_{\bar{z}_i}), \quad \bar{L}_- = \sum_i \partial_{\bar{z}_i},$$

$$\bar{L}_z = \sum_i (\bar{z}_i \partial_{\bar{z}_i} - n/2), \tag{179}$$

satisfying $[\bar{L}_+, \bar{L}_-] = 2\bar{L}_z$, $[\bar{L}_z, \bar{L}_\pm] = \pm \bar{L}_\pm$. One may verify that

$$[\hat{J}, S^\pm] = 0, \qquad [\hat{J}, L_\pm] = \pm \hbar L_\pm, \qquad [\hat{J}, \bar{L}_\pm] = \mp \hbar \bar{L}_\pm. \tag{180}$$

We point out that the algebra defined in Eq. (177) is the first quantization representation of the pseudospin algebra in Eq. (54), which is well-defined only when $j_i \geq 0$ (away from the boundary).

Consider now polynomials $P(Z, \bar{Z}) \in \bigoplus_J \hat{A} \mathcal{H}_{N,J,n,M}$ with well-defined angular momentum $J$ (parton-like states are examples of such polynomials). What is(are) the $P_{\min}(Z, \bar{Z})$ with lowest total angular momentum, i.e., $\hat{J} P_{\min}(Z, \bar{Z}) = J_{\min} P_{\min}(Z, \bar{Z})$? We will approach this question first by defining a *highest weight state(s)* of the algebra generated by the $A_\alpha^\varepsilon$ to be a polynomial $P_{hw}(Z, \bar{Z})$ satisfying

$$A_\alpha^\varepsilon P_{hw}(Z, \bar{Z}) = 0, \tag{181}$$

whenever $J_\alpha + \hbar \varepsilon < 0$ and $\tilde{n} \leq n$. For instance, for $n = 0$, the polynomial $\chi_1(Z)^M$ (Laughlin states) is a highest weight state of the algebra. Clearly, any $P_{\min}(Z, \bar{Z})$ must also be a heighest weight state, otherwise the condition of minimal angular momentum would be violated. By the same token, any minimum angular momentum parton-like state is a heighest weight state, as the action of the algebra preserves the parton-like character. We claim that for arbitrary $n$ the highest weight states of the algebra are parton-like states. If such a parton-like state satisfies the condition of being closed-shell, then, according to the claim, it must be the *unique* minimum angular momentum highest weight state. This is so since by Section 5.3, such a closed-shell parton state is the unique parton-like state with lowest total angular momentum.

Here we give a heuristic justification for the above claim. We wish to argue that the general symmetric operators given in Eq. (166) are the "generators of all polynomials with $M$-clustering exponent" in the following sense. Consider the sub-algebra that preserves $N_L$ LLs. The claim follows if it can be argued that this algebra is rich enough such that $\bigoplus_J \hat{A} \mathcal{H}_{N,J,n,M}$ is irreducible as a representation of this algebra. As is well known in the representation theory of algebras, every irreducible representation is cyclic, where every single state can serve to generate the whole representation: Otherwise, the cyclic module generated by such a state would be a proper invariant sub-module, contradicting irreducibility. Thus, then, every state

in $\bigoplus_J \hat{A}\mathcal{H}_{N,J,n,M}$ can be reached from any other via actions of the algebra. Moreover, since $\bigoplus_J \hat{A}\mathcal{H}_{N,J,n,M}$ contains parton-like states, and the action of the algebra preserves the property of being a parton-like superposition, any element of $\bigoplus_J \hat{A}\mathcal{H}_{N,J,n,M}$ must be a parton-like superposition. All the above conjectures then follow from properties of parton-like states, in particular the uniqueness of partons as minimum angular momentum parton-like states. While we find it plausible that the algebra defined here is rich enough in the precise sense defined above, we leave the proof of this as an interesting mathematical problem.

As a useful application, we note the following corollary: The MR Pfaffian and RR states cannot be densest–incompressible–ground states of the two-body frustration-free parent Hamiltonians of this work. The reason is that these states are not closed-shell parton states but are expanded in terms of parton-like states. For instance, consider the MR Pfaffian state

$$\Psi_{\text{MR}}(Z) = \text{Pf}\left(\frac{1}{z_i - z_j}\right) \chi_1(Z)^{M+1}. \tag{182}$$

One can check that for $N = 4$ it can be expanded as

$$\Psi_{\text{MR}}(Z) = \Psi_1(Z) - 2\Psi_2(Z) + 10\Psi_3(Z), \tag{183}$$

where the parton-like states are

$$\Psi_1(Z) = \begin{vmatrix} 1 & 1 & 1 & 1 \\ z_1 & z_2 & z_3 & z_4 \\ z_1^4 & z_2^4 & z_3^4 & z_4^4 \\ z_1^5 & z_2^5 & z_3^5 & z_4^5 \end{vmatrix} \chi_1(Z)^{M-1},$$

$$\Psi_2(Z) = \begin{vmatrix} 1 & 1 & 1 & 1 \\ z_1^2 & z_2^2 & z_3^2 & z_4^2 \\ z_1^3 & z_2^3 & z_3^3 & z_4^3 \\ z_1^5 & z_2^5 & z_3^5 & z_4^5 \end{vmatrix} \chi_1(Z)^{M-1},$$

$$\Psi_3(Z) = \begin{vmatrix} z_1 & z_2 & z_3 & z_4 \\ z_1^2 & z_2^2 & z_3^2 & z_4^2 \\ z_1^3 & z_2^3 & z_3^3 & z_4^3 \\ z_1^4 & z_2^4 & z_3^4 & z_4^4 \end{vmatrix} \chi_1(Z)^{M-1}. \tag{184}$$

Similarly, for the ($N = 4$) Read-Rezayi state one obtains $\Psi_{\text{RR}}(Z) = 2\Psi_{\text{MR}}(Z)$.

## 6 Partons, DNA, and MPS states

When we determined, in the preceding sections, the ground subspace of the general two-body frustration-free Hamiltonians of the type of Eq. (25), we discussed two seemingly distinct threads. These centered on the EPP on the one hand and the parton construction on the other. The emergent EPP establishes constraints on any pair of particles in the DNA or root pattern of the ground state. Thus far, we have, however, refrained from demonstrating that any ground states satisfying the EPP indeed qualify as ground states of our frustration-free Hamiltonian. The closed-shell parton states $\Phi_\nu(Z, \bar{Z})$ represent the densest ground states. Nevertheless, on its own, this property does not yield the complete set of ground states of Hamiltonians given by Eq. (25). By combining the rules set by the EPP and parton constructs, one can provide a rigorous method to establish completeness of parton-like states to span $\bigoplus_J \hat{A}\mathcal{H}_{N,J,n,M}$. In this Section, we will establish completeness for the special case of $M = 3, n = N_L - 1 = 3$. Prior to doing so, we will start our discussion by constructing root patterns and root states (DNA)

from a given parton-like state. We will firmly connect these to the EPPs. We will next show the MPS structure of the DNA illustrating the complex and interesting pattern of entanglement that encodes those non-Abelian fluids. We will then derive a scheme that will enable us to extract possible parton-like states given a root pattern.

## 6.1 Root pattern and DNA of parton-like states

Consider a single Slater determinant $\chi_p(Z,\bar{Z})$ with a root pattern $\{j\}_{\text{root}} = \{j_1, j_2, \ldots, j_N\}_{\text{root}}$ that is arranged in an ascending order of angular momenta $j_1 \leq j_2 \leq \cdots \leq j_N$, where $j_i$ is the angular momentum of particle i. It is clear that this root pattern is extracted from the monomial $\varphi_{\boldsymbol{\alpha}}(Z,\bar{Z})$, where $\chi_p(Z,\bar{Z}) = \hat{\mathcal{A}}\big[\varphi_{\boldsymbol{\alpha}}(Z,\bar{Z})\big]$. In the LLL, due to the Pauli exclusion principle, the monomial $\varphi_{\boldsymbol{\alpha}}(Z,\bar{Z})$ is unique. When multiple LLs are present, several orbitals may share the same angular momenta. This allows for many monomials $\varphi_{\boldsymbol{\alpha}}(Z,\bar{Z})$ satisfying the rule of ascending $j_i$. Among all $N!$ monomials comprising a Slater determinant, the number of distinct monomials $\varphi_{\boldsymbol{\alpha}}(Z,\bar{Z})$ satisfying this rule is $\mathcal{M}_p = \prod_{j=1}^{L} \lambda_j!$, where $\lambda_j$ is the multiplicity of angular momentum $j$. For example, consider $N = 3$ with $j_1 = j_2 < j_3$. The corresponding distinct monomials ($\mathcal{M}_p = 2$) would be

$$\varphi_{\boldsymbol{\alpha}}(Z,\bar{Z}) = \phi_{\alpha_1}(1)\phi_{\alpha_2}(2)\phi_{\alpha_3}(3),$$
$$\varphi_{\sigma\boldsymbol{\alpha}}(Z,\bar{Z}) = \phi_{\alpha_2}(1)\phi_{\alpha_1}(2)\phi_{\alpha_3}(3), \tag{185}$$

where $\sigma \in S_N$ is a permutation of the $\alpha_i$ indices. The root pattern of the product $\chi_p(Z,\bar{Z})\chi_{p'}(Z,\bar{Z})$ is [28, 79]

$$\{j_1 + j_1', \cdots, j_N + j_N'\}_{\text{root}} \equiv \{j\}_{\text{root}} + \{j'\}_{\text{root}}, \tag{186}$$

which is associated with the $\mathcal{M}_p \mathcal{M}_{p'}$ monomials

$$\varphi_{\boldsymbol{\alpha}}(Z,\bar{Z})\varphi_{\boldsymbol{\alpha}'}(Z,\bar{Z}) = \varphi_{\boldsymbol{\alpha}+\boldsymbol{\alpha}'}(Z,\bar{Z}). \tag{187}$$

We stress that, although $\mathcal{M}_{p_\mu}$ is the number of distinct monomials in each individual Slater $\chi_{p_\mu}$, not all of the monomials $\mathcal{M}_p \mathcal{M}_{p'}$ may be distinct. This implies that $\mathcal{M}_p \mathcal{M}_{p'}$ constitutes an upper bound on the number of distinct monomials in the root state of $\chi_p(Z,\bar{Z})\chi_{p'}(Z,\bar{Z})$. Now, in an arbitrary parton-like state with $M$ Slater determinants, the number of distinct monomials

$$\varphi_{\vec{\boldsymbol{\alpha}}}(Z,\bar{Z}) \equiv \prod_{\mu=1}^{M} \varphi_{\boldsymbol{\alpha}_{p_\mu}}(Z,\bar{Z}) = \varphi_{\sum_{\mu=1}^{M} \boldsymbol{\alpha}_{p_\mu}}(Z,\bar{Z}) \tag{188}$$

is upper bounded by $\prod_{\mu=1}^{M} \mathcal{M}_{p_\mu}$. With $M \in$ (odd)even, the (anti)symmetrization of each monomial provides a non-expandable (Slater determinant)permanent. The corresponding root state is a linear superposition of all such non-expandable (Slater determinants)permanents. Such a superposition encodes a specific pattern of entanglement. In what follows, we focus on the root pattern of parton-like states. We will then study the corresponding root states in the next subsection.

To illustrate our basic premise for the root patterns, we examine a specific example. We will then discuss the generalization to other states. Towards this end, we first consider the particular closed-shell parton state of $N = 7$ particles and $N_L = 3$ (see Fig. 15), $\Phi_{1/2}(Z,\bar{Z}) = \chi_1(Z,\bar{Z})\chi_2(Z,\bar{Z})\chi_2(Z,\bar{Z})$. The root occupation configuration for the two Slater determinants, $\chi_2(Z,\bar{Z})$ and $\chi_1(Z,\bar{Z})$ are, respectively,

$$\{j'\}_{\text{root}} = \{-1, 0, 0, 1, 1, 2, 2\}_{\text{root}},$$



Figure 15: Angular momentum (in units of $\hbar$) occupation configuration of Slater determinants components of the seven particles parton state $\Phi_{1/2}(Z,\bar{Z}) = \chi_1(Z,\bar{Z})\chi_2(Z,\bar{Z})\chi_2(Z,\bar{Z})$.

Table 7: Bulk root patterns, $\{\lambda\}_{\text{broot}}$, for densest closed-shell parton states $\Phi_\nu$ with third order zeroes, $M = 3$, and filling fractions $\nu$.

| $N_L$ | Bulk root pattern | $\nu$ |
|---|---|---|
| 1 | {100} | 1/3 |
| 2 | {10100} | 2/5 |
| 3 | {1100} | 1/2 |
| 4 | {200} | 2/3 |
| 5 | {20020110} | 3/4 |
| 6 | {2002101} | 6/7 |
| 7 | {300} | 1 |
| 8 | {30030120210} | 12/11 |

$$\{j''\}_{\text{root}} = \{0,1,2,3,4,5,6\}_{\text{root}}, \tag{189}$$

so that the final root occupation configuration of the parton is $\{j\}_{\text{root}} = \{j'\}_{\text{root}} + \{j'\}_{\text{root}} + \{j''\}_{\text{root}}$, with

$$\begin{aligned}
j_1 &= -1-1+0 = -2, \\
j_2 &= 0+0+1 = 1, \\
j_3 &= 0+0+2 = 2, \\
j_4 &= 1+1+3 = 5, \\
j_5 &= 1+1+4 = 6, \\
j_6 &= 2+2+5 = 9, \\
j_7 &= 2+2+6 = 10.
\end{aligned} \tag{190}$$

This leads to the following map

| $j$ | −2 | −1 | 0 | 1 | 2 | 3 | 4 | 5 | 6 | 7 | 8 | 9 | 10 |
|---|---|---|---|---|---|---|---|---|---|---|---|---|---|
| $\lambda_j$ | 1 | 0 | 0 | 1 | 1 | 0 | 0 | 1 | 1 | 0 | 0 | 1 | 1 |

with root pattern $\{\lambda\}_{\text{root}} = 1001100110011$. Neglecting boundaries ($j < 0$), in the bulk each four consecutive states are filled by two electrons. This defines the "*bulk* root pattern" $\{\lambda\}_{\text{broot}} = \{1100\}$. The above analysis may be repeated for other states. In Table 7, we show examples of bulk root patterns and their corresponding filling fractions for other closed-shell parton states.

### 6.1.1 Root states or DNAs from a given parton-like state

When confined to the LLL, the root patterns encode all of the important information regarding the parton-like states. However, in the presence of higher LL mixing, there is an entanglement between the patterns in the root states. This entanglement contains important information, such as zero-mode counting, on the parton-like states. In order to extract this information, we need to determine the root states, or DNAs, for the root patterns obtained from the given parton-like states.

As described above, given an $N$ particle parton-like state $\Psi_p(Z, \bar{Z})$ in the subspace of $N_L$ LLs, one can determine its root pattern $\{\lambda\}_{\text{root}}$. Since we are interested in the bulk part of the root pattern, $\{\lambda\}_{\text{broot}}$, consider the $\Psi_p(Z, \bar{Z})$ which consists of single-particle orbitals of non-negative angular momenta, $j_i \geq 0$. Our aim is to keep all of the non-expandable Slater determinants in the expansion of $\Psi_p(Z, \bar{Z})$ with pattern $\{\lambda\}_{\text{root}}$. Thus, we exclude inward-squeezed states.

To that end, consider a simple rescaling of the coordinates $z_i \to \zeta_i z_i$ and $\bar{z}_i \to \zeta_i^{-1} \bar{z}_i$, where $i = 1, \dots, N$. Let us denote the rescaled coordinates by $(Z', \bar{Z}')$. An algebraic algorithmic way of extracting the root state is the following: In the expansion of $\Psi_p(Z', \bar{Z}')$, the monomials with a common factor $\Pi_{i=1}^N \zeta_i^{j_i}$ relate to the non-expandable determinants and are determined by

$$\prod_{i=1}^N \frac{1}{j_i!} (\partial_{\zeta_i})^{j_i} \Psi_p(Z', \bar{Z}') = \Big(\prod_{i=1}^N z_i^{j_i}\Big) f_{N_L, \nu}^{\{\lambda\}_{\text{broot}}} = \sum_{\vec{\alpha}} C_{\vec{\alpha}} \, \varphi_{\vec{\alpha}}(Z, \bar{Z}). \tag{191}$$

Here, $f_{N_L, \nu}^{\{\lambda\}_{\text{broot}}}(Z, \bar{Z})$ has zero total angular momentum $\hat{J} f_{N_L, \nu}^{\{\lambda\}_{\text{broot}}} = 0$, $C_{\vec{\alpha}} = \pm 1$. The sum contains $\prod_{\mu=1}^M \mathcal{M}_{p_\mu}$ terms. As a result, the root state (DNA) can be obtained by performing a simple antisymmetrization,

$$\Psi_{\text{root}}(Z, \bar{Z}) = \hat{\mathcal{A}}\Bigg[ \Big(\prod_{i=1}^N z_i^{j_i}\Big) f_{N_L, \nu}^{\{\lambda\}_{\text{broot}}} \Bigg]. \tag{192}$$

Let us provide a few examples (with $N$ even). The root state for the $\nu = 1/M$ Laughlin state is obtained by $f_{1, \frac{1}{M}}^{\{10^{M-1}\}}(Z, \bar{Z}) = 1$ (representing a single Slater determinant). Next, consider the parton state with $\{10100\}$ ($\nu = 2/5$), $N_L = 2$ and root pattern $\{j\}_{\text{root}} = \{0, 2, 5, 7, \cdots\}_{\text{root}}$. For this state, we obtain

$$f_{2, \frac{2}{5}}^{\{10100\}} = \prod_{r=0}^{\frac{N-2}{2}} D_{\alpha_1 \alpha_2}(2r+1, 2r+2). \tag{193}$$

In this example and throughout this section, $\alpha_1 = (1, 1)$ and $\alpha_2 = (0, 0)$, i.e.,

$$D_{\alpha_1 \alpha_2}(2r+1, 2r+2) \equiv \begin{vmatrix} z_{2r+1} \bar{z}_{2r+1} & z_{2r+2} \bar{z}_{2r+2} \\ 1 & 1 \end{vmatrix}. \tag{194}$$

For the parton state with $\{200\}$ ($\nu = 2/3$), $N_L = 4$, and $\{j\}_{\text{root}} = \{0, 0, 3, 3, \cdots\}_{\text{root}}$, it can be checked that

$$f_{4, \frac{2}{3}}^{\{200\}} = \prod_{r=0}^{\frac{N-2}{2}} D_{\alpha_1 \alpha_2}^3(2r+1, 2r+2). \tag{195}$$

From the above structure, it is evident that the $\{200\}$ pattern is a simple product state of entangled pairs in the root pattern. Applying the pseudospin algebra of Eq. (177),

$$S^z f_{4, \frac{2}{3}}^{\{200\}} = S^{\pm} f_{4, \frac{2}{3}}^{\{200\}} = 0. \tag{196}$$

This suggests that the 2 in the root pattern indeed represents a singlet state.

We have, so far, discussed bulk root patterns for (closed-shell) parton states. For these, there is a one-to-one correspondence between the parton state and its root pattern. However, in general, several parton-like states may share the same root pattern. For instance, for the $\{110\}$ pattern, $N_L = 4$, we have four different root states corresponding to the four different parton-like states given by $\{110\}_{n_N}^{n_1} = 1_{n_1}10110...11011_{n_N}0$. Here, $n_1, n_N$ identify the pseudospin degrees of freedom. The corresponding parton-like states are $\chi_{n_1 n_N}(Z,\bar{Z})\chi_2(Z,\bar{Z})^2$, $n_1, n_N = 0, 1$, with $\chi_{n_1 n_N}(Z,\bar{Z})$ constructed such that the lowest and highest angular momentum orbitals are, respectively, occupied by electrons with LL indices given by $n_1$ and $n_N$. Consequently,

$$f_{4,\frac{2}{3}}^{\{110\}_{n_N}^{n_1}} = (z_1\bar{z}_1)^{n_1}(z_N\bar{z}_N)^{n_N} \prod_{r=0}^{\frac{N-2}{2}} D_{\alpha_1\alpha_2}^2(2r+1, 2r+2) \prod_{r=0}^{[\frac{N-3}{2}]} D_{\alpha_1\alpha_2}(2r+2, 2r+3), \quad (197)$$

where $[x]$ represents the integer part of $x$. As another example, consider the root pattern 20011011 with $N = 6$ particles, which includes a domain wall. We obtain

$$f_{4,\frac{5}{7}}^{\{200\}\{110\}_{n_6}^{n_3}} = (z_3\bar{z}_3)^{n_3}(z_6\bar{z}_6)^{n_6} D_{\alpha_1\alpha_2}^3(1,2) D_{\alpha_1\alpha_2}^2(3,4) D_{\alpha_1\alpha_2}(4,5) D_{\alpha_1\alpha_2}^2(5,6), \quad (198)$$

where $n_3, n_6 = 0, 1$. The choice of even $N$ allowed us to obtain compact expressions for $f_{N_L,\nu}^{\{\lambda\}_{\text{broot}}}$. We could have similarly considered states with odd $N$, starting from Eq. (192), by appropriate modifications of $f_{N_L,\nu}^{\{\lambda\}_{\text{broot}}}$. For instance, consider an $N = 5$ particle state with root pattern 1101011. We obtain the following eight states

$$f_{4,\frac{3}{4}}^{\{110\}^{n_1} 1_{n_3} \{011\}_{n_5}} = (z_1\bar{z}_1)^{n_1}(z_3\bar{z}_3)^{n_3}(z_5\bar{z}_5)^{n_5} D_{\alpha_1\alpha_2}^2(1,2) D_{\alpha_1\alpha_2}(2,3) D_{\alpha_1\alpha_2}(3,4) D_{\alpha_1\alpha_2}^2(4,5), \quad (199)$$

where $n_1, n_3, n_5 = 0, 1$. It turns out that $f_{N_L,\nu}^{\{\lambda\}_{\text{broot}}}$ carries the pseudospin structure of the root state since

$$S^z \Psi_{\text{root}}(Z,\bar{Z}) = \hat{\mathcal{A}}\left[\left(\prod_{i=1}^{N} z_i^{j_i}\right) S^z f_{N_L,\nu}^{\{\lambda\}_{\text{broot}}}\right]. \quad (200)$$

Furthermore, as we will show in the following, it contains the *pattern of entanglement* dictating the EPP rules for the complete zero-mode subspace.

### 6.1.2 MPS representation of DNA from polynomials

The EPP does not rule out entangled root states formed by combining the 11 and 101 patterns. Such root states do not only give rise to a ground state root pattern ...110110... with filling fraction 2/3 but also play an important role in determining the domain wall structures and braiding statistics as discussed before. These states cannot be formed by simple products of 11 and 101 patterns. If all such simple products were feasible then the ground state degeneracy would scale exponentially in the particle number. However, Eq. (197), unlike Eq. (195), indeed does not suggest the appearance of such simple products of entangled motifs in the root states. In Eq. (197), the $2r + 2$ particle is connected with particles $2r + 1$ and $2r + 3$. This construct hints at an underlying MPS structure. This representation may capture the root state entanglement while satisfying the EPP for the 11 and 101 patterns. Towards this end, we consider the following identity

$$(z_1\bar{z}_1)^{n_1}(z_2\bar{z}_2)^{n_2} D_{\alpha_1\alpha_2}^{\mathsf{b}}(1,2) = \widehat{\sum} (z_1\bar{z}_1)^{I_1} \epsilon_{I_2 I_1} (z_2\bar{z}_2)^{I_2}, \quad (201)$$

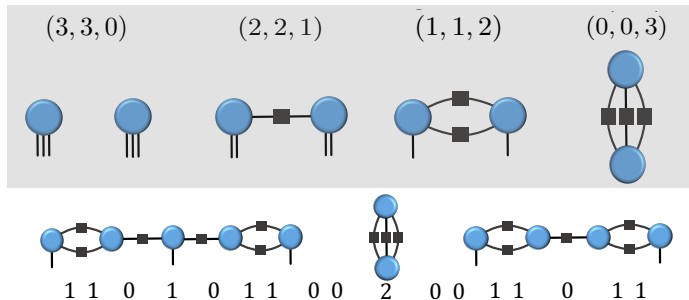

Figure 16: (Top) Diagrammatic representation of the polynomial $(z_i \bar{z}_i)^{n_i} (z_j \bar{z}_j)^{n_j} D^b_{\alpha_1 \alpha_2}(i,j)$ for all possible values of $(n_i, n_j, b)$ where $n_i = n_j \leq N_L - b - 1$, with $N_L = 4$. Each disk represents a particle associated with $N_L - 1$ bonds. Each bond connecting two disks is equivalent to an index contraction and decreases the relative angular momentum between $i^{th}$ and $j^{th}$ particle by 1. The number of bonds in between $i^{th}$ and $j^{th}$ particle is given by $b$ and relative angular momentum is given by, $M - b$, where $M$ is the order of two particle zeros in the ground state wave function. Thus for $M = 3$, $b = 0, 1, 2, 3$ represent the 1001, 101, 11 and 2 patterns in the root states, respectively. The $m$ non-contracted (dangling) bonds associated with each disk gives rise to a $m + 1$ degeneracy and can be represented by a $m/2$ pseudospin algebra. (Bottom) Above patterns glued together to construct an $N = 11$-particle MPS state. Same MPS structure can be obtained from the viewpoint of EPPs as shown in Fig. 4.

where $I_i = I_i^{(1)} + \cdots + I_i^{(N_L - 1)}$, $I_i^{(i)} = 0, 1$, $n_i \leq N_L - b - 1$, $i = 1, 2$, and $\epsilon_{I_2 I_1} \equiv \prod_{i=1}^b \epsilon_{I_2^{(i)} I_1^{(i)}}$, with $\epsilon_{I_2^{(i)} I_1^{(i)}}$ the Levi-Civita tensor ($\epsilon_{01} = +1$). The symbol $\widehat{\sum}$ ($N = 2$ in Eq. (201)) involves the contracted $I_i^{(i)}$ indices

$$\widehat{\sum} \equiv \sum_{\{I_1^{(i)}, I_2^{(i)}, \cdots I_N^{(i)}\}_{i=1,b}}, \tag{202}$$

i.e., sums only over $b$ indices that appear in the Levi-Civita tensor. The sum of free $I_i^{(i)}$, $i > b$, indices is identified with $n_i \leq N_L - b - 1$. Figure 16 (top) shows a diagrammatic representation of Eq. (201). The contraction of indices can be shown as horizontal connections between adjacent disks where each box contains a Levi-Civita symbol. On the other hand, contraction of $b$ leaves $n_i$ free indices. These are shown as dangling bonds in Fig. 16. Accordingly, the number of bonds and dangling bonds can be read directly from a generic $f_{N_L, \nu}^{\{\lambda\}_{\text{broot}}}$ polynomial that renders the corresponding diagram for such polynomial, e.g., see Fig. 16 (bottom).

The polynomial $(z_1 \bar{z}_1)^{n_1} (z_2 \bar{z}_2)^{n_2} D^b_{\alpha_1 \alpha_2}(1,2)$ can be written as an MPS by defining rank-3 tensors

$$M^{I_i}_{I_i^{(1)} I_i^{(2)} I_i^{(3)}} \equiv \delta_{I_i, I_i^{(1)} + I_i^{(2)} + I_i^{(3)}}, \tag{203}$$

so that,

$$(z_1 \bar{z}_1)^{n_1} (z_2 \bar{z}_2)^{n_2} D^b_{\alpha_1 \alpha_2}(1,2) = \widehat{\sum} M^{I_1} *^b M^{I_2} (z_1 \bar{z}_1)^{I_1} (z_2 \bar{z}_2)^{I_2}, \tag{204}$$

where the $*^b$ symbol indicates the number of inserted Levi-Civita symbols, $*^b \equiv \epsilon_{I_2 I_1}$.

The expression above facilitates a simple generalization of the MPS representation for a generic $f_{N_L, \nu}^{\{\lambda\}_{\text{broot}}}$ polynomial. Consider

$$f_{4, \frac{2}{3}}^{\{110\}^{n_1}_{n_4}} = (z_1 \bar{z}_1)^{n_1} (z_4 \bar{z}_4)^{n_4} D^2_{\alpha_1 \alpha_2}(1,2) D_{\alpha_1 \alpha_2}(2,3) D^2_{\alpha_1 \alpha_2}(3,4), \tag{205}$$

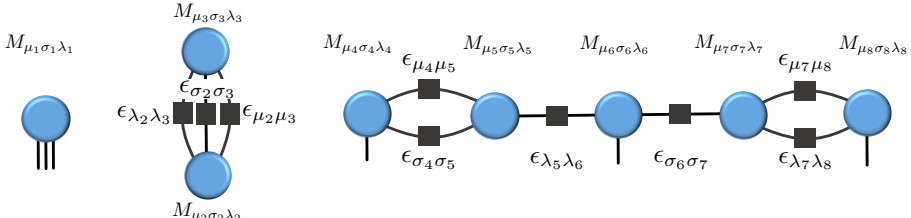

Figure 17: MPS diagrammatic representation of DNA for $N = 8$. We have identified $M^{I_i}_{I_i^{(1)} I_i^{(2)} I_i^{(3)}} = M_{\mu_i \sigma_i \lambda_i}$. Here, the diagram is for $f^{\{\lambda\}_{\text{broot}}}_{N_L, \nu} = (z_1 \bar{z}_1)^{\mu_1 + \sigma_1 + \lambda_1} (z_4 \bar{z}_4)^{\lambda_4} (z_6 \bar{z}_6)^{\mu_6} (z_8 \bar{z}_8)^{\sigma_8} D^3_{\alpha_1 \alpha_2}(2,3) D^2_{\alpha_1 \alpha_2}(4,5) D_{\alpha_1 \alpha_2}(5,6) D_{\alpha_1 \alpha_2}(6,7) D^2_{\alpha_1 \alpha_2}(7,8)$ where $\mu_1 = \sigma_1 = \lambda_1 = \lambda_4 = \mu_6 = \sigma_8 = 1$. The number of bonds can easily be read from the graph: $b_i = (0, 3, 0, 2, 1, 1, 2)$.

for $N = 4$, where

$$D^2_{\alpha_1 \alpha_2}(1,2) = \widehat{\sum} M^{I_1} *^2 M^{I_2} (z_1 \bar{z}_1)^{I_1} (z_2 \bar{z}_2)^{I_2}, \tag{206}$$

$$D_{\alpha_1 \alpha_2}(2,3) = \widehat{\sum} M^{I_2} *^1 M^{I_3} (z_2 \bar{z}_2)^{I_2} (z_3 \bar{z}_3)^{I_3}, \tag{207}$$

$$D^2_{\alpha_1 \alpha_2}(3,4) = \widehat{\sum} M^{I_3} *^2 M^{I_4} (z_3 \bar{z}_3)^{I_3} (z_4 \bar{z}_4)^{I_4}. \tag{208}$$

Combining the above representations one obtains

$$f^{\{110\}^{n_1}_{n_4}}_{4, \frac{2}{3}} = \widehat{\sum} M^{I_1} *^2 M^{I_2} *^1 M^{I_3} *^2 M^{I_4} (z_1 \bar{z}_1)^{I_1} (z_2 \bar{z}_2)^{I_2} (z_3 \bar{z}_3)^{I_3} (z_4 \bar{z}_4)^{I_4}, \tag{209}$$

where $n_1 = I_1^{(3)}$ and $n_4 = I_4^{(3)}$. Similarly, a generic polynomial $f^{\{\lambda\}_{\text{broot}}}_{N_L, \nu}$ can be represented by the MPS

$$f^{\{\lambda\}_{\text{broot}}}_{N_L, \nu} = \widehat{\sum} M^I \times \prod_{i=1}^{N} (z_i \bar{z}_i)^{I_i}, \tag{210}$$

where we have used the abbreviation for the tensor network

$$M^{I_1} *^{b_1} M^{I_2} \cdots M^{I_{N-1}} *^{b_{N-1}} M^{I_N} \equiv M^I. \tag{211}$$

Figure 17, shows and example of a tensor network for $N = 8$. Now, using Eq. (192), we can rewrite the root state as an MPS

$$\Psi_{\text{root}}(Z, \bar{Z}) = \hat{\mathcal{A}} \left[ \left( \prod_{i=1}^{N} z_i^{j_i} \right) f^{\{\lambda\}_{\text{broot}}}_{N_L, \nu} \right] = \widehat{\sum} M^I |I\rangle, \tag{212}$$

where

$$|I\rangle \equiv \hat{\mathcal{A}} \left[ \left( \prod_{i=1}^{N} z_i^{j_i} \right) (z_1 \bar{z}_1)^{I_1} (z_2 \bar{z}_2)^{I_2} \cdots (z_N \bar{z}_N)^{I_N} \right] \tag{213}$$

are the non-expandable Slater determinants in the root state.

## 6.2 Parton construction and the entangled Pauli principle for the $2/3$ state

We now return to the Hamiltonian construct for the assembled EPPs and the two particle selection rules for the root patterns of the ground state for the four LLs projected Hamiltonian. We have so far postponed the task of showing that there are indeed two particle solutions satisfying EPPs constraints. To establish a ground state obeying the EPPs, we start with parton

state $\chi_2(Z,\bar{Z})^3$. The DNA corresponding to this state is given by 100200200200... . This parton state has a {200} pattern in the bulk having two particles in three consecutive "sites", i.e. modes. The filling fraction for this state is 2/3 and our EPP construction already excludes any other parton-like state as being a zero-energy state of higher density. Each "2" in the 100200200.. pattern is entangled as predicted by Eq. (56).

Before examining other admissible patterns in the ground state root, we introduce the two Slater determinants $\chi'_{nn'}(Z,\bar{Z})$ and $\chi''_{nn'}(Z,\bar{Z})$ appearing in Fig. 18. The EPP predicted four possible 11 patterns which can be derived from the four parton states, $\chi'_{nn'}(Z,\bar{Z})\chi_2(Z,\bar{Z})^2$ with $n, n' = 0, 1$. The root patterns of these four 2-particle parton states can be mapped to Eq. (58) in the following way,

$$\tilde{\Psi}_0^{(1)}(Z,\bar{Z}) = \chi'_{00}(Z,\bar{Z})\chi_2(Z,\bar{Z})^2,$$
$$\tilde{\Psi}_0^{(2)}(Z,\bar{Z}) = \chi'_{01}(Z,\bar{Z})\chi_2(Z,\bar{Z})^2,$$
$$\tilde{\Psi}_0^{(3)}(Z,\bar{Z}) = \chi'_{10}(Z,\bar{Z})\chi_2(Z,\bar{Z})^2,$$
$$\tilde{\Psi}_0^{(4)}(Z,\bar{Z}) = \chi'_{11}(Z,\bar{Z})\chi_2(Z,\bar{Z})^2. \tag{214}$$

While parton states naturally realize the zero-energy modes, we must keep in mind that parton states constitute an overcomplete basis. Thus, caution must be exercised. It is possible to double count topologically identical zero modes (i.e., zero modes with same root patterns) in the parton construction. To convey this message more concretely, we will now consider the 101 pattern and their root states. While the EPP suggests that there are only 9 such states, from the parton construction we get 10 states of the form, $\chi'_{n_1 n_2}(Z,\bar{Z})\chi'_{n'_1 n'_2}(Z,\bar{Z})\chi_2(Z,\bar{Z})$ and 4 states of the form $\chi''_{nn'}(Z,\bar{Z})\chi_2(Z,\bar{Z})^2$. All of these 14 states can be constructed from the root pattern of Eq. (60). The zero modes corresponding to the root states of Eq. (60) are

$$\Psi_0^{(1)}(Z,\bar{Z}) = \chi'_{00}(Z,\bar{Z})\chi'_{00}(Z,\bar{Z})\chi_2(Z,\bar{Z}),$$
$$\Psi_0^{(2)}(Z,\bar{Z}) = \chi'_{00}(Z,\bar{Z})\chi'_{01}(Z,\bar{Z})\chi_2(Z,\bar{Z}),$$
$$\Psi_0^{(3)}(Z,\bar{Z}) = \chi'_{00}(Z,\bar{Z})\chi'_{10}(Z,\bar{Z})\chi_2(Z,\bar{Z}),$$
$$\Psi_0^{(4)}(Z,\bar{Z}) = \chi'_{11}(Z,\bar{Z})\chi'_{00}(Z,\bar{Z})\chi_2(Z,\bar{Z}),$$
$$\Psi_0^{(5)}(Z,\bar{Z}) = \chi'_{10}(Z,\bar{Z})\chi'_{10}(Z,\bar{Z})\chi_2(Z,\bar{Z}),$$
$$\Psi_0^{(6)}(Z,\bar{Z}) = \chi'_{11}(Z,\bar{Z})\chi'_{10}(Z,\bar{Z})\chi_2(Z,\bar{Z}),$$
$$\Psi_0^{(7)}(Z,\bar{Z}) = \chi'_{01}(Z,\bar{Z})\chi'_{01}(Z,\bar{Z})\chi_2(Z,\bar{Z}),$$
$$\Psi_0^{(8)}(Z,\bar{Z}) = \chi'_{11}(Z,\bar{Z})\chi'_{01}(Z,\bar{Z})\chi_2(Z,\bar{Z}),$$
$$\Psi_0^{(9)}(Z,\bar{Z}) = \chi'_{11}(Z,\bar{Z})\chi'_{11}(Z,\bar{Z})\chi_2(Z,\bar{Z}). \tag{215}$$

Here, the root state corresponding to $\Psi_0^{(i)}(Z,\bar{Z})$ is given by $\langle Z,\bar{Z}|\Psi_{\text{root}}^{(i)}\rangle$ as in Eq. (60) for $N = 2$. Using the parton construction, we have successfully determined all possible two particle ground states for the EPPs we have derived for $N_L = 4$ projected Hamiltonian. We will use these constraints for many-particle systems to construct many-particle root patterns of the zero-energy modes of our Hamiltonian.

## 6.3 Parton-like states from a given root pattern

We have, so far, discussed how to extract root patterns from parton-like states. One may be interested in determining whether a given root pattern is compatible with a valid ground state, that is, one that respects the EPP. We will now start with a given root pattern and construct

Figure 18: Slater determinants $\chi'_{nn'}(Z,\bar{Z})$ and $\chi''_{nn'}(Z,\bar{Z})$ for $2r-1$ particles in two LLs. In $\chi'_{nn'}(Z,\bar{Z})$ there is single occupancy in $j_i = 0$ and $r-1$. For $j_i = 0$ ($j_i = r-1$) the $n^{th}$ ($n'^{th}$) orbital is occupied. In $\chi''_{nn'}(Z,\bar{Z})$, the $j_i = r-1$ is unoccupied and $j_i = 0$ and $r$ are singly occupied. For $j_i = 0$ ($j_i = r$) the $n^{th}$ ($n'^{th}$) orbital is occupied. For the states above, one can use the pseudospin algebra of Eq. (177). For $\chi'_{nn'}$ and $\chi''_{nn'}$ the pseudospin is given by $n+n'-1$.

possible parton-like states. It is useful to remind the reader of the concise definition of parton-like states. A parton-like state is a product of $M$ building blocks. Each building block is a Slater determinant of particle coordinates $z_i, \bar{z}_i$, $i = 1, \cdots, N$. These Slater determinants can be further translated into an increasing set of angular momenta in second quantized language. For a given root pattern $\{\lambda\}_{\text{root}}$, in the angular momentum basis $\{j\}_{\text{root}} = \{j_1, j_2, j_3, ..., j_N\}_{\text{root}}$, an allowed parton-like state should enable an integer $M$-partition for each $j_i$

$$j_i = j_i^{(1)} + j_i^{(2)} + ... + j_i^{(M)}, \qquad i = 1, ..., N, \tag{216}$$

such that $j_i^{(\mu)} \leq j_j^{(\mu)}$, $\forall\, \mu \in \{1, ..., M\}$, $i < j$, with the constraint that for fixed $\mu$ the number of identical $j_i^{(\mu)}$'s must be $\leq N_m^{(\mu)}$ where $N_m^{(\mu)} = \min(N_L^{(\mu)} + j_i^{(\mu)}, N_L^{(\mu)})$ is the maximal multiplicity. Here, $N_L^{(\mu)}$, $1 \leq N_L^{(\mu)} \leq N_L$, represents the number of LLs making up the $\mu$ Slater determinant satisfying $\sum_\mu N_L^{(\mu)} = N_L + M - 1$ (see Eq. (145)). Under these constraints we can organize the data in the following table

$$
\begin{array}{cccccl}
j_1 & j_2 & j_3 & \cdots & j_N & \\
\begin{bmatrix}
j_1^{(1)} & j_2^{(1)} & j_3^{(1)} & \cdots & j_N^{(1)} \\
j_1^{(2)} & j_2^{(2)} & j_3^{(2)} & \cdots & j_N^{(2)} \\
\vdots & \vdots & \vdots & \ddots & \vdots \\
j_1^{(M)} & j_2^{(M)} & j_3^{(M)} & \cdots & j_N^{(M)}
\end{bmatrix}
&&&&&
\begin{matrix}
N_m^{(1)} \\
N_m^{(2)} \\
\vdots \\
N_m^{(M)}
\end{matrix}
\end{array}
\tag{217}
$$

and, if the constraints are satisfied, it leads to the parton-like state

$$\Psi_p(Z,\bar{Z}) = \prod_{\mu=1}^{M} \chi_{N_L^{(\mu)}}(Z,\bar{Z}), \tag{218}$$

where the $N$-particles Slater determinants $\chi_{N_L^{(\mu)}}(Z,\bar{Z})$ are made out of orbitals spanning $N_L^{(\mu)}$ LLs with angular momenta $\{j_1^{(\mu)}, j_2^{(\mu)}, j_3^{(\mu)}, \cdots, j_N^{(\mu)}\}_{\text{root}}$.

Let us illustrate the algorithm by applying it to a simple example. Consider the case of $N = 5$, $M = 3$, $N_L = 4$ with root pattern 1002002 ($\{-3, 0, 0, 3, 3\}_{\text{root}}$ in angular momentum representation). Our first step amounts to finding the possible integer partitions of six (since $N_L + M - 1 = 4 + 3 - 1 = 6$), subject to the above noted constraints, leading to $[N_L^{(1)}, N_L^{(2)}, N_L^{(3)}]$.

In this example, these integer partitions are $[2, 2, 2]$, $[1, 2, 3]$, and $[1, 1, 4]$. Next, following Eq. (216), we find all possible partitions of $\{-3, 0, 0, 3, 3\}_{\text{root}}$ in each decomposition. For $[2, 2, 2]$ the solution can be written as

$$
\begin{array}{ccccc}
-3 & 0 & 0 & 3 & 3
\end{array}
$$
$$
\begin{bmatrix}
-1 & 0 & 0 & 1 & 1 \\
-1 & 0 & 0 & 1 & 1 \\
-1 & 0 & 0 & 1 & 1
\end{bmatrix}
\begin{array}{c}
2 \\
2 \\
2
\end{array}
\tag{219}
$$

Notice that the boundary state with angular momentum $-1$ can only appear once as per our algorithm. For the other two decompositions, i.e., $[1, 2, 3]$, $[1, 1, 4]$, we do not have any solution which satisfies Eq. (216) and the constraints. In each decomposition, one Slater determinant has $N_L^{(3)} = 1$. Hence, $j_1^{(1)} < j_2^{(1)} < j_3^{(1)} < j_4^{(1)} < j_5^{(1)}$. But, $j_2 = j_3 = 0$, implying $j_2 - j_2^{(1)} > j_3 - j_3^{(1)}$. Using $j_i = j_i^{(1)} + j_i^{(2)} + j_i^{(3)}$ together with the above observation we get, $j_2^{(2)} + j_2^{(3)} > j_3^{(2)} + j_3^{(3)}$. This clearly contradicts our assumption that, $j_i^{(\mu)} \le j_j^{(\mu)}, \forall i < j$. Hence, we conclude that the root pattern 1002002 has associated only the parton state $[2, 2, 2]$ with identical Slater determinants of angular momentum $\{-1, 0, 0, 1, 1\}_{\text{root}}$. This state corresponds to the closed-shell parton structure $\chi_2(Z, \bar{Z})^3$, the unique parton solution allowed for the 1002002 root pattern.

For a (non closed-shell) less dense root pattern in the ground state, one usually has multiple parton-like solutions. Consider the root pattern 1101010101 ($N = 6$, $M = 3$, $N_L = 4$) having the angular momentum representation $\{0, 1, 3, 5, 7, 9\}_{\text{root}}$. This pattern admits more than one solution,

$$
\begin{array}{cccccc}
0 & 1 & 3 & 5 & 7 & 9
\end{array}
$$
$$
\begin{bmatrix}
0 & 0 & 1 & 1 & 2 & 2 \\
0 & 0 & 1 & 2 & 2 & 3 \\
0 & 1 & 1 & 2 & 3 & 4
\end{bmatrix}
\begin{array}{c}
2 \\
2 \\
2
\end{array}
\qquad \rightarrow [2, 2, 2],
\tag{220}
$$

$$
\begin{array}{cccccc}
0 & 1 & 3 & 5 & 7 & 9
\end{array}
$$
$$
\begin{bmatrix}
0 & 0 & 1 & 1 & 2 & 2 \\
0 & 0 & 1 & 1 & 2 & 2 \\
0 & 1 & 1 & 3 & 3 & 5
\end{bmatrix}
\begin{array}{c}
2 \\
2 \\
2
\end{array}
\qquad \rightarrow [2, 2, 2].
\tag{221}
$$

Both of these parton-like states share the same, 1101010101, root pattern.

Clearly, given a root pattern it is possible not to have any single parton-like state associated to it. To illustrate, we discuss $N = 7$, $M = 3$ and $N_L = 4$, which has root pattern 100111000111 with angular momentum representation $\{-3, 0, 1, 2, 6, 7, 8\}_{\text{root}}$. To derive this root pattern, we need to satisfy following constraints

$$
\begin{aligned}
j_2 &= j_2^{(1)} + j_2^{(2)} + j_2^{(3)} = 0, \\
j_3 &= j_3^{(1)} + j_3^{(2)} + j_3^{(3)} = 1, \\
j_4 &= j_4^{(1)} + j_4^{(2)} + j_4^{(3)} = 2.
\end{aligned}
\tag{222}
$$

To satisfy these constraints along with, $j_i^{(\mu)} \le j_j^{(\mu)}, \forall i < j$, $\{j_3^{(1)}, j_3^{(2)}, j_3^{(3)}\}$ must have at least two common elements with both $\{j_2^{(1)}, j_2^{(2)}, j_2^{(3)}\}$ and $\{j_4^{(1)}, j_4^{(2)}, j_4^{(3)}\}$. In other words, at least one $j_i^{(\mu)}$ must appear in all three cases. Without loss of generality, we assume all $j_i^{(1)}$'s for i = 2, 3, 4 are

identical. Thus, one Slater determinant must have, at least $N_L^{(1)} = 3$. Moreover, $\{j_2^{(1)}, j_2^{(2)}, j_2^{(3)}\}$ and $\{j_3^{(1)}, j_3^{(2)}, j_3^{(3)}\}$ have one more common element. Again, we assume without any loss of generality, identical $j_i^{(2)}$'s for i = 2, 3. Finally, $\{j_3^{(1)}, j_3^{(2)}, j_3^{(3)}\}$ and $\{j_4^{(1)}, j_4^{(2)}, j_4^{(3)}\}$ should have two common elements. Given identical $j_i^{(1)}$'s for i = 2, 3, 4 and identical $j_i^{(2)}$'s for i = 2, 3, we have two scenarios,

1. $j_3^{(2)} = j_4^{(2)}$. In this scenario, $j_i^{(2)}$'s for i = 2, 3, 4 are the same and $N_L^{(2)} \geq 3$. In this case, $N_L^{(1)} + N_L^{(2)} + N_L^{(3)} \geq 3 + 3 + 1 > 6$.

2. $j_3^{(3)} = j_4^{(3)} \implies N_L^{(2)} \geq 2$ and $N_L^{(3)} \geq 2$. Thus, $N_L^{(1)} + N_L^{(2)} + N_L^{(3)} \geq 3 + 2 + 2 > 6$.

In both scenarios, in order to have a parton-like solution, we need $N_L + M - 1 > 6$. However, this inequality is not satisfied given that $N_L = 4$ and $M = 3$.

As an illuminating application of our algorithm, we next show that the bulk root pattern $\{11000\}$ (equal to $\{0, 1, 5, 6, 10, 11, \ldots\}_{\text{root}}$ in angular momentum representation) for arbitrary $N_L$ (the Gaffnian 2/5 state [80] corresponds to $N_L = 1$) cannot have a closed-shell parton structure associated to it, although it can have a parton-like structure. For a closed-shell parton, we have one additional constraint, $j_{i+1}^{(\mu)} \leq j_i^{(\mu)} + 1$ for all $\mu$, i.e., all shells must be filled. Assume that there exists a closed-shell parton state with a zero of order $M \in$ odd. Then, starting at $j_1 = 0$

$$j_1 = j_1^{(1)} + j_1^{(2)} + j_1^{(3)} + \ldots + j_1^{(M)} = 0\,,$$
$$j_2 = j_2^{(1)} + j_2^{(2)} + j_2^{(3)} + \ldots + j_2^{(M)} = 1\,.$$

Thus, $j_1^{(\mu)} = j_2^{(\mu)}$ for any set of $M - 1$ $\mu$ values. Without any loss of generality, we assume $j_2^{(1)} = j_1^{(1)} + 1$. Being a closed-shell parton, this is possible only if the first Slater determinant has a single LL (no degeneracy) with angular momentum $\{0, 1, 2, 3, 4, 5, \ldots\}_{\text{root}}$. Subtracting $\{0, 1, 2, 3, 4, 5, \ldots\}_{\text{root}}$ from the original root pattern $\{0, 1, 5, 6, 10, 11, \ldots\}_{\text{root}}$ leads to $\{0, 0, 3, 3, 6, 6, \ldots\}_{\text{root}}$, which is the same as $\{\lambda\}_{\text{broot}} = \{200\}$. Then, the rest of the Slater determinants in the parton must form a root pattern of the form $\{200\}$. But we have already shown that the only closed-shell parton associated with such root pattern is $[2, 2, 2]$. Thus, for $\{11000\}$ the only possible closed-shell parton is $[1, 2, 2, 2]$ with a $4^{\text{th}}$ order zero, i.e., $M = 4$. However, a fermionic state should have $M \in$ odd. Thus, we proved that the fermionic $\{11000\}$ root pattern cannot have a closed-shell parton structure.

## 6.4 Completeness of parton-like states for $M = 3$ in 4 LLs

We begin by showing that there exist parton-like states giving every possible root pattern consistent with the EPP. We do so by solving explicitly Eq. (216) for $N_L^{(\mu)} = 2$, $\mu = 1, 2, 3$. This can be done in the following way:

$$
\begin{aligned}
j_i^{(1)} &= \lfloor j_i/3 \rfloor\,, \\
j_i^{(2)} &= \lceil j_i/3 \rceil\,, \\
j_i^{(3)} &= j_i - j_i^{(1)} - j_i^{(2)}\,,
\end{aligned}
\tag{223}
$$

where, $\lfloor \ \rfloor$ and $\lceil \ \rceil$ are the floor and ceiling functions, respectively. This obviously satisfies Eq. (216), is monotonically increasing in i, and it is consistent with $N_L^{(\mu)} = 2$ for the following reason: For any pattern consistent with the EPP, increasing the particle index i by 2 increases $j_i$ by at least 3, thus, every row in Eq. (223) by at least 1. Thus, for every fixed $\mu = 1, 2, 3$, $j_i^{(\mu)}$ can assume every value at most twice as a function of i. It remains to be shown that the value

of $-1$ can occur at most once. The EPP must be supplemented by boundary conditions "on the left", associated to negative angular momenta, as shown in Appendix D. For root *patterns*, these boundary conditions can be simply summarized as enforcing

$$j_2 \geq 0, \tag{224}$$

in addition to the already established rules. In Eq. (223), this then trivially also implies $j_i^{(\mu)} \geq 0$.

Next, we argue that, moreover, for every root *state* that is the product of MPS as constructed in Sec. 6.1.2 and general factors 00200 and $001_{s_z}00$, there is a corresponding parton-like state. Given now a root state $|\Phi\rangle$ with the MPS/product structure defined above, we construct a parton-like $|\psi\rangle$ state having a root state corresponding to the root pattern of $|\Phi\rangle$. Now we compare $|\Phi\rangle$ to the root state $|\psi_{\text{root}}\rangle$ of $|\psi\rangle$, each in general a tensor product of mutually unentangled units. Every factor '2' in this product must be the same in $|\Phi\rangle$ and $|\psi\rangle$, as $|\psi\rangle$ is a zero mode, and this determines the state of any '2' in its root state uniquely, as we have seen. Likewise, any 1100-string in $|\psi_{\text{root}}\rangle$ automatically follows the MPS construction principle of Sec. 6.1.2. However, a string involving 1's has certain degeneracies associated to it that must be recovered in general root states obtained (via the rule of (223)) from parton-like states. Indeed, one verifies from this rule that every leading 1 in a ...00101 unit will be mapped to a singly occupied $j$-orbital in exactly two of the three Slater determinants. An analogous statement holds for the left-right reversed situation. Every central 1 in a 10101-pattern will be mapped to a singly occupied $j$-orbital in exactly one of the three Slater determinants. Every $...001_{s_z}00$ unit will be mapped to singly occupied $j$-orbitals in all three Slater determinants. Each singly occupied $j$-orbital in a Slater determinant leads to a free spin-1/2 degree of freedom in the root level of the MPS. Indeed, all of the expected spin-1/2 degrees of freedom of the MPS associated with the 1-carrying patterns are generated in this way and lead to all of the possible MPS described in Sec. 6.1.2. This illustrates that for every possible MPS-solution for the EPP, we find a parton-like state whose root state or DNA is precisely this MPS-solution.

The completeness of the parton-like states as zero modes is now obtained as follows. We may assume that the MPS states described in Sec. 6.1.2 represent a complete set of solutions for the EEP governing the ...110110... pattern, based on general arguments for AKLT-type constructions [68]. Then, the MPS/product states discussed here represent a complete set of possible root states $\{|\Phi_d\rangle\}$, where $d$ is some label referencing all such states. (Here and in the following, we may restrict to fixed total angular momentum $J$ to keep the setting finite dimensional). By the above, we always have a parton-like state $|\psi_d\rangle$ whose root state is $|\Phi_d\rangle$. We may now reproduce the proof given in Ref. [14] for the completeness of the states $|\psi_d\rangle$ as zero modes. By construction, $\langle\Phi_d|\psi_d\rangle \neq 0$ yet the matrix $\langle\Phi_{d'}|\psi_d\rangle$ need not be diagonal. Nonetheless, for given $|\psi_d\rangle$, every $|\Phi_{d'}\rangle$ with $d' \neq d$ is not the root state of $|\psi_d\rangle$. If $\langle\Phi_{d'}|\psi_d\rangle \neq 0$, then $|\Phi_{d'}\rangle$ consists of Slater determinants that can be obtained from those of $|\Phi_d\rangle$ via inward-squeezing processes. This is enough to show that an ordering of the labels $\{d\}$ exists such that the matrix $\langle\Phi_{d'}|\psi_d\rangle$ is triangular. Thus, this matrix is invertible. Given any zero-mode $|\psi\rangle$, this fact allows the construction of a superposition $|\psi'\rangle$ of parton-like states $|\psi_d\rangle$ such that $\langle\Phi_d|\psi\rangle = \langle\Phi_d|\psi'\rangle$. The difference $|\psi\rangle - |\psi'\rangle$ is then a zero mode that is orthogonal to all possible root states, a contradiction unless $|\psi\rangle = |\psi'\rangle$. Therefore, $|\psi\rangle$ is a superposition of $|\psi_d\rangle$'s.

# 7 Conclusions and outlook

Traditionally, FQH systems have been largely examined either via wave function Ansatz or effective field theories. The links between these two approaches run deep with illuminating

insights between edge conformal field theories (CFTs) and bulk polynomial wave functions, and relations to edge CFTs and topological quantum field theories (TQFTs) on the other. In this paper, we proceeded along an inter-related third approach rooted in the study of microscopic many-body Hamiltonians. We studied the structure of *entangled multiple Landau level* (LL) *states* and their *local* (in real space) many-body parent Hamiltonians. Our results may potentially further help bridge the divide between the above noted microscopic wave functions and long distance continuum field theories. A focus of our work was the study of universal structures present in multiple, $N_L$, LL FQH systems. We have studied the general zero-mode structure of rather general positive semi-definite local Hamiltonians. In our analysis of their zero modes, a key role was played by the "$M$-clustering property" of QH wave functions- the existence of $M$-th order zeroes near a two-particle coincidence hyperplane. Since Laughlin's celebrated construction of his variational wave function for the 1/3 FQH state, numerous wave functions with $M$-clustering properties have seen proposed. Laughlin's wave function can be expressed as the product of $M$ Slater determinants of lowest LL (LLL) orbitals (expressed in terms of holomorphic variables), with filling fraction of the form $1/M$ (odd $M$). Jain has further extended this construct by introducing his composite fermion picture, wherein a single LL is replaced by multiple $\Lambda$-levels. In spite of their immense success in explaining a plethora of FQH plateaus, all of these states are qualitatively similar to integer QH states. Filling fractions observed at higher LLs, such as [81] 5/2, 12/5, 7/3, and 8/3 are, however, suspected to exhibit more intricate physics demanding far more complicated wave functions. A unified systematic understanding of these systems invites the search of general principles and tools of analysis.

Inspired by these all too well-known challenges, we extended the Hamiltonian approach to more general FQH states. In the current work, we examined, in great detail, the zero-mode subspace of the aforementioned local, multiple LL, two-body Hamiltonians. Consequently, we determined a basis for the Hilbert space formed by polynomials associated with general states that entangle the different LLs and obey the $M$-clustering property. We conjectured that these basis elements are parton-like, i.e., are products of $M$ Slater determinants, and rigorously demonstrated it for $M = 3$, $N_L \leq 4$. These parton-like structures capture a very rich class of states. Our parent Hamiltonians are frustration-free QH Hamiltonians. The construction of the Laughlin wave function as the above noted product of Slater determinants—a parton state built from LLL wave functions satisfying the two-body $M$-clustering properties—can be extended in several ways. These principally include the construction of (a) LLL (holomorphic) wave function with k-body (k > 2) $M$-clustering properties or, as we pursue in this paper, of (b) parton states (two-body $M$-clustering) from multiple LLs. The more traditional approach (a) gives rise to the Moore-Read (MR) 1/2 state (candidate for 5/2 filling fraction [3]), the Read-Rezayi (RR) 2/3 state (candidate for 12/5 filling fraction [82]), and many other CFT inspired states. The MR and RR states are members of a polynomial space satisfying the $M$-clustering property. We have, instead, followed the aforementioned approach (b). As emphasized above, we extended parton states to higher LLs. In particular, we constructed candidate FQH states that are topologically similar to the MR and RR states. These states not only provide candidate wave functions for several FQH plateaus but can also be further associated with frustration-free positive semi-definite two-body parent Hamiltonians in flat bands with higher Chern numbers [83]. Our specific analysis focused on degenerate LLs arising when the kinetic energy is quenched. As such, our results may capture the detailed physics of systems that allow for mixing between multiple nearly flat bands such as those realized in layered graphene [10,84]. In multi-layer graphene, multiple degenerate LLs can appear with quenched kinetic energy.

Within our parton construction, we are not limited to states with conformal block structures. Constructing a multiple LL parton state is far more challenging than that of its single LL counterpart. There exist no effective flux attachment analogies in the former case and the

resulting wave function can have very different edge excitation than the constituting Slater determinant states. In order to establish these states as the unique densest ground state of some two-body parent Hamiltonian, we have used fundamental organizing principles, known as the Entangled Pauli Principles (EPPs). The EPP provides a rigorous zero-mode counting method for zero-energy excitations and determines, in a way that we make precise, the quintessential DNA of the densest ground state. The zero-mode counting enabled by the EPP for the non-holomorphic multiple LL states prompted queries that go beyond known mathematical results. In particular, our results and resulting conjectures can be interpreted as new developments in commutative algebra, of which we prove special instances. Specifically, we posit that *all* polynomials $P(z_1, z_2, \ldots, z_N, \bar{z}_1, \bar{z}_2, \ldots \bar{z}_N)$ of complex variables $\{z_i\}$ and their complex conjugates $\{\bar{z}_i\}$ that (a) adhere to (anti-)symmetry under the interchange of any pair of complex coordinates, (b) satisfy the $M$-clustering property, and (c) are not of order higher than $n = N_L - 1$ in any $\bar{z}_i$, are linearly spanned by parton-like states. Our study of the EPP and FQH DNAs can further be used in torus geometry to construct a coherent state description of the ground state encoding useful information such as exchange statistics and topological classification for the corresponding parton states.

We have shown that in the toroidal geometry, our parent Hamiltonians satisfy an S-duality. That is, there exists a duality that links Hamiltonians associated with two different aspect ratios of the torus (i.e., those with $L_x/L_y < 1$ to those with $L_x/L_y > 1$). The S-duality enabled us to extract characteristic properties of the QH fluid, being a key ingredient in our approach to the quasiparticle statistics. For instance, for a QH system in the subspace of four LLs we have obtained the DNA of the fluid and topological classifications. We have furthermore demonstrated that the excitations are none other than *Fibonacci anyons*. As is well known, Fibonacci anyons may provide a simple platform for achieving universal topological quantum computation. In an earlier work, we found that the corresponding excitations for the Jain-221 state [15] are Majorana fermions. Majorana and Fibonacci statistics are naturally associated with groups suggested by underlying TQFTs (related to $SU(2)_2$ and $SU(2)_3$ respectively). Thus, the results of our braiding analysis exhibit natural connections between the DNA and associated EPPs, which emerge microscopically in our models, and TQFTs. It is noteworthy that the domain walls generally arising in our multiple LL setting are not simple domain walls such as those in classical spin chains nor a tensor product of such defects. Rather, these are bona fide *quantum topological defects* that may feature entanglement.

In the context of strongly correlated physics, one often uses the Hilbert space of Slater determinants as a basis for numerical calculations. However, the dimension of the Hilbert space of parton-like states with an $M$-clustering property is drastically smaller than the Hilbert space of Slater determinants. For this reason, we believe that using the Hilbert space of parton-like states reduces complexity of numerical calculations for strongly correlated many-body systems such as QH systems, or other non-Fermi liquids. In the particular context of Quantum Monte Carlo simulations, updating Slater determinants becomes polynomially efficient because of the Sherman-Morrison formula [85]. This procedure can also be extended to the case of parton-like states. We postpone the elaboration of these ideas for a later publication.

Beyond our specific results, our work provides a general framework that naturally highlights several broad concepts and further underscores several open questions. We elaborate on one of these below.

*The relation between generic parton states and boundary Conformal Field Theories.* In the current work, we derived numerous results for the system bulk. However, apart from insightful Chern-Simons theory type conjectures, [9] a systematic understanding of the edge theory of general parton states is non-existent. Our general approach to QH states lies outside the purview of standard CFT framework in which the boundary behaviors are transparent. Indeed, nowhere in our analysis have we relied on CFT notions. This is partially so since standard CFT

recipes cannot be straightforwardly applied to general non-holomorphic states such as the one that we investigate here. Obtaining the associated boundary theories for generic parton states is a non-trivial challenge.

We next briefly speculate on how our exact many-body Hamiltonian and zero-mode counting based scheme may be effective in establishing the link between non-holomorphic QH states and their effective boundary theories. The results of our approach must be in a one-to-one correspondence with the zero-mode counting of the conformal edge theory. In the conventional CFT type *modus operandi*, plausible CFTs are guessed and, subsequently, a check is performed to see whether the conformal blocks in these proposed CFTs match with those of the wave function. Given a candidate CFT, the number of possible edge modes of a given angular momentum may be computed. Our method should enable the unambiguous identification of the boundary theory by employing zero-mode counting that can be rigorously established via the use of the EPPs. This may afford as strong a connection between mixed-LL parton QH states and their effective edge CFTs as that which one usually takes for granted in the lowest LL. For the Jain-221 state [15], we have indeed made such a zero-mode counting based "bulk-boundary correspondence type" connection rigorous. We anticipate such a link to be far more general. This may complement, especially for non-holomorphic states, the insightful conformal block trick of Moore and Read. In other words, we speculate that a zero-mode counting of the bulk states (using the precise many-body microscopic Hamiltonian that we employed in the current work) may, generally, lead to the relevant edge theories. This approach will not invoke discussions of effective Chern-Simons theories. In particular, such a many-body based technique may be applicable for generic parton theories for which there are currently no known CFTs. As we additionally explained in the current work, the coherent state method enables a way to infer the bulk braiding statistics. Taken together, all of the above ingredients suggest how our many-body approach may allow unambiguous determination of effective field theory from microscopic Hamiltonians.

# Acknowledgements

It is a pleasure to acknowledge helpful discussions with Michael J. Larsen and for bringing to our attention Ref. [77]. We are also indebted to Jainendra K. Jain and Terrence Tao for valuable comments.

**Funding information** G. O. acknowledges support from the US Department of Energy grant DE-SC0020343. A. S. acknowledges support by the National Science Foundation under Grant No. DMR-2029401. M. T. A. acknowledges support by the US Department of Energy, Office of Basic Energy Sciences, under Award No. DE-SC0012190.

# A  Two-particle states: From wave functions to Fock states

Consider the following normalized vacua

$$|0, 0, 2j - m, m\rangle = \frac{1}{\sqrt{(2j-m)!m!}} \, b_c^{\dagger 2j-m} b_r^{\dagger m} |0\rangle = \frac{1}{\sqrt{(2j-m)!m!}} \sum_k \frac{C_{mjk}}{2^j} \, b_1^{\dagger j-k} b_2^{\dagger j+k} |0\rangle,$$

where

$$\langle z_c, \bar{z}_c; z_r, \bar{z}_r | 0 \rangle = \frac{1}{\sqrt{2\pi\ell_c^2}} e^{-\frac{z_c \bar{z}_c}{4\ell_c^2}} \frac{1}{\sqrt{2\pi\ell_r^2}} e^{-\frac{z_r \bar{z}_r}{4\ell_r^2}}, \tag{A.1}$$

Table 8: Quantum numbers labeling the fermionic basis operators $T^{n_1,n_2-}_{j,m}$ for fixed $j$ and $N_L = 4$ LLs.

| $I$ | 1 | 2 | 3 | 4 | 5 | 6 | 7 | 8 | 9 | 10 | 11 | 12 | 13 | 14 | 15 | 16 | 17 | 18 | 19 | 20 | 21 | 22 | 23 | 24 | 25 | 26 | 27 | 28 | 29 | 30 | 31 | 32 | 33 | 34 | 35 | 36 | 37 | 38 | 39 | 40 |
|---|---|---|---|---|---|---|---|---|---|---|---|---|---|---|---|---|---|---|---|---|---|---|---|---|---|---|---|---|---|---|---|---|---|---|---|---|---|---|---|---|
| $n_1$ | 0 | 0 | 1 | 0 | 1 | 1 | 2 | 2 | 2 | 2 | 2 | 2 | 2 | 2 | 2 | 2 | 2 | 2 | 2 | 0 | 0 | 3 | 3 | 3 | 3 | 3 | 3 | 3 | 3 | 3 | 3 | 3 | 3 | 3 | 2 | 2 | 2 | 3 | 3 | 3 |
| $n_2$ | 0 | 1 | 0 | 1 | 1 | 1 | 0 | 0 | 0 | 0 | 1 | 1 | 1 | 1 | 1 | 1 | 2 | 2 | 2 | 3 | 3 | 0 | 0 | 0 | 1 | 1 | 1 | 1 | 1 | 1 | 2 | 2 | 2 | 2 | 3 | 3 | 3 | 3 | 3 | 3 |
| $m$ | 1 | 0 | 2 | 1 | 1 | 3 | 1 | 3 | 0 | 2 | 1 | 3 | 0 | 2 | 4 | 1 | 3 | 5 | 1 | 3 | 0 | 2 | 4 | 0 | 2 | 4 | 1 | 3 | 5 | 0 | 2 | 4 | 6 | 1 | 3 | 5 | 1 | 3 | 5 | 7 |

with $z_c = (z_1 + z_2)/2$, $z_r = z_1 - z_2$, $\ell_c = \ell/\sqrt{2}$, and $\ell_r = \sqrt{2}\ell$. In this paper, unless specified otherwise, we set the magnetic length $\ell_r$ to be the unit of length, i.e., $\ell_r = 1$ or $\ell = 1/\sqrt{2}$. On the torus geometry, for convenience, we will work with $\ell = 1$.

Each individual vacuum state is of total angular momentum $2j$ and lies in the LLL (i.e., is a holomorphic state in a first quantization description). In a disk geometry, when using the symmetric gauge $\mathbf{A}(x_i, y_i) = \frac{B}{2}(y_i\hat{x} - x_i\hat{y})$, the LL and cyclotron-orbit-center ladder operators become

$$a_i = \frac{1}{\sqrt{2}}\left(\frac{z_i}{2\ell} + 2\ell\,\partial_{\bar{z}_i}\right), \qquad a_i^\dagger = \frac{1}{\sqrt{2}}\left(\frac{\bar{z}_i}{2\ell} - 2\ell\,\partial_{z_i}\right),$$
$$b_i = \frac{1}{\sqrt{2}}\left(\frac{\bar{z}_i}{2\ell} + 2\ell\,\partial_{z_i}\right), \qquad b_i^\dagger = \frac{1}{\sqrt{2}}\left(\frac{z_i}{2\ell} - 2\ell\,\partial_{\bar{z}_i}\right). \tag{A.2}$$

As a result, the coefficients in the expansion of Eq. (A.1) are given by [24]

$$C_{mjk} = (-1)^{m+j-k}\sum_{q=0}^{j-k}(-1)^q\binom{2j-m}{q}\binom{m}{j-k-q}. \tag{A.3}$$

This expression indicates that $j-k$ is an integer. Thus, if $j$ is an integer (half-odd integer) then $k$ is an integer (half-odd integer). The normalized vacua in Eq. (A.1) may be either symmetric ($m \in$ even) or antisymmetric ($m \in$ odd) under particle exchange, the fermionic basis states $|I\rangle_F$ given in Eq. (11) are, by definition, antisymmetric.

We want now to find the Slater determinant decomposition of the two-particle states $|I\rangle_F$. This decomposition will allow us to find an immediate representation of the corresponding state in terms of fermionic operators. In turn, this decomposition will reveal a fundamental two-particle generator in Fock space. It can be checked that Eq. (11) can be expressed as

$$|I\rangle_F = \frac{1}{\sqrt{n_1!n_2!\,2^{2j+1}(1+\delta_{n_1,n_2})}}\sum_{k=-j}^{j}\frac{C_{mjk}}{\sqrt{(2j-m)!m!}}\,Q(k)|0\rangle\,,$$

where we have used the property that, for $m$ odd (even), $C_{mj-k} = -(+)C_{mjk}$, and

$$Q(k) = \begin{vmatrix} a_1^{\dagger n_1}b_1^{\dagger j-k} & a_2^{\dagger n_1}b_2^{\dagger j-k} \\ a_1^{\dagger n_2}b_1^{\dagger j+k} & a_2^{\dagger n_2}b_2^{\dagger j+k} \end{vmatrix}. \tag{A.4}$$

In first quantization,

$$\langle z_1,\bar{z}_1; z_2,\bar{z}_2|Q(k)|0\rangle = \sqrt{n_1!n_2!(j-k)!(j+k)!}\,D_{\alpha_1\alpha_2}\,,$$

showing that the operator $Q(k)$ is a generator of two-particles Slater determinants. Now defining

$$\eta_k(j,m) \equiv \sqrt{\frac{(j-k)!(j+k)!}{2^{2j-1}(2j-m)!m!}}\,C_{mjk}\,, \tag{A.5}$$

Eq. (12) is obtained.

# B Interaction potential expansion

In this Appendix, we will obtain the general pseudopotential expansion for any sufficiently short range two-body (rotationally symmetric) interaction. Consider the two-body interaction $V(\mathbf{r}_i - \mathbf{r}_j) = V(\mathbf{r}_{ij})$ with Fourier transform

$$V(\mathbf{r}_{ij}) = \int \frac{d^2\mathbf{k}}{(2\pi)^2} \tilde{V}(\mathbf{k}) e^{i\mathbf{k}\cdot\mathbf{r}_{ij}} = \int_0^\infty \frac{dk}{2\pi} k \tilde{V}(k) J_0(k|\mathbf{r}_{ij}|), \tag{B.1}$$

where $J_0(x)$ is the zeroth spherical Bessel function [53]. From the definition of the delta function

$$L_\alpha(-\ell^2\nabla_{ij}^2)\delta^2(\mathbf{r}_{ij}) = \int \frac{d^2\mathbf{q}}{(2\pi)^2} L_\alpha(\ell^2 q^2) e^{i\mathbf{q}\cdot\mathbf{r}_{ij}} = \int_0^\infty \frac{dq}{2\pi} q L_\alpha(\ell^2 q^2) J_0(q|\mathbf{r}_{ij}|), \tag{B.2}$$

where $L_\alpha(x)$ is the $\alpha$th Laguerre polynomial [53], and $\ell$ the magnetic length. We multiply the last expression by

$$V_\alpha = \ell^2 \int_0^\infty dk^2 \, \tilde{V}(k) L_\alpha(\ell^2 k^2) e^{-\ell^2 k^2}, \tag{B.3}$$

and sum over the whole range of $\alpha$'s. After using the identity

$$\ell^2 \sum_{\alpha=0}^\infty L_\alpha(\ell^2 k^2) L_\alpha(\ell^2 q^2) = \delta(q^2 - k^2) e^{\ell^2(k^2+q^2)/2},$$

we arrive at the pseudopotential expansion [52,86]

$$V(\mathbf{r}_i - \mathbf{r}_j) = \sum_{\alpha=0}^\infty V_\alpha L_\alpha(-\ell^2\nabla_{ij}^2)\delta^2(\mathbf{r}_i - \mathbf{r}_j). \tag{B.4}$$

The expansion to lowest order is

$$V(\mathbf{r}_i - \mathbf{r}_j) = (V_0 + V_1 + V_1\ell^2 \nabla_{ij}^2)\delta^2(\mathbf{r}_i - \mathbf{r}_j). \tag{B.5}$$

Note that terms proportional to $\delta^2(\mathbf{r}_i - \mathbf{r}_j)$ have vanishing matrix elements for fermionic wave functions.

When projected onto the LLL, the expansion above coincides with the Haldane pseudopotential over the relative angular momentum $\hbar\alpha = \hbar m$.

Eigensolutions of Eq. (25) with vanishing eigenvalue must have (at least) third order zeros when two fermions coalesce. This defines the clustering properties of the zero modes, as we show next.

Given a zero energy state $|\Psi_0\rangle$, i.e., $\langle\Psi_0|H_{\text{int}}|\Psi_0\rangle = 0$, assume it is of the general form

$$\Psi_0(Z,\bar{Z}) = \sum_{q=0}^M z_{ij}^q \bar{z}_{ij}^{M-q} P_q(Z,\bar{Z}) e^{-\frac{1}{4\ell^2}\sum_{i=1}^N z_i\bar{z}_i} \tag{B.6}$$

in coordinate representation, with $Z = \{z_1, z_2, \cdots, z_N\}$ and $\bar{Z} = \{\bar{z}_1, \bar{z}_2, \cdots, \bar{z}_N\}$, $z_{ij} = z_i - z_j$, $\bar{z}_{ij} = \bar{z}_i - \bar{z}_j$, $P_q(Z,\bar{Z})$ a polynomial symmetric with respect to $(z_i, \bar{z}_i) \leftrightarrow (z_j, \bar{z}_j)$, and (anti-)symmetric with respect to other variables exchanges (it does not depend on the coordinate differences $z_{ij}$, $\bar{z}_{ij}$), and $M$ an (odd)even integer.

Then, from the zero energy condition and integration by parts

$$\langle\Psi_0|H_{\text{int}}|\Psi_0\rangle = V_1\ell^2 \int dZd\bar{Z} \sum_{i<j} \delta^2(\mathbf{r}_{ij})\partial_{\bar{z}_{ij}}\partial_{\bar{z}_{ij}}|\Psi_0(Z,\bar{Z})|^2 = 0, \tag{B.7}$$

given the general form of Eq. (B.6), $M$ must satisfy $M \geq 2$. For fermions, due to antisymmetry, $M$ should be larger than 3.

## C  Projection onto four LLs

Using the states defined in Eq. (11) in the subspace of four LLs such that $0 \leq n_i \leq 3$ for i $= 1, 2$, we find a forty dimensional basis $(n_1 + n_2 = n_c + n_r)$. This basis is not an eigenbasis of relative angular momentum $L_r$. In Table 8, we present the set of numbers $\{n_1, n_2, m\}$ used to construct each basis state $|I\rangle_F$. We express these states (in which subscript $F$ is dropped for brevity) in terms of $|n_c, n_r, 2j - m, m\rangle$ (see Section 2.1),

$$|I\rangle = G_{\pm}^{n_1, n_2} |0, 0, 2j - m, m\rangle = \sum_{n_c, n_r} C_{n_c n_r} |n_c, n_r, 2j - m, m\rangle, \qquad (C.1)$$

where

$$\langle z_c, \bar{z}_c; z_r, \bar{z}_r | n_c, n_r, 2j - m, m\rangle = \phi_{n_c, 2j - m}^c(z_c, \bar{z}_c) \, \phi_{n_r, m}^r(z_r, \bar{z}_r), \qquad (C.2)$$

with Landau orbital defined as [51] $(\vartheta = c, r)$

$$\phi_{n,s}^{\vartheta}(z, \bar{z}) = \frac{(-1)^n \sqrt{n!} \, e^{-\frac{z\bar{z}}{4\ell_{\vartheta}^2}}}{\sqrt{2\pi\ell_{\vartheta}^2} \, \sqrt{2^{s-n}s!}} \left(\frac{z}{\ell_{\vartheta}}\right)^{s-n} L_n^{s-n}\left(\frac{z\bar{z}}{2\ell_{\vartheta}^2}\right). \qquad (C.3)$$

Here, $L_n^{s-n}(x)$ is the associated Laguerre polynomial. Due to the fermionic nature of the states $|I\rangle$, $L_r/\hbar = m - n_r \in$ odd. Then, the basis vectors are given by

$$|1\rangle = |0, 0, 2j - 1, 1\rangle,$$
$$|2\rangle = |0, 1, 2j, 0\rangle,$$
$$|3\rangle = |0, 1, 2j - 2, 2\rangle,$$
$$|4\rangle = |1, 0, 2j - 1, 1\rangle,$$
$$|5\rangle = \frac{1}{\sqrt{2}}(|2, 0, 2j - 1, 1\rangle - |0, 2, 2j - 1, 1\rangle),$$
$$|6\rangle = \frac{1}{\sqrt{2}}(|2, 0, 2j - 3, 3\rangle - |0, 2, 2j - 3, 3\rangle),$$
$$|7\rangle = \frac{1}{\sqrt{2}}(|2, 0, 2j - 1, 1\rangle + |0, 2, 2j - 1, 1\rangle),$$
$$|8\rangle = \frac{1}{\sqrt{2}}(|2, 0, 2j - 3, 3\rangle + |0, 2, 2j - 3, 3\rangle),$$
$$|9\rangle = |1, 1, 2j, 0\rangle,$$
$$|10\rangle = |1, 1, 2j - 2, 2\rangle,$$
$$|11\rangle = \frac{1}{2}(\sqrt{3}|3, 0, 2j - 1, 1\rangle - |1, 2, 2j - 1, 1\rangle),$$
$$|12\rangle = \frac{1}{2}(\sqrt{3}|3, 0, 2j - 3, 3\rangle - |1, 2, 2j - 3, 3\rangle),$$
$$|13\rangle = \frac{1}{2}(|2, 1, 2j, 0\rangle - \sqrt{3}|0, 3, 2j, 0\rangle),$$
$$|14\rangle = \frac{1}{2}(|2, 1, 2j - 2, 2\rangle - \sqrt{3}|0, 3, 2j - 2, 2\rangle),$$
$$|15\rangle = \frac{1}{2}(|2, 1, 2j - 4, 4\rangle - \sqrt{3}|0, 3, 2j - 4, 4\rangle),$$
$$|16\rangle = \frac{1}{\sqrt{64}}(\sqrt{24}|4, 0, 2j - 1, 1\rangle + \sqrt{24}|0, 4, 2j - 1, 1\rangle$$
$$- 4|2, 2, 2j - 1, 1\rangle),$$

$$|17\rangle = \frac{1}{\sqrt{64}}(\sqrt{24}|4,0,2j-3,3\rangle + \sqrt{24}|0,4,2j-3,3\rangle$$
$$-4|2,2,2j-3,3\rangle),$$

$$|18\rangle = \frac{1}{\sqrt{64}}(\sqrt{24}|4,0,2j-5,5\rangle + \sqrt{24}|0,4,2j-5,5\rangle$$
$$-4|2,2,2j-5,5\rangle),$$

$$|19\rangle = \frac{1}{\sqrt{24}}(\sqrt{3!}|3,0,2j-1,1\rangle + 3\sqrt{2}|1,2,2j-1,1\rangle),$$

$$|20\rangle = \frac{1}{\sqrt{24}}(\sqrt{3!}|3,0,2j-3,3\rangle + 3\sqrt{2}|1,2,2j-3,3\rangle),$$

$$|21\rangle = \frac{1}{\sqrt{24}}(\sqrt{3!}|0,3,2j,0\rangle + 3\sqrt{2}|2,1,2j,0\rangle),$$

$$|22\rangle = \frac{1}{\sqrt{24}}(\sqrt{3!}|0,3,2j-2,2\rangle + 3\sqrt{2}|2,1,2j-2,2\rangle),$$

$$|23\rangle = \frac{1}{\sqrt{24}}(\sqrt{3!}|0,3,2j-4,4\rangle + 3\sqrt{2}|2,1,2j-4,4\rangle),$$

$$|24\rangle = \frac{1}{\sqrt{12}}(\sqrt{3!}|1,3,2j,0\rangle - \sqrt{3!}|3,1,2j,0\rangle),$$

$$|25\rangle = \frac{1}{\sqrt{12}}(\sqrt{3!}|1,3,2j-2,2\rangle - \sqrt{3!}|3,1,2j-2,2\rangle),$$

$$|26\rangle = \frac{1}{\sqrt{12}}(\sqrt{3!}|1,3,2j-4,4\rangle - \sqrt{3!}|3,1,2j-4,4\rangle),$$

$$|27\rangle = \frac{1}{\sqrt{48}}(\sqrt{4!}|4,0,2j-1,1\rangle - \sqrt{4!}|0,4,2j-1,1\rangle),$$

$$|28\rangle = \frac{1}{\sqrt{48}}(\sqrt{4!}|4,0,2j-3,3\rangle - \sqrt{4!}|0,4,2j-3,3\rangle),$$

$$|29\rangle = \frac{1}{\sqrt{48}}(\sqrt{4!}|4,0,2j-5,5\rangle - \sqrt{4!}|0,4,2j-5,5\rangle),$$

$$|30\rangle = \frac{1}{\sqrt{192}}(\sqrt{4!}|4,1,2j,0\rangle + \sqrt{5!}|0,5,2j,0\rangle$$
$$-4\sqrt{3}|2,3,2j,0\rangle),$$

$$|31\rangle = \frac{1}{\sqrt{192}}(\sqrt{4!}|4,1,2j-2,2\rangle + \sqrt{5!}|0,5,2j-2,2\rangle$$
$$-4\sqrt{3}|2,3,2j-2,2\rangle),$$

$$|32\rangle = \frac{1}{\sqrt{192}}(\sqrt{4!}|4,1,2j-4,4\rangle + \sqrt{5!}|0,5,2j-4,4\rangle$$
$$-4\sqrt{3}|2,3,2j-4,4\rangle),$$

$$|33\rangle = \frac{1}{\sqrt{192}}(\sqrt{4!}|4,1,2j-6,6\rangle + \sqrt{5!}|0,5,2j-6,6\rangle$$
$$-4\sqrt{3}|2,3,2j-6,6\rangle),$$

$$|34\rangle = \frac{1}{\sqrt{192}}(\sqrt{5!}|5,0,2j-1,1\rangle + \sqrt{4!}|1,4,2j-1,1\rangle$$
$$-4\sqrt{3}|3,2,2j-1,1\rangle),$$

$$|35\rangle = \frac{1}{\sqrt{192}}(\sqrt{5!}|5,0,2j-3,3\rangle + \sqrt{4!}|1,4,2j-3,3\rangle$$
$$-4\sqrt{3}|3,2,2j-3,3\rangle),$$

$$|36\rangle = \frac{1}{\sqrt{192}}(\sqrt{5!}|5,0,2j-5,5\rangle + \sqrt{4!}|1,4,2j-5,5\rangle$$
$$-4\sqrt{3}|3,2,2j-5,5\rangle),$$

$$|37\rangle = \frac{1}{\sqrt{2304}}(\sqrt{6!}|6,0,2j-1,1\rangle$$
$$-3\sqrt{4!}\sqrt{2}|4,2,2j-1,1\rangle - \sqrt{6!}|0,6,2j-1,1\rangle$$
$$+3\sqrt{4!}\sqrt{2}|2,4,2j-1,1\rangle),$$

$$|38\rangle = \frac{1}{\sqrt{2304}}(\sqrt{6!}|6,0,2j-3,3\rangle$$
$$-3\sqrt{4!}\sqrt{2}|4,2,2j-3,3\rangle - \sqrt{6!}|0,6,2j-3,3\rangle$$
$$+3\sqrt{4!}\sqrt{2}|2,4,2j-3,3\rangle),$$

$$|39\rangle = \frac{1}{\sqrt{2304}}(\sqrt{6!}|6,0,2j-5,5\rangle$$
$$-3\sqrt{4!}\sqrt{2}|4,2,2j-5,5\rangle - \sqrt{6!}|0,6,2j-5,5\rangle$$
$$+3\sqrt{4!}\sqrt{2}|2,4,2j-5,5\rangle),$$

$$|40\rangle = \frac{1}{\sqrt{2304}}(\sqrt{6!}|6,0,2j-7,7\rangle$$
$$-3\sqrt{4!}\sqrt{2}|4,2,2j-7,7\rangle - \sqrt{6!}|0,6,2j-7,7\rangle$$
$$+3\sqrt{4!}\sqrt{2}|2,4,2j-7,7\rangle). \tag{C.4}$$

Notice that all these 40 vectors contain a vector component $|n_c, n_r, 2j-m, m\rangle$ with $L_r = \pm 1\hbar$. These vectors are the ones that lead to non-vanishing matrix elements of the TK Hamiltonian. States $|I\rangle$ with $m > 7$ components do not contribute to the subspace of positive energy eigenvalues since $\max(n_r) = 6$.

We have diagonalized the interaction potential in this basis and obtained only 12 non-zero positive eigenvalues $E_\xi$, $\xi = 1, \cdots, 12$. The expansion coefficients $\Lambda_I^\xi$ are presented as elements of a $40 \times 12$ matrix in Table 9, with $(I, \xi)$ specifying number of rows and columns, respectively.

# D  The boundary root pattern

Here, we follow the method of Section 3.2.3 to establish the left boundary conditions for a generic $N$-particle zero-energy ground state $|\Psi_0\rangle$ with $N_L = 4$ LLs. By left boundary conditions, we specifically refer to the allowed negative angular momentum orbitals of $|\Psi_{\text{root}}\rangle$. Two-fermion operators $\mathcal{T}_j^{\xi-}$ and their linear superpositions annihilate $|\Psi_0\rangle$. For the present purposes, a convenient linear superposition, $\mathcal{T}_j^{\xi'}$, is shown in Table 10. Near the boundary, since the two-body basis elements must obey

$$n_1 + n_2 - m \geq -2j, \tag{D.1}$$

not all two-fermion operators $\mathcal{T}_j^{\xi'}$ satisfy the constraint. For example, when $j = -3 + 1/2 = -5/2$ only two-fermion operators with $\xi' = 1, 2$ are well defined. When $j = -2$, the well defined two-fermion operators are $\xi' = 1, \cdots, 4$. For $j = -3/2$, we get $\xi' = 1, \cdots, 6$. And for $j = -1$, we get $\xi' = 1, \cdots, 8$.

To study the multiplicity of orbitals with $-3 < j < 0$, we can utilize Eq. (65). Note that the smallest angular momentum orbital $j = -3$ can only be occupied by a single electron.

Table 9: The matrix $\Lambda_I^\xi$, with $\xi = 1, \cdots, 12$ and $I = 1, \cdots, 40$.

$$
\begin{pmatrix}
-\frac{4}{\sqrt{35}} & 0 & 0 & 0 & 0 & 0 & 0 & 0 & 0 & 0 & 0 & 0 \\
0 & \frac{\sqrt{17}-1}{\sqrt{30}} & 0 & 0 & -\frac{\sqrt{17}+1}{\sqrt{30}} & 0 & 0 & 0 & 0 & 0 & 0 & 0 \\
4\sqrt{\frac{2}{35}} & 0 & 0 & 0 & 0 & 0 & 0 & 0 & 0 & 0 & 0 & 0 \\
0 & 0 & 2\sqrt{\frac{2}{5}} & 0 & 0 & 0 & 0 & 0 & 0 & 0 & 0 & 0 \\
0 & 2\sqrt{\frac{2}{15}} & 0 & 0 & 2\sqrt{\frac{2}{15}} & 0 & 0 & 0 & 0 & 0 & 0 & 0 \\
2\sqrt{\frac{6}{35}} & 0 & 0 & 0 & 0 & 0 & 0 & 0 & 0 & 0 & 0 & 0 \\
0 & -\sqrt{\frac{2}{15}}\left(\sqrt{17}-3\right) & 0 & 0 & \sqrt{\frac{2}{15}}\left(\sqrt{17}+3\right) & 0 & 0 & 0 & 0 & 0 & 0 & 0 \\
-2\sqrt{\frac{6}{35}} & 0 & 0 & 0 & 0 & 0 & 0 & 0 & 0 & 0 & 0 & 0 \\
0 & 0 & 0 & \frac{1-\sqrt{33}}{4\sqrt{2}} & 0 & 0 & \frac{\sqrt{33}+1}{4\sqrt{2}} & 0 & 0 & 0 & 0 & 0 \\
0 & 0 & -\frac{4}{\sqrt{5}} & 0 & 0 & 0 & 0 & 0 & 0 & 0 & 0 & 0 \\
0 & 0 & 0 & -1 & 0 & 0 & -1 & 0 & 0 & 0 & 0 & 0 \\
0 & 0 & -\sqrt{\frac{6}{5}} & 0 & 0 & 0 & 0 & 0 & 0 & 0 & 0 & 0 \\
0 & 0 & 0 & 0 & 0 & \frac{5-\sqrt{57}}{4\sqrt{3}} & 0 & 0 & 0 & \frac{\sqrt{57}+5}{4\sqrt{3}} & 0 & 0 \\
0 & -\frac{\sqrt{17}+7}{2\sqrt{30}} & 0 & 0 & -\frac{7-\sqrt{17}}{2\sqrt{30}} & 0 & 0 & 0 & 0 & 0 & 0 & 0 \\
-4\sqrt{\frac{3}{35}} & 0 & 0 & 0 & 0 & 0 & 0 & 0 & 0 & 0 & 0 & 0 \\
0 & 0 & 0 & 0 & 0 & -\sqrt{\frac{2}{3}} & 0 & 0 & 0 & -\sqrt{\frac{2}{3}} & 0 & 0 \\
0 & -\frac{2}{\sqrt{5}} & 0 & 0 & -\frac{2}{\sqrt{5}} & 0 & 0 & 0 & 0 & 0 & 0 & 0 \\
-\sqrt{\frac{6}{7}} & 0 & 0 & 0 & 0 & 0 & 0 & 0 & 0 & 0 & 0 & 0 \\
0 & 0 & 0 & \frac{1}{2}\left(\sqrt{11}-\sqrt{3}\right) & 0 & 0 & -\frac{1}{2}\left(\sqrt{3}+\sqrt{11}\right) & 0 & 0 & 0 & 0 & 0 \\
0 & 0 & 3\sqrt{\frac{2}{5}} & 0 & 0 & 0 & 0 & 0 & 0 & 0 & 0 & 0 \\
0 & 0 & 0 & 0 & 0 & \frac{1}{4}\left(5-\sqrt{57}\right) & 0 & 0 & 0 & \frac{1}{4}\left(\sqrt{57}+5\right) & 0 & 0 \\
0 & \frac{3\sqrt{17}-11}{2\sqrt{10}} & 0 & 0 & -\frac{3\sqrt{17}+11}{2\sqrt{10}} & 0 & 0 & 0 & 0 & 0 & 0 & 0 \\
\frac{4}{\sqrt{35}} & 0 & 0 & 0 & 0 & 0 & 0 & 0 & 0 & 0 & 0 & 0 \\
0 & 0 & 0 & 0 & 0 & 0 & 0 & \frac{1}{8}\left(\sqrt{89}-5\right) & 0 & 0 & -\frac{1}{8}\left(\sqrt{89}+5\right) & 0 \\
0 & 0 & 0 & \frac{1}{8}\left(5\sqrt{3}+\sqrt{11}\right) & 0 & 0 & \frac{1}{8}\left(5\sqrt{3}-\sqrt{11}\right) & 0 & 0 & 0 & 0 & 0 \\
0 & 0 & \frac{4}{\sqrt{5}} & 0 & 0 & 0 & 0 & 0 & 0 & 0 & 0 & 0 \\
0 & 0 & 0 & 0 & 0 & \frac{\sqrt{57}-9}{3\sqrt{2}} & 0 & 0 & 0 & -\frac{\sqrt{57}+9}{3\sqrt{2}} & 0 & 0 \\
0 & \frac{\sqrt{17}-1}{\sqrt{15}} & 0 & 0 & -\frac{\sqrt{17}+1}{\sqrt{15}} & 0 & 0 & 0 & 0 & 0 & 0 & 0 \\
2\sqrt{\frac{2}{7}} & 0 & 0 & 0 & 0 & 0 & 0 & 0 & 0 & 0 & 0 & 0 \\
0 & 0 & 0 & 0 & 0 & 0 & 0 & 0 & \frac{1}{4}\left(\sqrt{43}-3\sqrt{3}\right) & 0 & 0 & -\frac{1}{4}\left(3\sqrt{3}+\sqrt{43}\right) \\
0 & 0 & 0 & 0 & \frac{1}{12}\left(\sqrt{57}+3\right) & 0 & 0 & 0 & \frac{1}{12}\left(3-\sqrt{57}\right) & 0 & 0 & 0 \\
0 & \frac{\sqrt{17}+15}{4\sqrt{15}} & 0 & 0 & \frac{15-\sqrt{17}}{4\sqrt{15}} & 0 & 0 & 0 & 0 & 0 & 0 & 0 \\
2\sqrt{\frac{3}{7}} & 0 & 0 & 0 & 0 & 0 & 0 & 0 & 0 & 0 & 0 & 0 \\
0 & 0 & 0 & 0 & 0 & 0 & 0 & 1 & 0 & 0 & 1 & 0 \\
0 & 0 & 0 & 1 & 0 & 0 & 1 & 0 & 0 & 0 & 0 & 0 \\
0 & 0 & 1 & 0 & 0 & 0 & 0 & 0 & 0 & 0 & 0 & 0 \\
0 & 0 & 0 & 0 & 0 & 0 & 0 & 1 & 0 & 0 & 0 & 1 \\
0 & 0 & 0 & 0 & 0 & 1 & 0 & 0 & 0 & 1 & 0 & 0 \\
0 & 1 & 0 & 0 & 1 & 0 & 0 & 0 & 0 & 0 & 0 & 0 \\
1 & 0 & 0 & 0 & 0 & 0 & 0 & 0 & 0 & 0 & 0 & 0
\end{pmatrix}
$$

When orbitals with $j = -2$ are occupied by two electrons, the resultant root state with a single coefficient $C_{n_1,n_2}^j = C_{2,3}^{-2}$ has to satisfy 4 constraints from Eq. (65). As a result, the multiplicity two in $j = -2$ orbitals is not allowed. Similarly, $j = -1$ orbitals occupied by two electrons cannot be allowed since such a root state has only 3 coefficients $C_{n_1,n_2}^{-1}$, which cannot simultaneously satisfy the 8 constraints of Eq. (65). As a result, in a generic root state, we conclude that multiplicity of $j < 0$ orbitals can be at most one. That is, $111\cdots$, $110\cdots$, $101\cdots$, $011\cdots$, $100\cdots$, $010\cdots$, or $001\cdots$, where $\cdots$ refers to some bulk root pattern with orbitals $j \geq 0$.

The root pattern $111\cdots$ can have 6 coefficients. Steps similar to those that led to Eq. (64) can be followed to obtain constraints governing the appearance of Slater determinants of the form $|\mathfrak{n}\rangle = c_{n_1,j-k'}^\dagger c_{n_2,j+k'}^\dagger |\mathfrak{n}_2\rangle$ in the root state. However, the corresponding root state must simultaneously satisfy the $j = -5/2, -2, -3/2$ constraints which indicates that such a pattern is not possible. The patterns $110\cdots$, $101\cdots$, and $011\cdots$ have, respectively, 2, 4, and 6 coefficients that are needed to satisfy, in each case, just as many constraints.

Invoking the linear independence of the equations, in the homogeneous set of linear equations, leads to only the trivial solution, where all the coefficients are zero. As a result, none of the patterns with two particles occupying the $j < 0$ orbitals are allowed.

Consequently, in a root state satisfying the EPP conditions in the bulk ($j \geq 0$) the boundary orbitals ($j < 0$) can only be occupied with a single particle. For example, pattern $100\cdots$ is generally allowed ($N \geq 2$). Fusing this admissible left boundary and the densest bulk patterns, and assuming no change in the bulk pattern to ensure the existence of no excitation, we obtain that the densest pattern consistent with the EPP is the root pattern

$$100200200\ldots2002. \tag{D.2}$$

Table 10: $T_j^I \equiv T_{j,m}^{n_1,n_2,-}$, for $I$ $(n_1, n_2\ m)$ given in Table 8. The ground state $|\Psi_0\rangle$ satisfies 12 linear constraints, $\mathcal{T}_j^{\xi'}|\Psi_0\rangle = 0$, for $\xi' = 1,\cdots,12$. These constraints are established using the matrix $\Lambda_I^\xi$ in Table 9.

| $\xi'$ | $\mathcal{T}_j^{\xi'}$ |
|---|---|
| 1 | $T_j^{30}$ |
| 2 | $T_j^{37}$ |
| 3 | $T_j^{24}$ |
| 4 | $T_j^{34}$ |
| 5 | $\sqrt{3}T_j^{13} + 3T_j^{21} - 2\sqrt{2}T_j^{27} - T_j^{31}$ |
| 6 | $-\frac{2}{\sqrt{3}}T_j^{13} + \sqrt{\frac{2}{3}}T_j^{16} - 2T_j^{21} + 2\sqrt{2}T_j^{27} - T_j^{38}$ |
| 7 | $\sqrt{6}T_j^9 - 4T_j^{19} - T_j^{25}$ |
| 8 | $T_j^{11} + 3\sqrt{3}T_j^{19} - T_j^{35} - 2\sqrt{2}T_j^9$ |
| 9 | $\sqrt{2}T_j^{14} - 2\sqrt{2}T_j^2 - 3\sqrt{6}T_j^{22} - 4T_j^{28} - T_j^{32} + 4\sqrt{2}T_j^7$ |
| 10 | $\frac{4}{\sqrt{5}}T_j^{10} + \sqrt{\frac{6}{5}}T_j^{12} - 3\sqrt{\frac{2}{5}}T_j^{20} - \frac{4}{\sqrt{5}}T_j^{26} - T_j^{36} - 2\sqrt{\frac{2}{5}}T_j^4$ |
| 11 | $-2\sqrt{\frac{2}{15}}T_j^{14} + \frac{2}{\sqrt{5}}T_j^{17} + 8\sqrt{\frac{2}{15}}T_j^2 + 14\sqrt{\frac{2}{5}}T_j^{22} + \frac{16}{\sqrt{15}}T_j^{28} - T_j^{39} - 2\sqrt{\frac{2}{15}}T_j^5 - 6\sqrt{\frac{6}{5}}T_j^7$ |
| 12 | $\frac{4}{\sqrt{35}}T_j^1 + 4\sqrt{\frac{3}{35}}T_j^{15} + \sqrt{\frac{6}{7}}T_j^{18} - \frac{4}{\sqrt{35}}T_j^{23} - 2\sqrt{\frac{2}{7}}T_j^{29} - 4\sqrt{\frac{2}{35}}T_j^3 - 2\sqrt{\frac{3}{7}}T_j^{33} - T_j^{40} - 2\sqrt{\frac{6}{35}}T_j^6 + 2\sqrt{\frac{6}{35}}T_j^8$ |

# E Braiding statistics: Proof of Eq. (75)

$$\psi_{n,j} = \frac{1}{\sqrt{L}}\sum_{j'}\omega^{jj'}\bar\psi_{n,j'} = \frac{1}{\sqrt{L}}\sum_{j',s'}e^{i\frac{2\pi}{L}jj'}\bar\phi_{n,j'+s'L} = \frac{1}{\sqrt{L}}\sum_{j',s,s'}e^{i\frac{2\pi}{L}(j+sL)(j'+s'L)}\bar\phi_{n,j'+s'L}$$

$$= \frac{1}{\sqrt{L}}\sum_{s,j'}e^{i\frac{2\pi}{L}(j+sL)j'}\bar\phi_{n,j'}$$

$$\implies \psi_{n,j} = \frac{1}{\sqrt{L}}\sum_{s,j'}e^{i\frac{2\pi}{L}(j+sL)j'}e^{-i2\pi j'x_\Delta}H_n\left(\sqrt{-i2\pi\tau L}\left(y_\Delta - \frac{j'}{L}\right)\right)e^{i\pi\tau L\left(y_\Delta - \frac{j'}{L}\right)^2}$$

$$\implies \psi_{n,j} = \frac{1}{\sqrt{L}}\sum_{s,j'}e^{i2\pi L\left(y_\Delta - \frac{j'}{L} - y_\Delta\right)\left(x_\Delta - \frac{j}{L} - s\right)}H_n\left(\sqrt{-i2\pi\tau L}\left(y_\Delta - \frac{j'}{L}\right)\right)e^{i\pi\tau L\left(y_\Delta - \frac{j'}{L}\right)^2}$$

$$\implies \psi_{n,j} = e^{-i2\pi Lx_\Delta y_\Delta}\sum_s e^{2\pi(j+sL)y_\Delta}\frac{1}{\sqrt{L}}\sum_{j'}e^{i2\pi L\left(y_\Delta - \frac{j'}{L}\right)\left(x_\Delta - \frac{j}{L} - s\right)}$$

$$\times H_n\left(\sqrt{-i2\pi\tau L}\left(y_\Delta - \frac{j'}{L}\right)\right)e^{i\pi\tau L\left(y_\Delta - \frac{j'}{L}\right)^2}$$

$$\implies \psi_{n,j} = e^{-i2\pi Lx_\Delta y_\Delta}\sum_s e^{2\pi(j+sL)y_\Delta}H_n\left(\sqrt{i2\pi\frac{L}{\tau}}\left(x - \frac{j}{L} - s\right)\right)e^{-i\pi\frac{L}{\tau}\left(x - \frac{j}{L} - s\right)^2}$$

$$= e^{-i2\pi Lx_\Delta y_\Delta}\sum_s \phi_{n,j+sL}\,.$$

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
