# Peer review of "Partons as unique ground states of quantum Hall parent Hamiltonians: The case of Fibonacci anyons"

_SciPost Physics, doi:SciPost Phys. 15, 043 (2023)_

## Round 1 · Referee Report · Mikael Fremling (Referee 1) · 2022-9-30

Report

I have read the paper "Partons as unique ground states of quantum Hall parent Hamiltonians: The case of Fibonacci anyons" with great interest.

The paper builds on and adds to the growing literature concerned with distilling the topological information encoded in FQH wave functions. Early ideas in this field were the introduction of Jack Polynomials and squeezed states, but these constructions are usually restricted to the Lowest Landau level (LLL). The Entangled Pauli Principle (EPP) has been introduced to capture more elaborate entanglement scenarios.

The authors present a framework based on the Entangled Pauli Principle (EPP), which allows them to construct microscopic, multiple Landau level quantum Hall fluids that are parton-like and whose excitations display either Abelian or non-Abelian braiding statistics. They further argue that these states are the densest zero-energy ground states of frustration-free and positive semi-definite parent Hamiltonians.

The authors further show how the EPP faithfully encodes topological braiding information and how the braiding statistics is then extracted, especially on the torus.

The authors show that when the closed-shell condition is satisfied for the partons, the densest zero-energy mode is a unique parton state. Of special interest is a conjecture that parton-like states generally span the subspace of many-body wave functions with a two-body M-clustering property. This conjecture would allow for a much more efficient representation of microscopic wave functions than a slater determinant expansion.

With this paper, the authors introduce several new concepts and results, and significantly forwards the state of knowledge in the field. I, therefore, recommend this paper to be published in SciPost after my questions and remarks (below) have been adequately answered.

  • The abstract begins with: "We present microscopic, multiple Landau level, (frustration-free and positive semi-definite) parent Hamiltonians whose ground states, realizing different quantum Hall fluids, are parton-like and whose excitations display either Abelian or non-Abelian braiding statistics."

From reading the main text, I understand that you argue for the EPP forms, which can give zero-energy solutions for frustration-free parent Hamiltonians. I could, however, not find a discussion or reference to how these Hamiltonians would look like, nor a recipe to construct them.

If it is in the paper, could this then be pointed out more clearly? If it is not, perhaps the abstract should be changed slightly. I note that the authors on and off work with the TK Hamiltonian, but I understand this construction to be more general than that.

  • A general question on the importance of the LL structure: In the later section, the authors focus on deriving results for the N_L=4 lowest LL:s. How important is the LL structure itself to these calculations? Can one replace the ladder operator algebra that moves particles up and down between LLs with, say, two internal spin-1/2 degrees of freedom and the corresponding S^(+/-) operators? This would be relevant for layered materials where each layer is in the LLL.

  • Between eqn. (8) and eqn. (9): The authors write that the eigenvalues of $n^b_{c,r}$ are $2j-m$ and $m$ respectively. Could some text be added here to point out that $2j = n^b_1 + b^b_2$ or similar, and that $j$ could be a half-integer? When reading the text as it stands, I was very confused as to where $j$ came from.

  • Around eqn. (12) and eqn. (13): In equation (13), the authors introduce the determinant $D_{...}(1,2)$, which they claim is being used in equations (12). However, as it is currently written since this is highly confusing since (12) is an equation for "kets" and (13) is a fave function. I can see that if one applied the "bra" <1,2| from the left in (12), then the object $D$ in (13) would appear, but it is not there at the moment.

Further, I find the notation with \alpha_1 and \alpha_2 quite inadequate since they are only indirectly defined. By comparing eqns. (12) and (13) I come to the conclusion that $\alpha_1 = n_1,j-k$ and $\alpha_2 = n_2,j+k$, but this is never written out explicitly.

  • Equation (15): This equation is suggested as a natural map for $D_\alpha1,\alpha2$. However, since the $D$ in equation (13) never appears in any other equation except for (15), it is unclear how to interpret the suggested map.

  • Above Figure 1: The authors write: "For finite size systems, the number of angular momentum orbitals in each LL is restricted by the number L of available distinct single particle angular momentum modes." Could a clarification be made here that $L$ is refereeing to the number of angular momentum modes in the highest Landau level and not the lowest Landau level? As a consequence, is it true that $L>=N_L$?

  • First paragraph section III: in the text: "As we will explain, one may indeed precisely find these zero-energy states for any given number of particles N at filling fractions ν = (N −1)/(L−1)." I find this statement weird as it would imply that zero energy states exist at any density. This seems to be contradictory to Figure 2, which makes it clear that there exists an $N$ for which the energy is always strictly positive. Also, i'm confused as to why there is a -1 shift in $N$ and $L$.

*Equation (41): Could the authors clarify what assumptions go into equation (41)? For instance, is there an assumption that $L$ is kept constant as $N$ is changed? Keeping $L$ fixed seems important e.g., to be able to draw the diagram in Fig 2 and be able to define the densest zero energy state.

*Figure 3 and text below eqn (49): 1) In the figure, can the arrows pointing from one color to the other be made larger (they are hardly visible)? 2) The text under equation (49) claims that the "blue state" is non-expandable. Is this, however, true? Can i not obtain this state (1_3,0,2_{1,3},0,3_{0,2,3}) from an inward squeeze of (1_3,1_1,0,1_3,3_{0,2,3}). If this is not the case, some extra text is needed explaining why. 3) Why are the $L=2$ orbitals included in the figure (and the state description?). If they are available, the blue state should also be reachable from an inward squeeze of a state that has electrons in the L=2 orbitals. To avoid confusion, I suggest that the $L=2$ orbitals are removed from the figure and the state description.

  • Between equations (52) and (53): Underneath equation (52) there is the sentence: "The corresponding pseudo-fermion basis states are $\bar z^{n_i} z^{j+n_i}$". This sentence seems detached from the rest of the paragraph, as a) it is not clear why the functional form of the basis states is relevant, and b) z=x+iy is not used at all in this entire section (apart from this sentence). Can it be removed?

  • Below eqn (53): a) The text says: "There are 12 coefficients up to an overall normalization to satisfy 12 linear constraints from Eq. (30)." I here wonder how the counting is made, are $n$ and $n^\prime$ considered independent variables, or not. Naively I would have thought that $n<n^\prime$, which should give 6 states. Alternatively I could let $n$ and $n^\prime$ be independent which should give $4 x 4 = 16 $ states. How do I get 12? b) The text further reads: "For a single angular momentum site constraints ξ = 9, 12 are not independent." This sentence would have been fine, except that the ξ-numbering is not clear. Looking at eqn (25) and (27), I see ξ appearing, but it is unclear which values it takes and if these are the same as the ξ=1,...,12 that are implicitly used in the text under equation (53). Perhaps they can be listed in the appendix?

*Equation (67): Can it be written out that the orbitals \phi in equation (67) are defined on a cylinder?

*Above (68): The authors introduce a gauge that is not the symmetric gauge nor the Landau gauge. In the literature, this gauge is often called the "tau-gauge" as it is perpendicular to the "tau"-vector.

  • Page 14 bottom: "Moreover, for L ∆ = 0, there are anti-unitary operators that implement the combination of a mirror symmetry (in x or y) with time-reversal symmetry." For completeness, this is also present for the hexagon, with $L_\Delta = L/2$ and $|\tau$=1.

  • Page 16 left column: There are two "equations" with string sequences 200200200200200200200200200200200200200 200110110020020011011002002001101100200 and 200200200200200200200 110110110110110110110 110101101011011010110 which are long and hard to read. Can some tricks with color coding or other visual aids be inserted to highlight where the domain walls are forming?

  • Page 21, middle of the page: This is not a complete sentence: "From the completeness of our coherent states, given that the operation of inversion as described will produce a zero mode in the same topological sector with a quasihole localized at 2h I − h."

  • Around eqn. (127): a) The text above equation (127) refers back to the basis mentioned with regards to the pseudofermions below eqn (52). However, since the form in (127) was never used there (and I still recommend removing it), this reference becomes dangling. If the authors choose to keep this reference as is, then it should at least explicitly refer to eqn (52) to allow the reader to find the pseudofermions. b) If the reference is kept, it is worthwhile to remind the reader that s_i=j+n_i. c) a technical remark may be warranted here: That is, equation (127) (when the gaussian factor is restored) is not the same as the LL basis (since different \phi_\alpha are not orthogonal).

  • Around eqn. (129): It may be advisable to mention that these calculations are specific for the symmetric gauge, and e.g. eqn (129) is only valid in that (symmetric) gauge.

  • eqn. (148): I never understood where equation 148 came from. For instance, for N_L=2 and M=3, why am I not allowed to multiply three \nu=2 wave functions? These states would also be a product of closed-shell wave functions, right?

Some minor language issues:

  • Above equation (19): The text "the average distance between electrons is" should be changed to "the average distance between neighboring electrons is"

*Page 8, Top first column: The text reads: "Since the size of LL orbitals is directly associated with the magnitude of its angular momentum....". Can a different word than "size" be used here? All the LL orbitals are equally large (cover the same area), but they have their density peaks at different radii.

  • Conclusions, second line: Inconsistent spelling of ansatz.

  • Conclusions, middle of page 43:: Should there be an extra "and" in "Laughlin's wave function can be expressed as the product of the lowest LL (LLL) wave function with M Slater determinants of holomorphic (or anti-holomorphic) wave functions several filling fractions of the form 1/M (odd M ), which are experimentally observed." such that it reads "Laughlin's wave function can be expressed as the product of the lowest LL (LLL) wave function with M Slater determinants of holomorphic (or anti-holomorphic) wave functions and several filling fractions of the form 1/M (odd M ), which are experimentally observed."?

  • validity: -
  • significance: -
  • originality: -
  • clarity: -
  • formatting: -
  • grammar: -

Author:  Gerardo Ortiz  on 2022-12-19  [id 3159]

(in reply to Report 1 by Mikael Fremling on 2022-09-30)

Response in file "Response.pdf"

Attachment:

Response.pdf

---

## Round 1 · Referee Report · Anonymous (Referee 2) · 2023-1-8

Strengths

1- Very through 2-Potentially novel (outside standard CFT)

Weaknesses

1- relevance and novelty unclear

Report

The authors study a frustration-free Hamiltonian that has ground states that are parton-like. The excitations have braiding statistics. This is shown by reducing the parton states to root states.

The main result comes from Eq.(26) (frustration-free), i.e. the parent Hamiltonian is non-interacting allowing for very high degeneracy.

I find the paper interesting and a nice addition to the line of investigation pursued by the authors. But it remains unclear to me whether the work meets the novelty standards of SciPost Physics. In particular in comparison with Ref.[15].

Requested changes

1- Discuss novelty especially compared to Ref. [15]

  • validity: high
  • significance: ok
  • originality: ok
  • clarity: good
  • formatting: excellent
  • grammar: excellent

Author:  Gerardo Ortiz  on 2023-01-23  [id 3259]

(in reply to Report 2 on 2023-01-08)
Category:
answer to question

Dear Editor, Referee:

Our response to the second referee is in the attached pdf file (ResponseII.pdf)
Best regards,
Gerardo

Attachment:

ResponseII.pdf

---

## Round 1 · Referee Report · Bo Yang (Referee 3) · 2023-1-31

Strengths

Good overview and very useful technical details on some of the fundamental understandings of the composite fermion/parton based fractional quantum Hall states.

Weaknesses

Not easy to distill the novel conceptual contribution of this paper, and the technical details may be overwhelming for a more general audience.

Report

This paper is a culmination of a few previous papers by some of the authors in understanding the dynamics of the FQH systems involving multiple Landau levels, and their relations to the Jain series. It is a nice combination of solid results and technical details, together with plausible conjectures. It focuses on the two-body short range TK Hamiltonian and the most interesting results come from the projection to the lowest 4 Landau levels, where non-Abelian Fibonacci states can be realised with just a two-body interaction. In particular with the entangled root configuration (a generalisation of the Jack polynomial formalism), the braiding properties of the Fibonacci anyons can be studied.

Most of the results in this paper are more or less known in the literature (mostly by the same authors), so the main contribution of this work is on the technical aspects. I personally think it is very nice to have a paper detailing the methodologies of establishing the “DNA” of the multiple LL FQH states, the study of the zero energy manifold and the entangled Pauli exclusion principle. It is important to have a more fundamental understanding of the largely phenomenological composite fermion (and the parton) theory in the framework of microscopic Hamiltonians and the conformal symmetry of the quasihole manifold. It is not there yet but the related works by the authors are some of the important steps towards that direction.

Having said that, indeed it is not easy to know (from just reading the paper) what results are “reviews” of previous results, and what are the novel contributions in this paper (in terms of physical ideals, apart from the technical aspects). So it would be very nice if the authors can highlight such novelties very prominently in the introduction.

The technical details require a much longer time to read and digest (which may not appeal to a broader audience apart from those who actually want to work along this direction), but the overall picture is quite clear. One thing I am confused about is the claim “The full two-fermion basis for NL = 4 LLs is of dimension 40”. I suppose the size of the Hilbert space definitely depends on the number of magnetic fluxes, so I must have missed something here (and the notations are a bit dense, though I am not sure if there is a way to simplify). Maybe the authors can make that part more transparent.

Given the usefulness of this work, I recommend it to be published at Scipost.

---

## Round 2 · Referee Report · Bo Yang (Referee 3) · 2023-3-19

Report

The revised manuscript is better in terms of letting the readers understand the new contributions of the current manuscript.

The authors did not respond to one technical questions I asked in the previous report:

“The full two-fermion basis for NL = 4 LLs is of dimension 40”, with respect to the definition of the two-fermion basis. If they are just states containing two fermions, then of course the dimension grows exponentially with the system size.

Thanks to private discussions with Li Chen from WUSTL, I now understand here the authors are defining the two-fermion basis as states containing two fermions that are highest weight (so lifting the center of mass degeneracy) with relative angular momentum penalised by the TK Hamiltonian.

While this definition should be there when we look carefully at the mathematical expressions, it is not obvious and one has to go through those different notations. Thus for readers who are not keen in reproducing the technical machinery and just want to get the general idea, it is better for the "two-fermion basis" to be properly defined by physically intuitive terms.

Requested changes

Proper definition of "the two-fermion basis" as the basis of highest weight (center of mass angular momentum 0) states with relative angular momentum having finite energy cost with respect to the TK Hamiltonian.

  • validity: high
  • significance: high
  • originality: high
  • clarity: good
  • formatting: excellent
  • grammar: excellent

Author:  Gerardo Ortiz  on 2023-04-08  [id 3565]

(in reply to Report 1 by Bo Yang on 2023-03-19)
Category:
answer to question

Dear Dr Yang,
Thank you very much for your positive and encouraging report. We have now added explanation about the way this 40 vectors basis is generated. Essentially, the basis is cutoff by the number of Landau levels one uses (in this case 4 Landau levels) and the maximum m (7 in this case) such that matrix elements of the TK Hamiltonian are non-vanishing. The new version of the manuscript has been uploaded in the arxiv. (Section II C and Appendix C have been modified accordingly).
Best regards,
Gerardo Ortiz

---

## Round 2 · Referee Report · Anonymous (Referee 2) · 2023-3-23

Report

The authors have convinced me of the novelty of their work. I now recommend publication.

---

## Round 2 · Referee Report · Mikael Fremling (Referee 1) · 2023-4-13

Report

I have read the revised parts of the manuscript, And I think the authors have in a satisfactory way addressed all of my concerns.

I now recommend publications in SciPost.

---

## Round 2 · Author Response

Dear Editor,

We thank you for taking care of our paper, and the referees for such positive reviews. It is our
understanding that we are required to put the latest revised version of our manuscript in the arxiv. I have already done it. On the
other hand, we would like to highlight all the changes introduced in the revised manuscript
prompted by the referees comments. It is simpler for us to provide a (color) highlighted version
of our revised manuscript. Is that possible and if so, how do we send such a version to you ?

Response to Third referee:

We thank the referee for a positive review. As the referee emphasized this present work is
part of a program that our group started a few years back. However, this is not a review of
prior work. Our work centers on new original results, including a complete new theory of multivariate
non-holomorphic polynomials relevant for parton-like states, technical results (theorems)
on monotonicity of ground states energies in special k-body Hamiltonians and S-duality, and
derivation of a non-Abelian fluid with Fibonacci topological excitations (rigorously proved
within the coherent state approach), among others. Since we understand that people may
get confused about what is new and what is not, we have modified the Introduction to
make it more clear.

---

## Editorial Decision

published